# PolyMath: Evaluating Mathematical Reasoning in Multilingual Contexts

**Yiming Wang**[1,2]  **Pei Zhang**[2]  **Jialong Tang**[2]  **Haoran Wei**[2]  **Baosong Yang**[2,✉]  **Rui Wang**[1,✉]

**Chenshu Sun**[*]  **Feitong Sun**[*]  **Jiran Zhang**[*]  **Junxuan Wu**[*]  **Qiqian Cang**[*]

**Yichang Zhang**[2]  **Fei Huang**[2,†]  **Junyang Lin**[2]  **Fei Huang**[2,‡]  **Jingren Zhou**[2]

[1]Shanghai Jiao Tong University    [2]Qwen Team, Alibaba Group

✉: Corresponding Author
Email:  [1]{yiming.wang, wangrui12}@sjtu.edu.cn
[2]yangbaosong.ybs@alibaba-inc.com

 https://github.com/QwenLM/PolyMath
 https://hf.co/datasets/Qwen/PolyMath
 https://Qwen-PolyMath.github.io/
[0] https://arxiv.org/abs/2504.18428

## Abstract

In this paper, we introduce **PolyMath**, a multilingual mathematical reasoning benchmark covering 18 languages and 4 easy-to-hard difficulty levels. Our benchmark ensures difficulty comprehensiveness, language diversity, and high-quality translation, making it a highly discriminative multilingual mathematical benchmark in the era of reasoning LLMs. We conduct a comprehensive evaluation for advanced LLMs and find that even Qwen3-235B-A22B-Thinking and Gemini-2.5-pro, achieve only 54.6 and 52.2 benchmark scores, with about 40% accuracy under the highest level. From a language perspective, our benchmark reveals several key challenges of LLMs in multilingual reasoning: (1) Reasoning performance varies widely across languages for current LLMs; (2) Input-output language consistency is low in reasoning LLMs and may be correlated with performance; (3) The thinking length differs significantly by language for current LLMs. Additionally, we demonstrate that controlling the output language in the instructions has the potential to affect reasoning performance, especially for some low-resource languages, suggesting a promising direction for improving multilingual capabilities in LLMs.

## 1 Introduction

The rapid development of Artificial Intelligence (AI) has positioned Large Language Models (LLMs) as a promising path towards achieving Artificial General Intelligence (AGI) [27, 3, 2]. The research focus has recently shifted from fast to slow thinking, transforming the LLM paradigm into reasoning models such as OpenAI-o1 and Deepseek-R1 [18, 14]. This evolution significantly enhances the reasoning capabilities of language models.

---

[*]Equal contribution in alphabetical order. Details of language responsibilities are provided in Appendix C.
[†]Google Scholar ID: 7udAEzMAAAAJ  [‡] Google Scholar ID: 9r98PpoAAAAJ.
[0]Due to space limitations in the camera-ready version of the conference paper, many important analyses are provided in the appendix. For a more seamless reading experience, we recommend this full version.

39th Conference on Neural Information Processing Systems (NeurIPS 2025) Track on Datasets and Benchmarks.

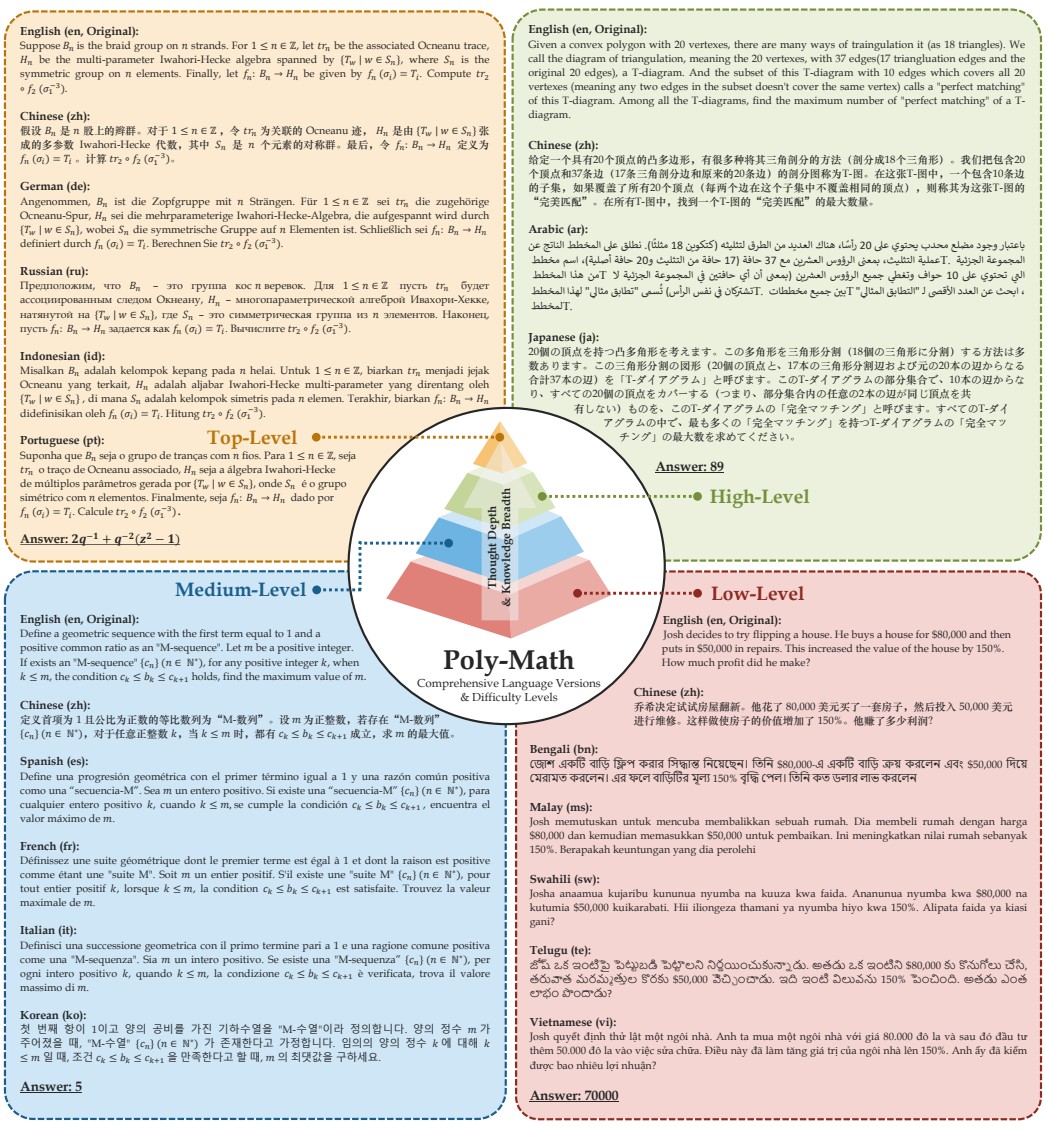

Figure 1: Illustration and question-answer examples of our **PolyMath** benchmark: We partition difficulty in the mathematical field into 4 levels from a macro perspective. Each level consists of 125 problems, with each problem available in 18 parallel language versions. Besides English and Chinese, examples in other languages are randomly displayed under one of the levels in the figure.

Mathematics serves as a fundamental field for evaluating LLM reasoning intelligence. In recent years, mathematical reasoning benchmarks have closely evolved alongside LLMs, covering almost every domain from basic math word problems [22, 9] to complex multidisciplinary calculations and proofs [17, 6], and even to Olympiad competitions [16, 11] and frontier mathematical challenges that approach the limits of human intelligence [13, 21].

However, the connection between "language" and "reasoning" is not explored sufficiently. Popular multilingual mathematical datasets, such as MGSM [23] and XSVAMP [4], are too simple to effectively assess the reasoning capabilities of advanced reasoning LLMs. As a result, multilingual reasoning benchmarks have lagged behind recent advances in LLM performance, with most challenging datasets still limited to English [12]. Recent efforts like MT-AIME [24] offer modest difficulty improvements but include only very limited data per language. Moreover, all translations are generated by LLMs without human validation, raising concerns about data quality. Thus, building a high-quality and challenging multilingual mathematical reasoning benchmark is essential for advancing the study of multilingual reasoning in the reasoning LLM era.

To bridge this gap, we build **PolyMath**, a multilingual benchmark organized by comprehensive difficulty levels, spanning from K-12 to Olympiad and advanced frontier mathematics. **Figure 1** depicts the structure of PolyMath and includes representative examples from it, which is designed with the following key principles:

- **Broad Difficulty Range**: PolyMath is meticulously structured into four difficulty levels based on two macro dimensions: Thought Depth and Knowledge Breadth. Each level contains 125 problems per language, spanning a broad range of mathematical topics and domains. This design helps clearly differentiate the reasoning abilities of LLMs.

- **Language Diversity**: Each problem in PolyMath is available in 18 parallel language versions, encompassing over 75% of the world's native speakers and major language families, ensuring diversity across both high-resource and low-resource languages.

- **High-Quality Translations**: Each translation in PolyMath is carefully calibrated by language experts, who are either linguistics graduate students from top universities or native speakers. This ensures precise terms and logical clarity while avoiding the direct use of LLM-generated output.

We conduct extensive experiments (Section 3.2) on advanced non-reasoning and reasoning LLMs, finding that even Qwen3-235B-A22B-Thinking and Gemini-2.5-pro achieve benchmark scores of only 54.6 and 52.2, with accuracy about 40% at the highest level. Crucially, PolyMath effectively reveals language gaps in reasoning performance across different difficulty levels, with differences of up to 10 points even at low-accuracy settings. Beyond performance, we further explore the *consistency between input and output languages* (Section 3.3), the *impact of controlling output language on performance* (Section 3.4), and the *thinking length variations across languages* (Section 3.5). These conclusions offer valuable insights into the slow-thinking pattern in multilingual contexts. We hope PolyMath serves as a strong benchmark to advance the research of multilingual reasoning.

## 2 Construction of PolyMath Benchmark

This section details the construction process of our PolyMath benchmark. Section 2.1 defines the easy-to-hard difficulty levels for mathematical reasoning from a macro perspective, creates a comprehensive gradient for the benchmark. Section 2.2 presents data collection, ensuring each level includes representative mathematical problems. Section 2.3 describes how we translate the original English problems into multiple language versions. Finally, Section 2.4 provides statistics of PolyMath, and Section 2.5 introduces a difficulty-weighted accuracy metric tailored to PolyMath.

### 2.1 Difficulty Level Partition

Existing English mathematical benchmarks typically classify levels based on either "learning stages" (*e.g.*, high school, undergraduate, graduate) [16] or "problem sources" (*e.g.*, exams and competitions) [11], mainly to indicate problem difficulty. However, our goal is to build a benchmark that spans a wide range of the mathematical field, which poses challenges for current partitioning methods: (1) Different benchmarks use *inconsistent criteria*, such as learning stages or problem sources, making integration difficult; (2) From a broader view of mathematics, existing categories are often *overly fine-grained*. For example, distinguishing between "high school" and "undergraduate" math may not correspond to meaningful difficulty differences. For LLMs with broad knowledge, such categories may offer similar difficulty levels when thought depth remains comparable.

Therefore, we define and partition difficulty levels in the mathematical field using two key dimensions: **Thought Depth** and **Knowledge Breadth**. Thought Depth corresponds to human IQ, while Knowledge Breadth represents the extent of a person's mathematical knowledge. Specific partition standards and explanations are shown in Table 1. The existing benchmark partition dimensions about "learning stage" and "problem source" help categorize data within each level.

### 2.2 Data Collection

PolyMath consists of 500 high-quality mathematical reasoning problems, with 125 problems at each level. All original problems are presented in English and later translated into other languages (see Section 2.3). Our data collection process integrates two methods: incorporating existing publicly

Table 1: Difficulty level partition in our PolyMath and corresponding explanations.

| Difficulty Level | Dimension | | Problem Type |
|---|---|---|---|
| | **Thought Depth** | **Knowledge Breadth** | |
| **Low** | ★ | ★ | • **K-12 Mathematics:** Basic algebra, geometry, and probability & statistics, primarily presented as Math Word Problems (MWP). |
| **Medium** | ★★ | ★★ | • **Exercises and Exams (High School & University):** Post-class exercises from various math branches and authoritative entrance exams.
• **Competitions (Low Difficulty):** Publicly accessible competitions that are slightly more challenging than standard in-class exam problems. |
| **High** | ★★★ | ★★ | • **Competitions (Mid-to-High Difficulty):** Problems that require critical thinking but do not demand deep theoretical knowledge. In comparison to the competitions at the medium level, the participants in these contests have already undergone preliminary selection. |
| **Top** | ★★★★ $\to \infty$ | ★★★ $\to \infty$ | • **Competitions (Top Olympiad):** The highest-tier international/national/regional mathematics Olympiads, representing the upper limits of human IQ.
• **Frontier Mathematics:** Advanced mathematical disciplines and emerging research areas, approaching the limits of human mathematical systems. |

available benchmarks and scraping official repositories from the Internet (main). The problems are collected according to the four-level partition in Table 1:

- Low-level: Due to the relatively uniform problem format in K-12 mathematics, we directly source all 125 samples from MGSM [23] with 10 language versions. Additionally, we acquire translations in 4 more languages from P-MMEval [37].

- Medium-level: For exam problems, we include post-class exercises from College Math used in university courses, as well as final questions from China's Gaokao and post-graduate entrance exams. For low-difficulty competition problems, we focus on the USA's AMC and China's CNMO (first round, provincial selection) and the CMC, which are widely accessible to a general participant. All problems are collected from official websites and parsed from PDFs.

- High-level: For mid-to-high-difficulty competition problems, we focus on competitions that have an initial selection process and an entry threshold but are not the top international or national contests, thus maintaining a broad selection nature, such as the USA's AIME and China's CNMO.

- Top-level: For the top Olympiads competition, we select 100 problems from IMO/IMO-shortlist and various national/regional Olympiads (*e.g.*, CMO, USAMO, Putnam). All competition problems are sourced from their official websites or AoPS Wiki. For frontier mathematics, we select 25 problems from the HLE dataset [21], which represent challenging problems as perceived by the world's top mathematicians.

The specific sources and problem numbers are detailed in Appendix D.1. Unlike prior work that relies on LLMs to assign difficulty levels [11], our level tagging is conducted entirely by doctoral students with strong backgrounds in mathematics and competition experience. Each label is further reviewed by two additional mathematical experts. This process avoids the uncertainties of LLM-based tagging and minimizes human bias, ensuring both accuracy and professionalism in level assignment. The four-level classification criteria have also been reviewed and endorsed by experts.

## 2.3 Multilingual Translation Annotation

After collecting original English problems, we translate them into multiple languages. We select **18 languages** (including English), covering several major language families and 75% of the world population. Detailed language information is shown in Table 2.

Mathematical translation is far more challenging than general translation: (1) Translators must be fluent in the target languages and have a basic grasp of mathematics. They need to accurately interpret mathematical terms and understand the problem's logic. Such specialized annotators are rare. (2) The process is also highly time-consuming, as it requires precise handling of domain-specific terms. A single mistake in a key term can distort the entire problem, rendering the sample completely invalid.

Considering these annotation challenges, we leverage LLM to assist in the translation process and reduce costs. Our annotation pipeline consists of three stages: (1) **LLM Pre-Translation:** We first prompt GPT-4o to generate preliminary translations of the original English problem $Q_1$ into various target languages $\{Q_i\}_{i=2}^{18}$. (2) **Terms Extraction:** Next, we recruit mathematical experts to extract key terms from $Q_1$ to form a term list $\mathcal{T}$, representing what must be accurately translated in the problem; otherwise, the sample correctness will be affected. The annotator extracts about 30 terms per hour. (3) **Translation Calibration:** Finally, we recruit language experts with basic mathematical knowledge to modify GPT-4o's translations $\{Q_i\}_{i=2}^{18}$ into the updated translation versions $\{Q'_i\}_{i=2}^{18}$, with a primary focus on verifying the translation precision of each term $T \in \mathcal{T}$. Meanwhile, they must ensure that formulas are losslessly transferred across different language versions, paying special attention to cases

Table 2: Detailed information of all 18 languages supported by our PolyMath. Statistical Data are from `https://www.ethnologue.com/`.

| Code | Full Name | Language Family | Native Speakers (M) |
|------|-----------|-----------------|---------------------|
| en | English | Indo-European | 1,500 |
| zh | Chinese | Sino-Tibetan | 1,400 |
| es | Spanish | Indo-European | 595 |
| ar | Arabic | Afro-Asiatic | 400 |
| fr | French | Indo-European | 300 |
| bn | Bengali | Indo-European | 300 |
| pt | Portuguese | Indo-European | 270 |
| ru | Russian | Indo-European | 260 |
| id | Indonesian | Austronesian | 200 |
| de | German | Indo-European | 135 |
| ja | Japanese | Japonic | 130 |
| sw | Swahili | Niger-Congo | 100 |
| vi | Vietnamese | Austroasiatic | 86 |
| it | Italian | Indo-European | 85 |
| te | Telugu | Dravidian | 81 |
| ko | Korean | Koreanic | 80 |
| th | Thai | Kra-Dai | 80 |
| ms | Malay | Austronesian | 77 |

Total: 6,079 (∼75% of total world population)

where notation differs but the compiled result remains the same. Annotators calibrate about 8 samples per hour, with $Q_1$ and $\{Q'_i\}_{i=2}^{18}$ constituting the final set of 18 parallel language versions of the problems. Detailed annotation process and annotator backgrounds are shown in Appendix C.

Table 3: The ratio of samples with content errors (content disagreement) or fluency issues (fluency disagreement) identified by annotators in GPT-4o's pre-translation for each language and level.

| | | zh | ar | bn | de | es | fr | id | it | ja | ko | ms | pt | ru | sw | te | th | vi |
|---|---|----|----|----|----|----|----|----|----|----|----|----|----|----|----|----|----|----|
| Content Disagreement Rate (%) | Low-level | - | - | - | - | - | - | 0.0 | 0.0 | - | - | 0.0 | - | - | - | - | - | - |
| | Medium-level | 4.8 | 8.8 | 9.6 | 8.0 | 9.6 | 0.0 | 8.0 | 1.6 | 12.8 | 4.0 | 4.8 | 9.6 | 9.6 | 15.2 | 12.8 | 15.2 | 1.6 |
| | High-level | 4.8 | 2.4 | 14.4 | 7.2 | 7.2 | 1.6 | 7.2 | 2.4 | 15.2 | 4.0 | 4.0 | 16.8 | 16.8 | 20.0 | 13.6 | 16.8 | 4.8 |
| | Top-level | 6.4 | 4.0 | 14.4 | 9.6 | 9.6 | 0.8 | 8.0 | 2.4 | 12.8 | 7.2 | 8.8 | 8.8 | 13.6 | 20.0 | 18.4 | 20.0 | 8.0 |
| Fluency Disagreement Rate (%) | Low-level | - | - | - | - | - | - | 4.8 | 3.2 | - | - | 5.6 | - | - | - | - | - | - |
| | Medium-level | 7.2 | 5.6 | 16.8 | 6.4 | 4.8 | 3.2 | 0.0 | 4.0 | 5.6 | 4.8 | 6.4 | 1.6 | 4.8 | 10.4 | 9.6 | 6.4 | 8.0 |
| | High-level | 8.8 | 8.0 | 15.2 | 6.4 | 8.8 | 2.4 | 2.4 | 4.8 | 9.6 | 2.4 | 9.6 | 2.4 | 8.0 | 14.4 | 14.4 | 7.2 | 8.0 |
| | Top-level | 13.6 | 9.6 | 21.6 | 14.4 | 4.8 | 2.4 | 1.6 | 4.0 | 14.4 | 3.2 | 6.4 | 1.6 | 4.8 | 15.2 | 13.6 | 4.0 | 8.0 |

**Why Not Rely on LLM Translations?** During the calibration process, we simultaneously count the number of samples where annotators disagree with the translation results of GPT-4o, mainly on two key aspects: (1) **Content Disagreement**: The number of samples where annotators identify errors in term translation, errors in formula migration, or other content that affects the problem meanings.; (2) **Fluency Disagreement**: The number of samples where annotators find GPT-4o's translation introduces unclear nested conditions or logic, making comprehension more difficult. Table 3 presents the statistical results, showing that the disagreement number is nonzero across almost all languages. Notably, the content disagreement rate directly reflects the proportion of unusable samples — any nonzero value indicates the presence of erroneous samples. This underscores that relying entirely on LLM-based translation will introduce significant noise into the benchmark.

## 2.4 Benchmark Statistics

**MetaData.** Table 4 presents the general metadata of our PolyMath, with the full metadata of each language shown in Appendix D.2. (1) **Problem** length generally increases with higher levels, but the differences across languages are minimal, indicating that translation has little impact on the length of the same problem. (2) **Answers** in our PolyMath are presented in diverse forms, including Numeric, Expression, Equation, Interval, Set, Tuple. As the level increases, the diversity of answers grows, leading to greater variance in answer length. (3) **Natural Language Coverage (NLC)** refers to the proportion of text remaining after excluding language-independent formula blocks from the problem. A higher NLC suggests that LLMs may be more influenced by language when understanding the problem. We find that as the level increases, NLC also rises due to the increasing number of formulas involved. However, the overall value remains around 50%, staying within a reasonable range.

**Domain Diversity.** We conduct separate domain statistics for each level, demonstrating that our problem domains remain diverse at every level. Detailed statistical results can be found in Appendix D.3.

**Semantic Visualization.** We use the T-SNE projection to visualize the text embeddings of the English problems at each level. The problems are encoded using `gte-Qwen2-7B-instruct` [36], and the visualization is shown in Figure 2. We find that low-level problems differ significantly from the other levels, while the remaining three levels, though distinguishable, exhibit some degree of overlap. This occurs because the difference in "thought depth" among the last three levels is greater than the difference in "knowledge breadth", which is difficult to fully capture in embeddings [29, 30]. Therefore, problems that are close in embedding space can differ significantly in difficulty, further underscoring the importance of human intervention in data collection and difficulty labeling.

## 2.5 Benchmark Score: Difficulty-Weighted Accuracy

Our PolyMath is structured into four levels. Using standard accuracy as the evaluation metric would equate solving a low-level problem with solving a top-level one, which is inherently unfair as the efforts required to solve them successfully differ significantly.. To address this, we introduce the **Difficulty-Weighted Accuracy (DW-ACC)** as our benchmark metric. This metric assigns level-specific weights $w_1, w_2, w_3, w_4$ to each problem from the low-, medium-, high-, and top-level, respectively. **The weights double at each ascending level: By default, we set $w_1 = 1$, leading to** $w_2 = 2, w_3 = 4, w_4 = 8$. This means that solving eight low-level problems is equivalent to solving a single top-level problem in terms of contribution to the final score.

DW-ACC provides a more reliable performance metric by downweighting success on easier problems and giving more importance to correct answers at higher levels. Given the accuracy at each level $\{a_i\}_{i=1}^4$, DW-ACC is defined as:

$$\text{DW-ACC} = \frac{\sum_{i=1}^4 w_i a_i}{\sum_{i=1}^4 w_i} = \sum_{i=1}^4 \left( \frac{2^{i-1}}{15} a_i \right) \tag{1}$$

## 3 Experiments

### 3.1 Setup

**Baselines.** We categorize the existing LLMs into two types for evaluation: **Non-Reasoning LLMs** and **Reasoning LLMs**. For each category, we select 8 advanced LLMs. The non-reasoning LLMs are: GPT-4.5-Preview, ChatGPT-4o-latest, Qwen-2.5-Max, Deepseek-v3, Claude-3.7-sonnet, Qwen-2.5-72B-Instruct, Qwen-2.5-Math-72B-Instruct, and Llama-3.3-70B-Instruct. The reasoning LLMs are: OpenAI-o3-mini-medium, OpenAI-o1-mini, Gemini-2.5-pro, Gemini-2.0-flash-thinking, Claude-3.7-sonnet-thinking, Qwen-3-235B-A22B-Thinking, Qwen-QwQ-32B, and Deepseek-R1-671B. Snapshot versions, citations, and website links of these models are provided in Appendix E.1.

Table 4: **Metadata of PolyMath**. "*" indicates the average across all languages, with the standard deviation shown in the bottom-right cell. Lengths are computed after tokenization using the Gemma3 tokenizer [26].

| Statistical Item | Number |
|---|---|
| Language Type | 18 |
| Difficulty Level | 4 |
| Problem Each Level | 125 |
| Total Data | 125*4*18=9000 |
| Average Problem Length* | |
| • Low-level | $72.2_{8.9}$ |
| • Medium-level | $101.2_{6.8}$ |
| • High-level | $126.4_{10.7}$ |
| • Top-level | $133.7_{11.7}$ |
| Average Answer Length | |
| • Low-level | 2.3 |
| • Medium-level | 9.2 |
| • High-level | 6.2 |
| • Top-level | 9.4 |
| Average Natural Language Coverage* | |
| • Low-level | $97.4\%_{1.4\%}$ |
| • Medium-level | $43.5\%_{3.7\%}$ |
| • High-level | $55.7\%_{3.2\%}$ |
| • Top-level | $54.7\%_{3.8\%}$ |

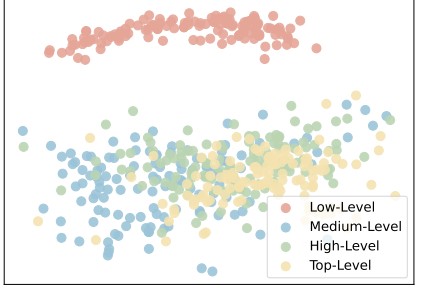

Figure 2: T-SNE projection for problem (English) embeddings at each level.

Table 5: The accuracy for non-reasoning and reasoning language models across four levels and 18 languages in PolyMath. Models with "†" are closed-source. **Bold** indicates the best performance overall. Blue and red shadings show the best/poorest-performing language for each model.

(a) **PolyMath-Top**

| | avg. | en | zh | ar | bn | de | es | fr | id | it | ja | ko | ms | pt | ru | sw | te | th | vi | std. | range |
|---|---|---|---|---|---|---|---|---|---|---|---|---|---|---|---|---|---|---|---|---|---|
| **Non-Reasoning LLMs** | | | | | | | | | | | | | | | | | | | | | |
| Llama-3.3-70B-Instruct | 5.7 | 12.0 | 2.4 | 8.0 | 1.6 | 6.4 | 5.6 | 6.4 | 5.6 | 6.4 | 3.2 | 2.4 | 6.4 | 10.4 | 6.4 | 5.6 | 2.4 | 6.4 | 4.8 | 2.7 | 10.4 |
| Qwen-2.5-72B-Instruct | 7.6 | 8.8 | 10.4 | 5.6 | 9.6 | 6.4 | 11.2 | 4.0 | 12.0 | 9.6 | 5.6 | 7.2 | 6.4 | 8.8 | 5.6 | 4.8 | 4.8 | 7.2 | 9.6 | 2.3 | 8.0 |
| Qwen-2.5-Math-72B-Instruct | 10.7 | 10.4 | 11.2 | 12.8 | 11.2 | 8.8 | 10.4 | 15.2 | 8.8 | 12.8 | 9.6 | 9.6 | 11.2 | 10.4 | 8.0 | 10.4 | 7.2 | 12.8 | 11.2 | 1.9 | 8.0 |
| Deepseek-v3 | 9.6 | 9.6 | 9.6 | 12.0 | 6.4 | 8.8 | 11.2 | 12.8 | 11.2 | 12.8 | 11.2 | 7.2 | 11.2 | 7.2 | 8.8 | 10.4 | 4.8 | 8.8 | 8.8 | 2.2 | 8.0 |
| Qwen-2.5-Max † | 9.3 | 12.8 | 4.8 | 11.2 | 7.2 | 7.2 | 11.2 | 11.2 | 12.0 | 10.4 | 7.2 | 8.8 | 9.6 | 11.2 | 9.6 | 8.0 | 7.2 | 7.2 | 8.0 | 2.1 | 8.0 |
| Claude-3.7-sonnet † | 11.0 | 11.2 | 11.2 | 12.8 | 8.0 | 11.2 | 12.8 | 14.4 | 14.4 | 11.2 | 8.8 | 7.2 | 12.8 | 12.8 | 9.6 | 10.4 | 8.0 | 8.8 | 12.8 | 2.2 | 7.2 |
| ChatGPT-4o-latest † | 13.7 | 18.4 | 16.8 | 11.2 | 13.6 | 13.6 | 15.2 | 15.2 | 16.0 | 18.4 | 8.0 | 12.0 | 12.8 | 13.6 | 12.8 | 13.6 | 12.8 | 12.8 | 12.8 | 2.6 | 10.4 |
| GPT-4.5-preview † | 14.7 | 18.4 | 17.6 | 15.2 | 14.4 | 14.4 | 13.6 | 17.6 | 16.0 | 16.0 | 13.6 | 12.0 | 14.4 | 16.0 | 16.0 | 12.0 | 11.2 | 11.2 | 14.4 | 2.1 | 7.2 |
| **Reasoning LLMs** | | | | | | | | | | | | | | | | | | | | | |
| Deepseek-R1-671B | 33.7 | 35.2 | 32.0 | 32.5 | 36.0 | 37.6 | 37.6 | 32.8 | 32.5 | 36.0 | 32.0 | 27.2 | 32.8 | 36.8 | 37.6 | 29.7 | 29.6 | 30.4 | 37.6 | 3.3 | 10.4 |
| Qwen-QwQ-32B | 30.6 | 36.8 | 31.2 | 25.6 | 30.4 | 34.4 | 37.6 | 34.4 | 30.4 | 34.4 | 25.6 | 24.0 | 33.6 | 33.6 | 28.8 | 28.0 | 24.0 | 26.4 | 32.0 | 4.3 | 13.6 |
| Qwen3-235B-A22B-Thinking | 40.3 | 40.0 | 39.2 | 39.2 | 38.4 | 41.6 | 44.0 | 44.0 | 39.5 | 40.8 | 38.4 | 41.6 | 40.8 | 45.6 | 38.4 | 35.2 | 39.2 | 38.4 | 40.8 | 2.7 | 10.4 |
| Claude-3.7-sonnet-thinking † | 20.8 | 25.6 | 17.6 | 21.6 | 20.0 | 20.0 | 19.2 | 21.6 | 21.6 | 22.4 | 22.4 | 18.4 | 20.8 | 19.2 | 21.6 | 21.6 | 16.0 | 25.6 | 19.2 | 2.4 | 9.6 |
| Gemini-2.0-flash-thinking † | 22.3 | 26.4 | 22.4 | 20.8 | 20.8 | 26.4 | 25.6 | 22.4 | 23.2 | 26.4 | 22.0 | 19.2 | 22.4 | 21.6 | 21.6 | 21.6 | 17.6 | 22.4 | 18.4 | 2.5 | 8.8 |
| Gemini-2.5-pro † | 38.0 | 39.2 | 36.8 | 39.2 | 36.0 | 38.4 | 40.0 | 37.6 | 37.9 | 37.6 | 43.2 | 35.2 | 36.8 | 36.8 | 38.4 | 36.8 | 34.4 | 41.6 | 38.4 | 2.4 | 8.8 |
| OpenAI-o1-mini † | 22.5 | 20.8 | 23.2 | 25.6 | 22.4 | 24.8 | 23.2 | 23.2 | 19.2 | 26.4 | 25.6 | 20.8 | 21.6 | 21.6 | 24.0 | 23.2 | 17.6 | 20.0 | 21.6 | 2.3 | 8.8 |
| OpenAI-o3-mini-medium † | 23.0 | 25.6 | 25.6 | 27.2 | 18.4 | 23.2 | 20.8 | 20.8 | 20.8 | 26.4 | 24.8 | 22.4 | 22.4 | 20.0 | 17.6 | 28.8 | 20.8 | 22.4 | 23.2 | 3.4 | 11.2 |

(b) **PolyMath-High**

| | avg. | en | zh | ar | bn | de | es | fr | id | it | ja | ko | ms | pt | ru | sw | te | th | vi | std. | range |
|---|---|---|---|---|---|---|---|---|---|---|---|---|---|---|---|---|---|---|---|---|---|
| **Non-Reasoning LLMs** | | | | | | | | | | | | | | | | | | | | | |
| Llama-3.3-70B-Instruct | 7.3 | 14.4 | 6.4 | 5.6 | 1.6 | 7.2 | 4.8 | 5.6 | 4.8 | 11.2 | 5.6 | 4.8 | 11.2 | 10.4 | 10.4 | 6.4 | 4.8 | 5.6 | 10.4 | 3.2 | 12.8 |
| Qwen-2.5-72B-Instruct | 11.6 | 14.4 | 12.0 | 11.2 | 10.4 | 12.0 | 12.0 | 9.6 | 11.2 | 12.0 | 11.2 | 5.6 | 12.0 | 13.6 | 13.6 | 5.6 | 12.0 | 14.4 | 11.2 | 2.0 | 8.8 |
| Qwen-2.5-Math-72B-Instruct | 16.8 | 16.0 | 18.4 | 17.6 | 16.8 | 22.4 | 19.2 | 16.8 | 13.6 | 13.6 | 16.8 | 17.6 | 16.8 | 19.2 | 16.0 | 15.2 | 16.0 | 12.8 | 16.8 | 2.2 | 9.6 |
| Deepseek-v3 | 16.5 | 16.8 | 17.6 | 16.8 | 15.2 | 17.6 | 16.8 | 16.8 | 12.8 | 16.0 | 16.0 | 17.6 | 17.6 | 20.0 | 19.2 | 12.8 | 12.8 | 14.4 | 16.0 | 2.0 | 7.2 |
| Qwen-2.5-Max † | 14.6 | 12.0 | 17.6 | 16.8 | 10.4 | 13.6 | 16.8 | 16.0 | 16.0 | 14.4 | 10.4 | 14.4 | 16.8 | 16.0 | 16.0 | 14.4 | 13.6 | 12.8 | 15.2 | 2.1 | 7.2 |
| Claude-3.7-sonnet † | 15.0 | 21.6 | 17.6 | 14.4 | 13.6 | 17.6 | 15.2 | 12.8 | 16.8 | 12.0 | 16.0 | 13.6 | 15.2 | 14.4 | 12.8 | 16.0 | 15.2 | 13.6 | 14.4 | 2.3 | 9.6 |
| ChatGPT-4o-latest † | 20.4 | 22.4 | 21.6 | 20.0 | 16.8 | 23.2 | 20.0 | 24.8 | 16.0 | 20.0 | 22.4 | 20.0 | 23.2 | 20.8 | 21.6 | 17.6 | 17.6 | 17.6 | 20.8 | 2.4 | 8.8 |
| GPT-4.5-preview † | 27.4 | 34.4 | 25.6 | 24.8 | 24.0 | 24.8 | 29.6 | 27.2 | 27.2 | 27.2 | 28.8 | 27.2 | 25.6 | 29.6 | 27.2 | 31.4 | 25.6 | 24.0 | 29.6 | 2.7 | 10.4 |
| **Reasoning LLMs** | | | | | | | | | | | | | | | | | | | | | |
| Deepseek-R1-671B | 50.7 | 48.8 | 46.4 | 50.4 | 46.4 | 52.8 | 55.2 | 52.8 | 52.0 | 56.8 | 51.2 | 46.4 | 51.2 | 52.0 | 51.2 | 51.2 | 44.8 | 49.6 | 52.8 | 3.1 | 12.0 |
| Qwen-QwQ-32B | 53.9 | 62.4 | 55.2 | 47.2 | 50.4 | 63.2 | 60.0 | 58.4 | 56.0 | 56.8 | 44.8 | 47.2 | 57.6 | 59.2 | 55.2 | 45.6 | 43.2 | 47.2 | 60.0 | 6.4 | 20.0 |
| Qwen3-235B-A22B-Thinking | 63.3 | 66.4 | 62.9 | 62.4 | 62.4 | 63.2 | 64.8 | 66.4 | 60.8 | 70.4 | 61.6 | 64.8 | 59.2 | 64.8 | 60.0 | 60.0 | 60.0 | 64.0 | 65.6 | 3.1 | 11.2 |
| Claude-3.7-sonnet-thinking † | 36.7 | 36.0 | 38.4 | 36.8 | 38.4 | 35.2 | 29.6 | 32.0 | 36.8 | 34.4 | 37.6 | 39.2 | 37.6 | 40.8 | 40.0 | 38.4 | 37.6 | 33.6 | 34.4 | 2.8 | 11.2 |
| Gemini-2.0-flash-thinking † | 42.9 | 43.2 | 43.2 | 42.4 | 44.0 | 40.8 | 42.4 | 48.0 | 41.6 | 44.0 | 36.8 | 40.8 | 44.0 | 46.4 | 47.2 | 41.6 | 36.8 | 46.4 | 43.2 | 3.0 | 11.2 |
| Gemini-2.5-pro † | 62.2 | 66.4 | 66.4 | 62.4 | 62.4 | 62.4 | 65.6 | 64.8 | 63.2 | 62.8 | 64.8 | 61.6 | 60.8 | 62.4 | 62.4 | 56.8 | 56.0 | 60.0 | 60.0 | 3.8 | 18.4 |
| OpenAI-o1-mini † | 40.5 | 46.4 | 44.8 | 37.6 | 37.6 | 36.0 | 43.2 | 40.0 | 40.8 | 40.8 | 41.6 | 43.2 | 38.4 | 40.8 | 38.4 | 36.0 | 40.0 | 42.4 | 41.6 | 2.8 | 10.4 |
| OpenAI-o3-mini-medium † | 50.0 | 54.4 | 52.8 | 51.2 | 52.8 | 53.6 | 51.2 | 50.4 | 56.0 | 45.6 | 52.0 | 50.4 | 50.4 | 50.4 | 39.2 | 51.2 | 41.6 | 44.8 | 52.8 | 4.3 | 16.8 |

(c) **PolyMath-Medium**

| | avg. | en | zh | ar | bn | de | es | fr | id | it | ja | ko | ms | pt | ru | sw | te | th | vi | std. | range |
|---|---|---|---|---|---|---|---|---|---|---|---|---|---|---|---|---|---|---|---|---|---|
| **Non-Reasoning LLMs** | | | | | | | | | | | | | | | | | | | | | |
| Llama-3.3-70B-Instruct | 16.8 | 32.0 | 18.4 | 16.0 | 12.0 | 13.6 | 16.8 | 14.4 | 20.0 | 20.8 | 12.0 | 18.4 | 18.4 | 23.2 | 12.0 | 12.8 | 12.8 | 11.2 | 16.8 | 5.0 | 20.8 |
| Qwen-2.5-72B-Instruct | 29.6 | 36.8 | 25.6 | 27.2 | 28.8 | 29.6 | 30.4 | 27.2 | 30.4 | 32.8 | 30.4 | 31.2 | 26.4 | 29.6 | 36.8 | 24.8 | 22.4 | 28.8 | 32.8 | 3.7 | 14.4 |
| Qwen-2.5-Math-72B-Instruct | 37.4 | 36.8 | 39.2 | 37.6 | 35.2 | 40.8 | 36.8 | 37.6 | 36.8 | 40.0 | 35.2 | 36.8 | 39.2 | 39.2 | 37.6 | 34.4 | 36.8 | 39.2 | 38.4 | 1.8 | 6.4 |
| Deepseek-v3 | 36.1 | 40.8 | 39.2 | 34.4 | 31.2 | 36.0 | 40.0 | 40.0 | 33.6 | 38.4 | 35.2 | 38.4 | 33.6 | 36.8 | 37.6 | 32.0 | 28.8 | 34.4 | 40.0 | 3.4 | 12.0 |
| Qwen-2.5-Max † | 33.0 | 41.6 | 31.2 | 32.0 | 25.6 | 37.6 | 34.4 | 28.0 | 33.6 | 35.2 | 33.6 | 35.2 | 31.2 | 33.6 | 28.8 | 27.2 | 35.2 | 36.0 | 38.0 | 3.8 | 16.0 |
| Claude-3.7-sonnet † | 28.3 | 30.4 | 33.6 | 29.6 | 24.0 | 30.4 | 26.4 | 28.8 | 28.8 | 28.8 | 26.4 | 28.0 | 26.4 | 26.4 | 25.6 | 29.6 | 29.6 | 29.6 | 27.2 | 2.2 | 9.6 |
| ChatGPT-4o-latest † | 40.8 | 42.4 | 46.4 | 41.6 | 38.4 | 41.6 | 42.4 | 46.4 | 38.4 | 41.6 | 44.0 | 39.2 | 38.4 | 45.6 | 32.8 | 41.6 | 36.8 | 36.8 | 40.8 | 3.7 | 13.6 |
| GPT-4.5-preview † | 42.9 | 42.4 | 46.4 | 41.6 | 40.0 | 40.4 | 42.4 | 41.6 | 42.4 | 48.0 | 48.0 | 45.6 | 41.6 | 43.5 | 46.3 | 42.4 | 37.6 | 36.8 | 44.8 | 3.1 | 11.2 |
| **Reasoning LLMs** | | | | | | | | | | | | | | | | | | | | | |
| Deepseek-R1-671B | 70.4 | 72.8 | 69.6 | 68.8 | 63.2 | 70.4 | 72.8 | 73.6 | 71.2 | 71.2 | 67.2 | 68.8 | 69.6 | 73.6 | 70.4 | 72.0 | 72.0 | 68.0 | 72.8 | 2.6 | 10.4 |
| Qwen-QwQ-32B | 68.9 | 73.6 | 73.6 | 68.8 | 66.4 | 72.0 | 73.6 | 74.4 | 75.2 | 72.0 | 59.2 | 64.0 | 69.6 | 74.4 | 67.2 | 57.6 | 60.0 | 64.8 | 74.4 | 6.1 | 17.6 |
| Qwen3-235B-A22B-Thinking | 75.8 | 76.8 | 77.6 | 74.4 | 73.6 | 78.4 | 76.0 | 76.8 | 74.4 | 78.4 | 73.6 | 76.0 | 76.0 | 77.6 | 76.0 | 74.4 | 74.4 | 72.0 | 76.8 | 1.8 | 6.4 |
| Claude-3.7-sonnet-thinking † | 48.8 | 44.8 | 52.0 | 52.8 | 50.4 | 48.8 | 44.0 | 52.0 | 48.0 | 46.4 | 53.6 | 52.0 | 51.2 | 41.6 | 52.8 | 48.0 | 48.0 | 47.2 | 45.6 | 3.4 | 12.0 |
| Gemini-2.0-flash-thinking † | 59.4 | 62.4 | 59.2 | 62.4 | 54.4 | 59.2 | 64.0 | 60.8 | 60.0 | 61.6 | 53.6 | 53.6 | 60.0 | 63.2 | 60.8 | 60.0 | 59.2 | 56.0 | 59.2 | 3.0 | 10.4 |
| Gemini-2.5-pro † | 72.0 | 73.6 | 73.6 | 68.8 | 69.6 | 75.2 | 76.0 | 75.2 | 68.8 | 72.8 | 72.8 | 68.8 | 69.6 | 76.0 | 70.4 | 68.0 | 72.8 | 68.8 | 76.0 | 2.9 | 8.0 |
| OpenAI-o1-mini † | 58.2 | 58.4 | 62.4 | 60.8 | 56.0 | 59.2 | 59.2 | 57.6 | 61.6 | 56.8 | 60.8 | 61.6 | 55.2 | 59.2 | 57.6 | 56.8 | 53.6 | 59.2 | 52.0 | 2.8 | 10.4 |
| OpenAI-o3-mini-medium † | 52.8 | 55.2 | 48.8 | 49.6 | 48.0 | 60.0 | 51.2 | 55.2 | 56.0 | 52.0 | 55.2 | 54.4 | 56.8 | 47.2 | 43.2 | 53.6 | 50.4 | 56.0 | 57.6 | 4.6 | 16.8 |

(d) **PolyMath-Low**

| | avg. | en | zh | ar | bn | de | es | fr | id | it | ja | ko | ms | pt | ru | sw | te | th | vi | std. | range |
|---|---|---|---|---|---|---|---|---|---|---|---|---|---|---|---|---|---|---|---|---|---|
| **Non-Reasoning LLMs** | | | | | | | | | | | | | | | | | | | | | |
| Llama-3.3-70B-Instruct | 64.2 | 96.8 | 66.4 | 26.4 | 53.6 | 60.0 | 70.4 | 62.4 | 83.2 | 78.4 | 32.8 | 51.2 | 84.0 | 68.8 | 64.8 | 60.8 | 60.8 | 68.0 | 67.2 | 16.4 | 70.4 |
| Qwen-2.5-72B-Instruct | 87.5 | 96.0 | 89.6 | 90.4 | 88.8 | 85.6 | 92.8 | 86.4 | 94.4 | 95.2 | 84.8 | 88.0 | 92.8 | 92.0 | 92.0 | 58.4 | 68.0 | 88.0 | 92.0 | 9.3 | 37.6 |
| Qwen-2.5-Math-72B-Instruct | 87.3 | 96.8 | 88.8 | 89.6 | 86.4 | 88.8 | 92.8 | 89.6 | 92.8 | 93.6 | 88.0 | 88.8 | 92.0 | 92.0 | 92.8 | 48.0 | 78.4 | 88.0 | 84.0 | 10.3 | 48.8 |
| Deepseek-v3 | 91.4 | 97.6 | 90.4 | 91.2 | 89.6 | 88.8 | 95.2 | 89.6 | 95.2 | 96.0 | 87.2 | 88.8 | 92.8 | 94.4 | 92.0 | 86.4 | 88.0 | 91.2 | 90.4 | 3.1 | 11.2 |
| Qwen-2.5-Max † | 91.3 | 97.6 | 89.6 | 91.2 | 91.2 | 88.0 | 94.4 | 89.6 | 96.0 | 94.4 | 86.4 | 90.4 | 95.2 | 92.8 | 92.8 | 80.8 | 87.2 | 92.8 | 92.8 | 3.9 | 16.8 |
| Claude-3.7-sonnet † | 90.9 | 97.6 | 90.4 | 95.2 | 91.2 | 87.2 | 93.6 | 88.8 | 93.6 | 95.2 | 86.4 | 87.2 | 92.8 | 88.0 | 93.6 | 84.8 | 84.8 | 92.0 | 89.6 | 3.4 | 12.8 |
| ChatGPT-4o-latest † | 91.6 | 97.6 | 89.6 | 92.8 | 94.4 | 85.6 | 95.2 | 87.2 | 93.6 | 95.2 | 85.6 | 89.6 | 92.8 | 92.0 | 94.4 | 91.2 | 86.4 | 92.0 | 94.4 | 3.5 | 12.0 |
| GPT-4.5-preview † | 91.5 | 96.8 | 92.8 | 92.8 | 88.0 | 88.0 | 92.0 | 88.0 | 91.2 | 95.2 | 84.8 | 91.2 | 94.4 | 93.6 | 92.8 | 92.0 | 86.4 | 92.8 | 93.6 | 3.1 | 12.0 |
| **Reasoning LLMs** | | | | | | | | | | | | | | | | | | | | | |
| Deepseek-R1-671B | 92.4 | 96.8 | 88.8 | 96.0 | 89.6 | 88.8 | 96.8 | 89.6 | 95.2 | 97.6 | 88.8 | 92.8 | 94.4 | 95.2 | 92.8 | 86.4 | 86.4 | 92.8 | 95.2 | 3.6 | 11.2 |
| Qwen-QwQ-32B | 89.9 | 96.0 | 92.0 | 93.6 | 92.8 | 89.6 | 94.4 | 88.8 | 92.8 | 94.4 | 83.2 | 92.0 | 93.6 | 96.0 | 90.4 | 76.0 | 68.8 | 90.4 | 93.6 | 6.9 | 27.2 |
| Qwen3-235B-A22B-Thinking | 92.5 | 97.6 | 91.2 | 93.6 | 96.0 | 88.8 | 96.8 | 89.6 | 95.2 | 96.0 | 90.4 | 92.0 | 95.2 | 94.4 | 93.6 | 84.0 | 85.6 | 92.0 | 93.6 | 4.3 | 13.6 |
| Claude-3.7-sonnet-thinking † | 90.8 | 97.6 | 91.2 | 92.8 | 91.2 | 86.4 | 92.8 | 85.6 | 94.4 | 92.8 | 88.4 | 90.4 | 96.0 | 92.0 | 89.6 | 84.8 | 85.6 | 90.4 | 92.8 | 3.5 | 12.8 |
| Gemini-2.0-flash-thinking † | 87.3 | 97.6 | 84.0 | 86.4 | 80.0 | 83.2 | 94.4 | 89.6 | 93.6 | 96.8 | 60.8 | 84.8 | 93.6 | 94.4 | 91.2 | 91.2 | 71.2 | 87.2 | 91.2 | 9.1 | 36.8 |
| Gemini-2.5-pro † | 86.4 | 90.4 | 84.0 | 88.0 | 84.8 | 87.2 | 84.8 | 88.0 | 91.2 | 90.4 | 80.8 | 84.8 | 93.6 | 83.2 | 86.4 | 84.0 | 81.6 | 88.8 | 82.4 | 3.8 | 12.8 |
| OpenAI-o1-mini † | 89.8 | 96.0 | 91.2 | 89.6 | 84.8 | 84.8 | 92.8 | 88.0 | 90.4 | 96.8 | 84.8 | 89.6 | 91.2 | 91.2 | 92.0 | 86.4 | 80.8 | 92.0 | 88.8 | 3.8 | 16.0 |
| OpenAI-o3-mini-medium † | 89.8 | 93.6 | 94.4 | 87.2 | 80.8 | 89.6 | 90.4 | 95.2 | 90.4 | 88.0 | 84.8 | 84.0 | 92.0 | 89.6 | 92.0 | 85.6 | 95.2 | 91.2 | 92.0 | 4.3 | 14.4 |

**Prompts.** In addition to the original input problem $Q$, we append the instruction *"Note: Please put the final answer in $\boxed{}$."* after it to help extract the final answer. Each language uses its own version of this instruction, as detailed in Appendix E.2.

**Evaluation.** We set the maximum output token limit to 65,536. For models with stricter limits (*e.g.*, Claude-3.7-sonnet with 64,000 tokens), we adhere to their respective limits. At each difficulty level, we compute accuracy (ACC) for every model-language pair, using the pass@1 metric. For

non-reasoning LLMs, we apply greedy decoding on open-source models. For reasoning models, where greedy decoding often leads to instability and repetition [14], we use sampling with $T = 0.6$, $p = 0.95$, and $k = 20$ for open-source models following [33]. Each model is tested 16 times under fixed hyperparameters, and we report the average as `average@16`. Detailed sampling procedures and standard deviation analyses are provided in Appendix E.4. The final benchmark scores are computed using the DW-ACC metric introduced in Section 2.5. For answer verification, we employ a rule-based matching script that achieves over 98% precision based on sampled human inspection. The evaluation code is available at: `https://github.com/QwenLM/PolyMath`.

## 3.2 Main Results

Table 5 presents the detailed ACC scores for each level, while Figure 4 in Appendix B shows the overall benchmark score leaderboard with DW-ACC. Our findings are as follows:

**PolyMath Differentiates Reasoning Performances.** From the average ACCs across different languages, the four levels of PolyMath effectively differentiate the reasoning abilities of LLMs:

- **Absolute Performances.** Qwen3-235B-A22B-Thinking outperforms all other LLMs across all levels, with its advantage growing at higher levels. At the top level, some non-reasoning models (*e.g.*, Llama-3.3-70B-Instruct) fail almost completely, while reasoning models like Gemini-2.5-pro and Qwen3-235B-A22B-Thinking can achieve 40% ACCs, demonstrating the strength of reasoning LLMs. However, at the low level, non-reasoning models may outperform reasoning models, suggesting that simple tasks may not benefit from slow-thinking processes. This highlights that low-difficulty benchmarks may underestimate true reasoning abilities of advanced LLMs.

- **Performances Across Levels.** All LLMs show performance drops as difficulty increases, but the rate of decline varies by model type. Non-Reasoning LLMs exhibit sharp, often geometric drops (e.g., ChatGPT-4o-latest: $91.6 \rightarrow 40.8 \rightarrow 20.4 \rightarrow 13.7$; Llama3.3-70B-Instruct: $64.2 \rightarrow 16.8$ from low to medium level). In contrast, reasoning LLMs degrade more gradually (e.g., Gemini-2.0-flash-thinking: $87.3 \rightarrow 59.4 \rightarrow 42.9 \rightarrow 22.3$), with some maintaining relative stability (e.g., Gemini-2.5-pro: $72.0 \rightarrow 62.2$ from medium to high level). These trends suggest that *reasoning LLMs are more stable to increasing difficulty, reflecting stronger reasoning capabilities*.

**PolyMath Reveals Language Gaps.** The last two columns of Table 5 show the ACC differences (*std.* and *range*) across languages for each LLM, revealing the following language gaps:

- As levels increase, although ACC steadily decreases, all models maintain high language gaps. At the higher three levels, *range* typically stays around 10%, with some reasoning models, such as Qwen-QwQ-32B, reaching nearly 20%. Since fluctuations in multiple runs are mostly within 0.5–1.5% ACC (see Appendix E.4), these gaps are significant. These results underscore that *bridging cross-lingual reasoning gaps remains a major challenge for advanced LLMs*.

- In addition, at the same difficulty level, stronger reasoning LLMs tend to exhibit larger language gaps. For example, in the three higher levels, Qwen-QwQ-32B consistently shows the highest *std.* and *range*. Models such as OpenAI-o3-mini-medium, Deepseek-R1-671B, and Gemini-2.5-pro also display significant language gaps at some levels, though inconsistently. In contrast, models such as OpenAI-o1-mini and Gemini-2.0-flash-thinking generally maintain smaller gaps. The notable exception is Qwen3-235B-A22B-Thinking, which achieves both strong reasoning performance and a relatively small language gap, largely because it has conducted multilingual slow-thinking alignment during the post-training phase [33]. These results underscore *the urgency of migrating slow-thinking abilities in multilingual contexts for reasoning LLMs*.

## 3.3 Input-Output Language Consistency

Beyond performance, Language Consistency (LC) between input and output is critical, especially for users who only understand their native language. If LLMs reply in a different language, they require extra translation and harm the user experience. Therefore, we next provide a detailed analysis of LC.

We define LC as follows. For a language $\mathcal{L}$ and model $\mathcal{M}$, let there be $n$ input queries, we identify the language(s) of each output $a$ for input $q$ using the `langdetect` Python library. Let $f(\cdot)$ denote the detected language(s). If the output contains exactly one language ($|f(a)| = 1$) and it matches

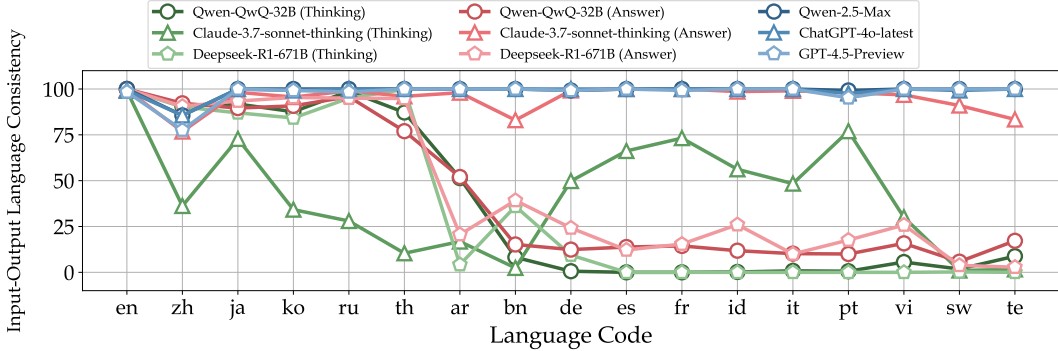

Figure 3: The **input-output language consistency** of different LLMs across each language and level. 100 indicates complete consistency, while 0 indicates complete inconsistency.

the input ($f(a) = f(q) = \mathcal{L}$), we consider the output language-consistent. Let $\mathbb{I}(\cdot)$ be the indicator function, then the LC of model $\mathcal{M}$ in language $\mathcal{L}$, denoted as $\mathrm{LC}_\mathcal{M}^\mathcal{L}$, is defined as:

$$\mathrm{LC}_\mathcal{M}^\mathcal{L} = \frac{\sum_{i=1}^n \mathbb{I}(|f(a_i)| = 1 \wedge f(a_i) = f(q_i) = \mathcal{L})}{n}, \tag{2}$$

**Overall LC.** We report LC results for various non-reasoning and reasoning LLMs under each language in Figure 3. For reasoning LLMs, we further separate the output into *thinking* and *answer* parts[3]. The left half of Table 17 in Appendix F.1 presents the average LC and the standard deviation across all languages at each level, and we have the following findings:

- Non-reasoning LLMs consistently achieve near-perfect language consistency (over 99% on average) across all levels, indicating that their language alignment is generally well handled.

- Reasoning LLMs consistently exhibit lower LC scores. For models like Qwen-QwQ-32B and Deepseek-R1-671B, both the thinking and answer parts remain around 40% with little variation across difficulty levels. Claude-3.7-sonnet-thinking shows similarly low LC in the thinking part (∼40%) but achieves much higher LC in answers (∼90%). For languages with low LC, the thinking language is often dominated by English or Chinese. In addition, LC varies widely across languages: Qwen-QwQ-32B achieves near-perfect LC in English (en), Chinese (zh), Japanese (ja), Korean (ko), and Russian (ru), but close to zero in many other languages.

Appendix G showcases various instances of LLM responses and the languages utilized. These results indicate that *language alignment is still a key challenge for reasoning LLMs — particularly in the thinking stage*, but current alignment degrees vary across models.

**Correlation Between LC and Performance.** We also examine the relationship between LC and performance in reasoning models and find notable correlations. ***For some reasoning models, lower LC may indicate better reasoning performance***. Details are shown in Appendix F.1.

### 3.4 Language Control

In Section 3.3, we have observed that reasoning LLMs exhibit varying LC across languages, suggesting a tendency to avoid less proficient languages during slow thinking. This raises a further question: *Can explicitly controlling the language used by reasoning LLMs improve their performance?*

To investigate this, we introduce three types of language control instructions in the input prompts: (1) forcing the same language as the query for response; (2) forcing English for response; and (3) allowing the model to choose the language it is proficient in for response freely. Detailed prompt templates are provided in Appendix E.3.

---

[3]Note that OpenAI's o series and the Gemini series do not provide access to their output contents of thinking processes, so we exclude them from our language analysis.

Table 6: The benchmark scores (DW-ACC) of different reasoning LLMs after adding **Language Control** in the instructions. **Bold** indicates the best performance overall. Blue shading shows the best-performing language for each model, and red shading indicates the poorest-performing language.

| | avg. | en | zh | ar | bn | de | es | fr | id | it | ja | ko | ms | pt | ru | sw | te | th | vi | std. | range |
|---|---|---|---|---|---|---|---|---|---|---|---|---|---|---|---|---|---|---|---|---|---|
| Qwen-QwQ-32B | 45.9 | 52.5 | 47.3 | 41.6 | 44.6 | 50.7 | 52.3 | 49.8 | 47.5 | 49.6 | 39.4 | 40.1 | 48.9 | 50.0 | 45.3 | 40.0 | 36.8 | 41.4 | 49.1 | 5.4 | 15.7 |
| + Forcing Output in Query-Language | 43.4 | 51.0 | 45.9 | 42.0 | 38.8 | 43.5 | 50.0 | 49.2 | 48.5 | 46.5 | 38.2 | 38.8 | 47.8 | 48.6 | 43.9 | 28.0 | 32.5 | 42.3 | 45.2 | 6.4 | 23.0 |
| + Forcing Output in English | 47.9 | 51.0 | 46.7 | 50.0 | 47.0 | 49.5 | 51.0 | 48.0 | 48.8 | 50.0 | 47.6 | 49.4 | 49.8 | 50.3 | 45.9 | 37.3 | 43.5 | 46.8 | 49.0 | 3.0 | 13.7 |
| + Forcing Output in Preferred Language | 46.2 | 50.9 | 45.7 | 45.6 | 45.1 | 48.8 | 50.1 | 50.8 | 48.4 | 49.6 | 40.0 | 39.9 | 49.7 | 49.4 | 45.4 | 37.0 | 39.2 | 44.9 | 49.7 | 4.1 | 13.9 |
| Deepseek-R1-671B | 47.0 | 48.0 | 44.9 | 46.3 | 45.9 | 49.6 | 50.9 | 47.4 | 47.1 | 49.9 | 45.6 | 42.3 | 46.7 | 49.7 | 49.2 | 45.1 | 43.4 | 44.8 | 50.1 | 2.4 | 8.6 |
| + Forcing Output in Query-Language | 46.3 | 49.0 | 45.6 | 43.3 | 42.3 | 46.1 | 48.7 | 48.0 | 46.8 | 49.2 | 44.6 | 42.0 | 47.4 | 47.5 | 46.4 | 43.8 | 45.0 | 45.2 | 48.0 | 2.4 | 7.2 |
| + Forcing Output in English | 47.6 | 49.0 | 43.3 | 47.7 | 46.2 | 49.0 | 48.9 | 47.4 | 44.2 | 51.4 | 48.6 | 45.0 | 49.0 | 48.1 | 47.5 | 45.3 | 45.4 | 44.3 | 49.2 | 2.3 | 8.1 |
| + Forcing Output in Preferred Language | 46.8 | 49.3 | 45.5 | 45.1 | 44.2 | 48.5 | 48.3 | 48.8 | 46.9 | 49.2 | 44.4 | 43.0 | 48.9 | 48.5 | 46.9 | 46.8 | 45.1 | 45.8 | 47.8 | 2.0 | 6.3 |
| Claude-3.7-sonnet-thinking | 33.5 | 35.7 | 32.6 | 34.6 | 33.7 | 32.3 | 30.2 | 32.7 | 34.0 | 34.6 | 34.2 | 32.8 | 34.8 | 31.8 | 35.6 | 34.2 | 30.9 | 36.0 | 31.5 | 1.6 | 5.8 |
| + Forcing Output in Query-Language | 32.5 | 34.8 | 34.5 | 30.8 | 32.6 | 31.1 | 31.4 | 30.9 | 32.8 | 34.1 | 32.9 | 31.8 | 32.7 | 30.8 | 32.9 | 35.1 | 30.9 | 32.8 | 31.1 | 1.4 | 4.3 |
| + Forcing Output in English | 34.9 | 34.8 | 36.0 | 36.9 | 33.2 | 35.5 | 32.7 | 35.4 | 33.3 | 34.1 | 38.1 | 33.8 | 36.4 | 34.5 | 34.6 | 33.7 | 33.2 | 36.3 | 34.5 | 1.4 | 5.4 |
| + Forcing Output in Preferred Language | 34.7 | 35.3 | 34.9 | 38.9 | 34.3 | 33.1 | 34.7 | 33.5 | 33.2 | 34.5 | 32.3 | 34.7 | 35.0 | 33.7 | 33.5 | 38.8 | 33.0 | 37.1 | 34.6 | 1.8 | 6.6 |

**Performances After Control.** We test three reasoning LLMs under the above language control settings, with results summarized in Table 6. Our findings are as follows:

- Forcing responses in English yields the best performance and reduces language disparities. This is especially helpful for low-performing languages (*e.g.*, in Qwen-QwQ-32B, Arabic: 41.6 → 50.0, Japanese: 39.4 → 47.6, Telugu: 36.8 → 43.5), suggesting that reasoning in English helps compensate for weaknesses in them. In contrast, allowing models to choose their preferred language does not outperform enforcing English, though some models still show modest gains.

- Forcing the response language to match the query leads to the poorest performance, often amplifying cross-lingual variance. This effect is particularly pronounced in languages where the model itself performs poorly, such as Swahili (sw) and Telugu (te) in Qwen-QwQ-32B, where performance drops significantly (sw: 40.0 → 28.0; te: 36.8 → 32.5).

These results empirically demonstrate ***the potential of output language control in enhancing the multilingual reasoning capabilities of LLMs***, and in general, for reasoning LLMs with poor LC, the advantage of forcing them to think and answer in English is more effective.

**Instruction-following Degree Under Control.** We further analyze how well LLMs follow language control instructions. Reasoning LLMs show significantly better adherence when forced to respond in English compared to when forced to use the query language. Interestingly, ***in languages where models typically follow the query language well, this constraint often leads to performance drops***. Detailed results and analyses are provided in Appendix F.2.

### 3.5 Thinking Length Across Languages

We also analyze LLM thinking lengths in multilingual contexts and find that reasoning LLMs can exhibit slow-thinking behavior across all languages, while non-reasoning models often struggle with cross-lingual behavior alignment. Also, switching to a language that triggers longer thinking does not always improve reasoning performance on the same problem. Details are shown in Appendix F.3.

## 4 Conclusion

In this paper, we construct PolyMath, a multilingual mathematical benchmark with hierarchical difficulty levels. Our extensive experiments demonstrate the strong reasoning capabilities of reasoning LLMs, while revealing key challenges: significant performance gaps across languages, input-output inconsistencies, and varying thinking-length patterns. We also observe potential performance gains through explicit language control. We hope that this challenging benchmark and our findings can help advance the development of multilingual reasoning LLMs.

## Limitations

Due to the high cost of annotation, we have not yet expanded the current benchmark. However, compared to existing multilingual reasoning benchmarks, our dataset already offers a substantial increase in scale. Additionally, not all languages are annotated by native speakers, so fully idiomatic expression cannot be guaranteed, but we at least ensure that all annotations are accurate and fluent.

## Acknowledgement

This work was done during Yiming's internship at Qwen Team, Alibaba Group. It was supported by the Alibaba Research Intern Program. This paper was also supported by the General Program of National Natural Science Foundation of China (62176153). Finally, we sincerely express our gratitude to all the linguistics specialists who participated in this project and acknowledge their professionalism.

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

# Appendix

## A    Related Work and Discussion

Table 7: Comparison between PolyMath and other multilingual mathematical reasoning benchmarks.

| Benchmark Name | Difficulty | Annotator | Answer Type | Language Number | Sample Per Language | Total Data Size |
|---|---|---|---|---|---|---|
| MGSM [23] | Low | Expert | Numeric | 10 | 250 | 2500 |
| MSVAMP [4] | Low | Crowd-Sourcing | Numeric | 10 | 500 | 5000 |
| MT-AIME [24] | High | Machine | Numeric | 55 | 30 | 1750 |
| PolyMath (Ours) | Low, Medium High, Top | Expert | Numeric, Expression, Equation, Interval Set, Tuple | 18 | 500 | 9000 |

**Multilingual Mathematical Reasoning Benchmarks.**    Systematic multilingual mathematical reasoning benchmarks remain scarce. Table 7 provides a multi-dimensional comparison with existing benchmarks. Most prior multilingual research relies on MGSM [23], a translated version of GSM8K [9], which is too simple for modern reasoning LLMs and fails to reveal their true capabilities. MSVAMP [4] faces similar limitations due to its low difficulty. MT-AIME [24] is a more recent effort, but it is fully translated by LLMs, raising concerns about data quality. Its small sample size per language also leads to high variance. PolyMath overcomes these limitations, and further introduces a hierarchical difficulty design and supports more diverse and complex answer formats, unlike existing datasets that focus mainly on numerical answers. In general, PolyMath better matches the reasoning capabilities of advanced LLMs while ensuring high quality, scale, and diversity, making it a more robust and challenging multilingual reasoning benchmark for the current reasoning LLM era.

**Multilingual Research Challenges in Current LLMs.**    Despite rapid progress in reasoning LLMs, multilingual capability remains a key challenge. The Deepseek-R1 technical report [14] notes that language mixing in responses is still unresolved, echoing our findings on language consistency in Section 3.3. Recent surveys [12, 5] also highlight language alignment and low-resource language support as critical future directions — both empirically supported as challenges by our experiments in Section 3.2 and analysis in Section 3.3. Overall, multilingual reasoning is a promising but underexplored area, and PolyMath can offer a strong benchmark to drive progress.

# B    Overall Leaderboard of Benchmark Score

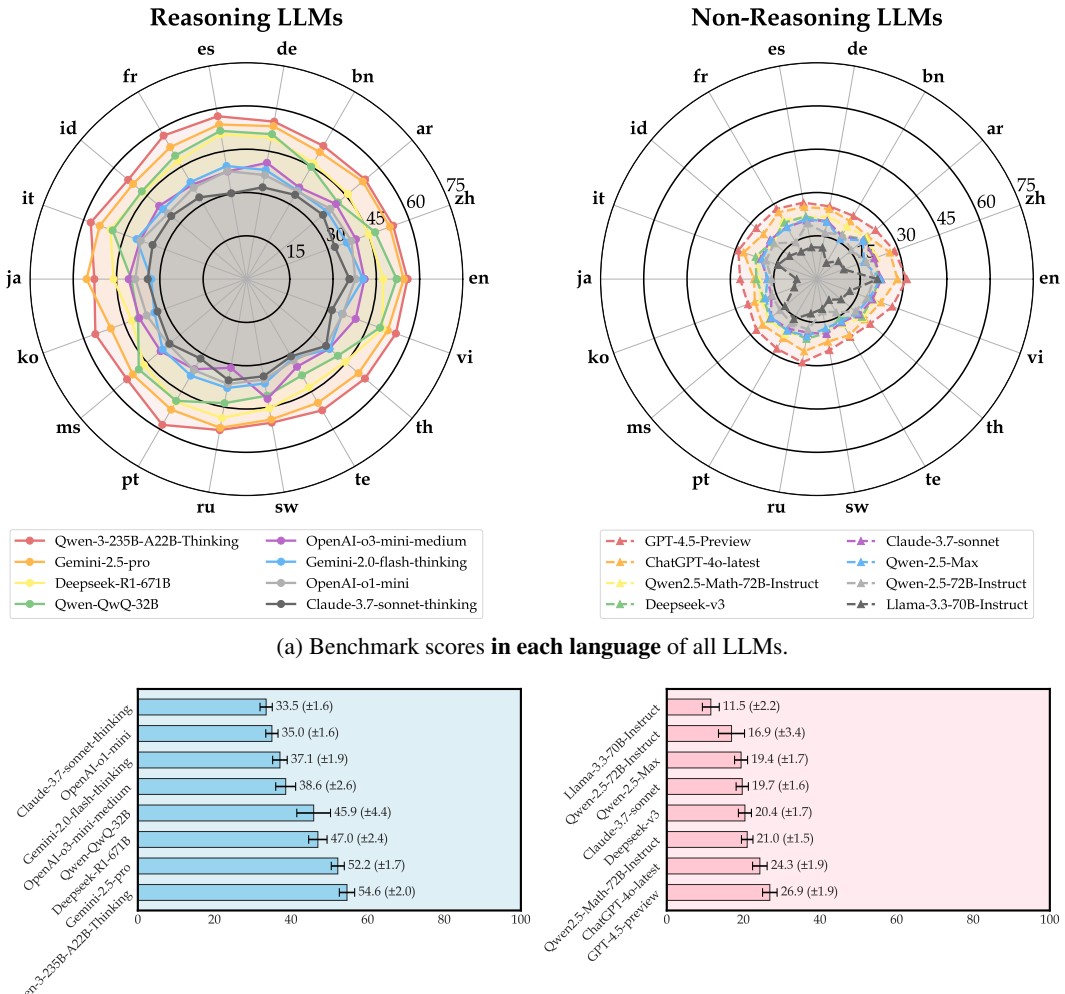

(a) Benchmark scores **in each language** of all LLMs.

(b) Average benchmark scores and standard variations **across languages** of all LLMs.

Figure 4: Overall benchmark scores of various advanced LLMs in our PolyMath. Refer to Section 2.5 for the detailed score calculation method, and the full leaderboard is shown in: `https://qwen-polymath.github.io/`.

# C    Human Annotation Process

## C.1    Annotator Background

We uphold high standards of professionalism among our annotation staff. Due to the difficulty in recruiting native speakers, we prioritized leveraging existing resources by collaborating with second-language experts who have a degree in linguistics and rich experience in specialized domain translation. For some low-resource languages where suitable second-language collaborators were harder to find, we then sought assistance from native speakers.

**Mathematical Term Extraction and Chinese Calibration.**    The first author of this paper, a Chinese Ph.D. candidate majoring in computer science with a strong mathematical foundation and competition experience, performs the two tasks.

**Calibration in Arabic, French, Italian, Japanese, Korean, Spanish, Thai, Vietnamese.**    We collaborated with a professional translation team and recruited language experts (non-native speakers) who hold degrees in their respective languages and have experience in at least three large-scale translations in fields such as science or literature.  Their translation expertise enables accurate terminology search and matching.

**Calibration in German, Indonesian, Portuguese, Russian.**    We engaged graduate students specializing in these languages from top universities (ensuring good mathematical ability) throughout the country, including:

- German: Feitong Sun (Tongji University)

- Indonesian: Qiqian Cang (Beijing Language and Culture University)

- Portuguese: Junxuan Wu (Beijing Foreign Studies University)

- Russian: Chenshu Sun (Peking University)

    Jiran Zhang (Shanghai International Studies University)

**Calibration in Bengali, Malay, Swahili, and Telugu.**    We directly recruited native speakers with a background in mathematics to perform annotation tasks in their respective languages.

All annotators (excluding the authors) were paid based on their individual or team rates, with full respect for their willingness to participate. As a result, there are no ethical concerns.

## C.2    Annotation Guidance

The annotation platform presented to annotators displays the **original English version** of each sample along with the **terms requiring attention**. Additionally, for some annotators from China, we also provide calibrated Chinese translations to help them understand the questions more quickly. However, English content remains the gold standard. During the annotation process, annotators should focus on three key aspects: (1) ensure terms are translated completely accurately; (2) express logic as smoothly and concisely as possible; (3) migrate formula blocks completely accurately. When submitting results, in addition to providing the **modified translations**, annotators must also **flag samples with term and fluency issues** for statistical analysis in Section 2.3. The annotation platform interface is shown in Figure 5, and the specific annotation guideline for annotators is as follows:

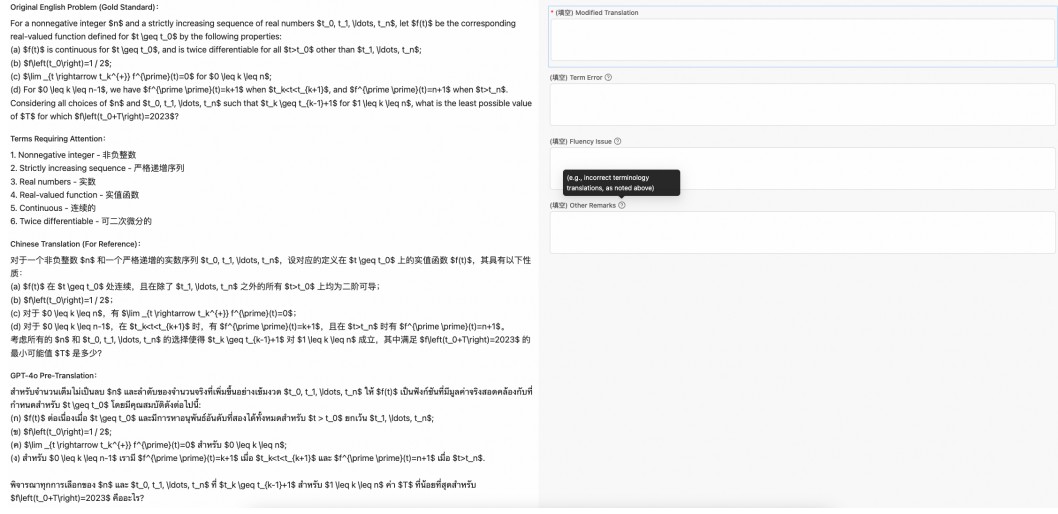

Figure 5: The annotation platform interface (take Thai for an example).

# Annotation Requirements:

- **Accurate Term Translation**: We provide key mathematical terms that need to be focused on. If you encounter terms that GPT-4o translates incorrectly, please modify them and extract them to list in three lines, including (the original English term, GPT-4o's incorrect translation, and the correct term translation).

- **Natural and Fluent Sentences**: Mathematical problems may involve nested conditions that require appropriate rearrangement of the text for the reader to understand the logic within the problem easily. You should carefully organize their expressions.

- **Overall Formula Migration without Translation**: Common formula environments include $ $, $$ $$, \[ \], \( \), and [asy] [/asy]. The content inside these environments should not be modified in any way. For example, in the following problem, the parts highlighted in red should be directly migrated (punctuation can be moved out), and it is important to check if GPT-4o has done this accurately. If there are migration errors inside the formula, corrections should be made based on the English version.

> Example: Let $P_n$ be the number of permutations $\pi$ of ${1,2,\dots,n}$ such that $|i-j|=1$ implies $|\pi(i)- \pi(j) | \le 2$ for all $i, j \in {1,2,\dots,n}$. Show that for $n \ge 2$, the quantity
> \[
> P_{n+5}-P_{n+4}-P_{n+3}+P_n
> \]
> does not depend on $n$, and find its value.

- **Complete Alignment of Translated Text with English Text**: There should be no information in the English text that is absent in the translated text. If there are redundant pieces of information in GPT-4o's translation, remove them directly.

- **Transliteration of Names**: Names can be transliterated directly, and specific phrasing is not important.

# Annotation Content:

- **Modified Translation**.

- **(Optional) Term Error**: Does the original GPT-4o translation contain major errors (e.g., affecting the logic of the problem, or incorrect term translations; a simple reason can be noted)? If yes, fill in "1", otherwise, leave it blank.

- **(Optional) Fluency Issue**: Does the original GPT-4o translation have fluency issues (e.g., odd translations of nested conditions; a simple reason can be noted)? If yes, fill in "1", otherwise, leave it blank.

- **(Optional) Other Remarks**: (e.g., incorrect terminology translations, as noted above).

# D Benchmark Metadata

## D.1 Specific Data Source

Table 8: Data sources for each level in PolyMath.

| Source Name | Source Type | | | Directly Sample From Existing Benchmarks | Problem Number |
|---|---|---|---|---|---|
| | Exercise | Exam | Competition | | |
| Low-level | | | | | |
| K-12 Mathematics | ✓ | | | MSGM [23] | 125 |
| Medium-level | | | | | |
| China GaoKao (Last Question) | | ✓ | | N/A | 52 |
| China KaoYan | | ✓ | | N/A | 10 |
| College Math | ✓ | | | N/A | 24 |
| CNMO (Preliminary Round) | | | ✓ | N/A | 12 |
| CMC (2021-2024) | | | ✓ | N/A | 17 |
| AMC (2012) | | | ✓ | N/A | 10 |
| High-level | | | | | |
| AIME (2015-2024) | | | ✓ | N/A | 47 |
| CNMO (2011-2024) | | | ✓ | N/A | 30 |
| CWMO (2023-2024) | | | ✓ | N/A | 5 |
| CGMO (2010-2024) | | | ✓ | N/A | 17 |
| IMC (2015-2023) | | | ✓ | N/A | 11 |
| SMMC (2023) | | | ✓ | N/A | 1 |
| CIIM (2019-2024) | | | ✓ | N/A | 6 |
| KMO (2022-2024) | | | ✓ | N/A | 3 |
| THMO (2018-2024) | | | ✓ | N/A | 5 |
| Top-level | | | | | |
| IMO (2022-2024) | | | ✓ | N/A | 4 |
| IMO Shortlist (2014-2023) | | | ✓ | N/A | 29 |
| Putnam (2015-2024) | | | ✓ | N/A | 30 |
| CMO (2011-2024) | | | ✓ | N/A | 18 |
| USAMO (2010-2023) | | | ✓ | N/A | 12 |
| ELMO (2022-2024) | | | ✓ | N/A | 3 |
| Alibaba Global Contest (2021) | | | ✓ | N/A | 4 |
| Frontier Math Challenge | ✓ | | | HLE [21] | 25 |

## D.2 Detailed Metadata of All Languages

Table 9: Metadata of PolyMath under all languages.

| | $avg._{std.}$ | en | zh | ar | bn | de | es | fr | id | it | ja | ko | ms | pt | ru | sw | te | th | vi |
|---|---|---|---|---|---|---|---|---|---|---|---|---|---|---|---|---|---|---|---|
| **Question Length** | | | | | | | | | | | | | | | | | | | |
| Low-level | $72.2_{8.9}$ | 59.9 | 63.7 | 76.2 | 70.4 | 70.6 | 65.2 | 71.8 | 64.0 | 69.4 | 67.3 | 76.9 | 66.7 | 66.5 | 72.9 | 88.7 | 96.5 | 82.1 | 71.5 |
| Middle-level | $101.2_{6.8}$ | 92.8 | 92.2 | 104.2 | 103.3 | 101.9 | 96.8 | 99.7 | 96.3 | 99.1 | 101.1 | 103.0 | 97.1 | 97.4 | 100.0 | 116.9 | 119.3 | 101.8 | 99.1 |
| High-level | $126.4_{10.7}$ | 112.8 | 113.8 | 130.5 | 127.7 | 127.2 | 120.1 | 124.1 | 122.4 | 122.8 | 124.5 | 128.1 | 119.4 | 120.3 | 125.2 | 151.4 | 155.8 | 129.1 | 119.6 |
| Top-level | $133.7_{11.7}$ | 117.4 | 119.2 | 136.8 | 134.6 | 133.2 | 126.1 | 130.9 | 125.5 | 129.6 | 130.4 | 138.1 | 128.8 | 126.6 | 132.1 | 160.9 | 164.6 | 139.3 | 132.3 |
| **Natural Language Coverage** | | | | | | | | | | | | | | | | | | | |
| Low-level | $97.4\%_{1.4\%}$ | 96.3% | 100.0% | 99.9% | 96.3% | 100.0% | 96.3% | 96.9% | 96.3% | 97.2% | 97.2% | 97.6% | 96.4% | 96.5% | 96.4% | 96.6% | 96.2% | 96.5% | 99.7% |
| Middle-level | $43.5\%_{3.7\%}$ | 39.0% | 37.5% | 43.7% | 45.0% | 44.1% | 41.8% | 43.6% | 42.3% | 42.8% | 42.7% | 41.9% | 42.3% | 42.0% | 43.6% | 52.9% | 52.5% | 43.8% | 42.3% |
| High-level | $55.7\%_{3.2\%}$ | 51.2% | 50.4% | 56.7% | 57.3% | 57.0% | 54.4% | 55.4% | 54.8% | 55.1% | 54.4% | 52.4% | 55.2% | 54.6% | 56.6% | 62.5% | 63.6% | 55.5% | 55.5% |
| Top-level | $54.7\%_{3.8\%}$ | 49.2% | 49.2% | 55.4% | 55.6% | 55.5% | 52.9% | 54.6% | 53.9% | 53.5% | 53.7% | 51.3% | 53.5% | 52.5% | 55.5% | 65.0% | 62.6% | 54.5% | 55.6% |

## D.3 Data Domain

We adopt different domain classification standards for questions at each level.

- **Low-level:** All questions at the low level are from K-12 mathematics, so classification is unnecessary.

- **Medium-level:** The medium-level questions are mostly derived from exams and exercises and cover a narrower range of topics. Most questions involve a single knowledge point. For example:

> Function $f(x) = \cos\omega x - 1$ $(\omega > 0)$ has 3 roots on the interval $[0, 2\pi]$, what is the range of $\omega$?

> The two focus points of ellipse $\frac{x^2}{9} + \frac{y^2}{6} = 1$ are $F_1$ and $F_2$. O is the origin. Point P is a point on the ellipse such that $\cos\angle F_1 P F_2 = \frac{3}{5}$. Find $|PO|$.

The first problem assesses trigonometric functions in the elementary algebra domain, and the second assesses conic sections in the elementary geometry domain. We partition the domain into subdomains (*i.e.*, specific knowledge points), with detailed statistics presented in Table 10.

- **High/Top-level:** For the high-level and top-level questions, the scope of the problems is broad, and each problem may involve diverse knowledge points. For example:

> Find all primes $p > 5$ for which there exists an integer $a$ and an integer $r$ satisfying $1 \leq r \leq p - 1$ with the following property: the sequence $1, a, a^2, \ldots, a^{p-5}$ can be rearranged to form a sequence $b_0, b_1, b_2, \ldots, b_{p-5}$ such that $b_n - b_{n-1} - r$ is divisible by $p$ for $1 \leq n \leq p - 5$.

Solving this problem draws on the number theory domain, including congruences, Fermat's Little Theorem, and primitive roots, and even touches on group theory in the algebra domain. Therefore, we do not further divide domains into subdomains. Based on these references [4][5][6][7][8][9][10], we established six major domains, with detailed domain statistics presented in Table 11.

---

[4] https://mathematics.stanford.edu/academics/graduate-students/phd-program/phd-qualifying-exams

[5] http://yau-contest.com/uploads/allfiles/exam/SyllabusonAlgebraandNumberTheory.pdf

[6] http://yau-contest.com/uploads/allfiles/exam/SyllabusonAnalysisandDifferentialEquations.pdf

[7] http://yau-contest.com/uploads/allfiles/exam/SyllabusonGeometryandTopology.pdf

[8] http://yau-contest.com/uploads/allfiles/exam/SyllabusonProbbilityandStatistics.pdf

[9] http://yau-contest.com/uploads/allfiles/exam/SyllabusonComputationalandAppliedMathematics.pdf

[10] https://www.imo-official.org/

Table 10: Domain statistics of PolyMath at the medium level.

| Domain | Question Number |
| --- | --- |
| **Elementary Algebra** | **47** |
| Equations and Inequalities | 11 |
| Elementary Functions | 6 |
| Trigonometric Functions | 12 |
| Sequences | 18 |
| **Elementary Geometry** | **23** |
| Plane Geometry | 8 |
| Solid Geometry | 4 |
| Analytic geometry (Conic Sections) | 11 |
| **Elementary Number Theory** | **4** |
| Prime Numbers | 1 |
| Divisibility | 2 |
| Greatest Common Divisors | 1 |
| **Combinatorics** | **16** |
| Probability and Statistics | 9 |
| Counting Principles | 2 |
| Binomial Theorem | 2 |
| Set Theory | 3 |
| **Calculus** | **30** |
| Limits | 8 |
| Derivative | 11 |
| Integral | 7 |
| Series | 4 |
| **Matrix Theory** | **5** |

Table 11: Domain statistics of PolyMath at the high and top levels.

| Domain | Question Number (high level) | Question Number (top level) |
| --- | --- | --- |
| Analysis and Differential Equations | 17 | 18 |
| Geometry and Topology | 16 | 11 |
| Algebra | 30 | 33 |
| Number Theory | 26 | 25 |
| Combinatorics and Probability | 28 | 35 |
| Applied Mathematics | 8 | 3 |

# E Experimental Settings

## E.1 Model Citation and Source

Table 12: Paper citations and URL source links of all models used in this paper. The symbol "†" indicates that the model is closed-source.

| Model Name | Snapshot | Citation | URL Source |
|---|---|---|---|
| *Non-Reasoning LLMs* | | | |
| Llama-3.3-70B-Instruct | — | [10] | https://huggingface.co/meta-llama/Llama-3.3-70B-Instruct |
| Qwen-2.5-72B-Instruct | — | [34] | https://huggingface.co/Qwen/Qwen2.5-72B-Instruct |
| Qwen-2.5-Math-72B-Instruct | — | [35] | https://huggingface.co/Qwen/Qwen2.5-Math-72B-Instruct |
| Deepseek-v3 | 2024-12-26 | [19] | https://huggingface.co/deepseek-ai/DeepSeek-V3 |
| Claude-3.7-sonnet† | 2025-02-19 | — | https://www.anthropic.com/news/claude-3-7-sonnet |
| Qwen-2.5-Max† | — | [34] | https://qwenlm.github.io/blog/qwen2.5-max/ |
| ChatGPT-4o-latest† | 2025-03-26 | [2] | https://openai.com/index/hello-gpt-4o/ |
| GPT-4.5-preview† | 2025-02-27 | — | https://openai.com/index/introducing-gpt-4-5/ |
| *Reasoning LLMs* | | | |
| Deepseek-R1-671B | — | [14] | https://huggingface.co/deepseek-ai/DeepSeek-R1 |
| Qwen-QwQ-32B | — | — | https://huggingface.co/Qwen/QwQ-32B |
| Qwen-3-235B-A22B-Thinking | — | [33] | https://qwenlm.github.io/blog/qwen3/ |
| Claude-3.7-sonnet-thinking† | 2025-02-19 | — | https://www.anthropic.com/news/claude-3-7-sonnet |
| Gemini-2.0-flash-thinking† | 2025-01-21 | — | https://deepmind.google/technologies/gemini/flash-thinking/ |
| Gemini-2.5-pro† | 2025-03-25 | — | https://blog.google/technology/google-deepmind/gemini-model-thinking-updates-march-2025/ |
| OpenAI-o1-mini† | 2024-09-12 | — | https://openai.com/index/openai-o1-mini-advancing-cost-efficient-reasoning/ |
| OpenAI-o3-mini-medium† | 2025-01-31 | — | https://openai.com/index/openai-o3-mini/ |

## E.2 Main Prompts

| Language | Instruction Prompt |
|---|---|
| en | Note: Please put the final answer in the $\boxed{}$. |
| zh | 注意：请将最终答案放在 $\boxed{}$ 中。 |
| ar | ملاحظة: يُرجى وضع الإجابة النهائية في $\boxed{}$. |
| bn | বিঃদ্রঃ: অনুগ্রহ করে চূড়ান্ত উত্তরটি $\boxed{}$ এর মধ্যে রাখুন।. |
| de | Hinweis: Bitte setzen Sie die endgültige Antwort in $\boxed{}$. |
| es | Nota: Por favor, coloque la respuesta final en el $\boxed{}$. |
| fr | Remarque : Veuillez mettre la réponse finale dans le $\boxed{}$. |
| id | Catatan: Silakan letakkan jawaban akhir di dalam $\boxed{}$. |
| it | Nota: Per favore, metti la risposta finale nel $\boxed{}$. |
| ja | 注意：最終的な答えを $\boxed{}$ に入れてください。 |
| ko | 참고: 최종 답안을 $\boxed{}$ 안에 넣어 주세요. |
| ms | Nota: Sila letakkan jawapan akhir dalam $\boxed{}$. |
| pt | Nota: Por favor, coloque a resposta final no $\boxed{}$. |
| ru | Примечание: Пожалуйста, поместите окончательный ответ в $\boxed{}$. |
| sw | Kumbuka: Tafadhali weka jibu la mwisho katika $\boxed{}$. |
| te | గమనిక: దయచేసి తుది జవాబును $\boxed{}$ లో ఉంచండి. |
| Th | หมายเหตุ: กรุณาใส่คำตอบสุดท้ายใน $\boxed{}$. |
| vi | Lưu ý: Vui lòng đặt câu trả lời cuối cùng trong $\boxed{}$. |

Figure 6: Instruction prompts appended after the input query in our main experiments.

## E.3 Language Control Prompts

| Lang. | Control Prompt (Query-Language) | Control Prompt (English) | Control Prompt (Preferred Language) |
|---|---|---|---|
| en | Use English to think and answer. | Use English to think and answer. | Choose the language you are most proficient in to think and answer. |
| zh | 使用中文进行思考和回答。 | 使用英文进行思考和回答。 | 自选一种你最擅长的语言进行思考和回答。 |
| ar | .استخدم العربية للتفكير والإجابة | استخدم اللغة الإنجليزية للتفكير والإجابة | اختر اللغة التي تجيدها أكثر للتفكير والإجابة |
| bn | বাংলা ব্যবহার করে চিন্তা এবং উত্তর দিন। | ইংরেজি ব্যবহার করে চিন্তা এবং উত্তর দিন। | আপনি যে ভাষাটি সবচেয়ে পারদর্শী সেটি বেছে নিয়ে চিন্তা এবং উত্তর দিন। |
| de | Verwende Deutsch, um zu denken und zu antworten. | Verwenden Sie Englisch, um zu denken und zu antworten. | Wählen Sie die Sprache, in der Sie am kompetentesten sind, um zu denken und zu antworten. |
| es | Usa español para pensar y responder. | Usa inglés para pensar y responder. | Elige el idioma en el que eres más competente para pensar y responder. |
| fr | Utilisez le français pour penser et répondre. | Utilisez l'anglais pour penser et répondre. | Choisissez la langue dans laquelle vous êtes le plus compétent pour penser et répondre. |
| id | Gunakan bahasa Indonesia untuk berpikir dan menjawab. | Gunakan bahasa Inggris untuk berpikir dan menjawab. | Pilih bahasa yang paling Anda kuasai untuk berpikir dan menjawab. |
| it | Usa italiano per pensare e rispondere. | Usa inglese per pensare e rispondere. | Scegli la lingua in cui sei più competente per pensare e rispondere. |
| ja | 日本語を使って考え、回答してください。 | 英語を使って考え、回答してください。 | 最も得意な言語を選んで考え、回答してください。 |
| ko | 한국어로 생각하고 답변하세요. | 영어로 생각하고 답변하세요. | 가장 능숙한 언어를 선택하여 생각하고 답변하세요. |
| ms | Gunakan bahasa Melayu untuk berfikir dan menjawab. | Gunakan bahasa Inggeris untuk berfikir dan menjawab. | Pilih bahasa yang paling anda mahir untuk berfikir dan menjawab. |
| pt | Use português para pensar e responder. | Use inglês para pensar e responder. | Escolha o idioma em que você é mais competente para pensar e responder. |
| ru | Используйте русский язык для размышлений и ответов. | Используйте английский язык, чтобы думать и отвечать. | Выберите язык, в котором вы наиболее компетентны, чтобы думать и отвечать. |
| sw | Tumia Kiswahili kufikiri na kujibu. | Tumia Kiingereza kufikiri na kujibu. | Chagua lugha ambayo unamudu zaidi kufikiri na kujibu. |
| te | తెలుగును ఉపయోగించి ఆలోచించి సమాధానం ఇవ్వండి. | ఇంగ్లీసును ఉపయోగించి ఆలోచించి ఉత్తరించండి. | మీరు అత్యంత ప్రావీణ్యం ఉన్న భాషను స్పెక్టింఛి ఆలోచించి. |
| Th | ใช้ภาษาไทยในการคิดและตอบคำถาม. | ใช้ภาษาอังกฤษในการคิดและตอบคำถาม. | เลือกภาษาที่คุณมีความสามารถมากที่สุดในการคิดและตอบคำถาม. |
| vi | Sử dụng tiếng Việt để suy nghĩ và trả lời. | Sử dụng tiếng Anh để suy nghĩ và trả lời. | Chọn ngôn ngữ mà bạn thành thạo nhất để suy nghĩ và trả lời. |

Figure 7: Language control prompts.

## E.4 Sampling Details

Greedy decoding in reasoning LLMs often leads to instability and repetition [14], so we use sampling-based decoding instead in reasoning LLMs. For each model, language, and difficulty level ($N = 125$ samples), we run 16 trials with identical hyperparameters and report the average accuracy as average16. Denoting the number of correct answers in trial $i$ by $n_i$, to preserve accuracy granularity (in 0.8 increments), we compute the average number of correct answers across runs, round it to the nearest integer, and divide by the total number of samples:

$$\texttt{average@16} = \frac{\left\lfloor \frac{1}{16} \sum_{i=1}^{16} n_i \right\rfloor}{N} \tag{3}$$

We report the standard deviation of accuracies under each level and language for each reasoning LLM, as shown in Tables 13, 14, 15, and 16. Most standard deviations are within the range of 0.5 to 1.5, corresponding to under two questions' variation (0.8% per question). Considering the sensitivity of reasoning models to sampling and hyperparameters [1, 8], this level of fluctuation is reasonable.

Table 13: Standard deviation of accuracies across 16 tests for each reasoning LLM at the low level.

| | Average | en | zh | ar | bn | de | es | fr | id | it | ja | ko | ms | pt | ru | sw | te | th | vi |
|---|---|---|---|---|---|---|---|---|---|---|---|---|---|---|---|---|---|---|---|
| Deepseek-R1-671B | 0.52 | 0.33 | 0.20 | 0.33 | 1.65 | 0.39 | 0.20 | 0.39 | 0.20 | 0.39 | 0.77 | 0.51 | 0.57 | 0.20 | 0.57 | 0.82 | 1.05 | 0.52 | 0.33 |
| Qwen-QwQ-32B | 0.43 | 0.40 | 0.70 | 0.20 | 0.69 | 0.23 | 0.23 | 0.33 | 0.52 | 0.39 | 0.39 | 0.51 | 0.20 | 0.39 | 0.20 | 0.57 | 0.95 | 0.69 | 0.23 |
| Qwen-3-235B-A22B-Thinking | 0.44 | 0.41 | 0.66 | 0.26 | 0.53 | 0.29 | 0.21 | 0.39 | 0.48 | 0.36 | 0.45 | 0.50 | 0.22 | 0.43 | 0.25 | 0.61 | 0.88 | 0.74 | 0.27 |
| Claude-3.7-sonnet-thinking | 0.38 | 0.33 | 0.60 | 0.33 | 0.33 | 0.51 | 0.20 | 0.20 | 0.20 | 0.23 | 1.40 | 0.20 | 0.57 | 0.33 | 0.20 | 0.20 | 0.00 | 0.60 | 0.39 |
| Gemini-2.0-flash-thinking | 0.89 | 1.02 | 0.63 | 0.70 | 1.82 | 0.35 | 0.91 | 1.26 | 0.76 | 0.39 | 0.56 | 0.43 | 0.48 | 0.98 | 0.85 | 1.21 | 1.47 | 0.75 | 1.40 |
| Gemini-2.5-pro | 0.99 | 0.72 | 2.07 | 0.99 | 0.20 | 1.61 | 0.84 | 0.72 | 1.64 | 0.82 | 0.94 | 0.49 | 0.72 | 0.82 | 0.82 | 1.00 | 0.58 | 0.64 | 0.99 |
| OpenAI-o1-mini | 0.74 | 0.52 | 0.66 | 0.52 | 1.06 | 0.77 | 0.73 | 0.87 | 0.39 | 0.46 | 1.18 | 0.23 | 0.70 | 0.70 | 0.39 | 1.46 | 1.88 | 0.51 | 0.39 |
| OpenAI-o3-mini-medium | 0.75 | 1.27 | 0.33 | 0.60 | 0.51 | 1.20 | 1.05 | 0.69 | 0.82 | 0.82 | 0.82 | 0.87 | 0.23 | 0.70 | 0.39 | 0.51 | 1.18 | 0.57 | 0.40 |

Table 14: Standard deviation of accuracies across 16 tests for each reasoning LLM at the medium level.

| | Average | en | zh | ar | bn | de | es | fr | id | it | ja | ko | ms | pt | ru | sw | te | th | vi |
|---|---|---|---|---|---|---|---|---|---|---|---|---|---|---|---|---|---|---|---|
| Deepseek-R1-671B | 0.85 | 1.18 | 1.71 | 0.87 | 0.52 | 0.51 | 0.20 | 0.82 | 0.20 | 0.39 | 0.87 | 0.60 | 1.15 | 1.68 | 0.46 | 1.44 | 1.48 | 0.20 | 1.06 |
| Qwen-QwQ-32B | 0.83 | 0.89 | 1.10 | 1.32 | 0.52 | 1.16 | 0.57 | 0.57 | 1.05 | 0.89 | 0.80 | 1.58 | 0.40 | 0.39 | 0.82 | 1.20 | 0.39 | 0.73 | 0.51 |
| Qwen-3-235B-A22B-Thinking | 0.82 | 0.92 | 0.79 | 1.04 | 0.60 | 1.22 | 0.49 | 0.68 | 1.00 | 0.84 | 0.95 | 1.41 | 0.47 | 0.56 | 0.70 | 1.09 | 0.42 | 0.87 | 0.63 |
| Claude-3.7-sonnet-thinking | 1.34 | 1.05 | 2.42 | 1.37 | 1.80 | 0.60 | 0.39 | 1.83 | 1.64 | 0.60 | 1.28 | 0.98 | 1.74 | 2.50 | 0.73 | 1.58 | 0.87 | 1.64 | 1.18 |
| Gemini-2.0-flash-thinking | 0.76 | 0.38 | 0.82 | 0.44 | 0.48 | 0.58 | 0.77 | 1.00 | 0.45 | 0.59 | 0.89 | 0.39 | 0.58 | 1.16 | 0.90 | 0.90 | 1.45 | 1.22 | 0.63 |
| Gemini-2.5-pro | 0.93 | 1.26 | 0.86 | 1.14 | 1.42 | 0.37 | 0.10 | 0.94 | 0.20 | 1.11 | 0.45 | 0.41 | 0.93 | 1.20 | 0.84 | 0.86 | 2.05 | 1.49 | 1.11 |
| OpenAI-o1-mini | 1.14 | 1.24 | 0.76 | 1.15 | 1.15 | 1.71 | 0.87 | 0.89 | 0.57 | 1.51 | 0.89 | 0.87 | 0.52 | 1.85 | 1.54 | 0.77 | 1.37 | 1.37 | 1.51 |
| OpenAI-o3-mini-medium | 1.21 | 1.65 | 1.95 | 0.39 | 0.39 | 1.01 | 1.27 | 1.24 | 1.37 | 1.48 | 1.15 | 0.70 | 1.61 | 1.16 | 1.36 | 1.57 | 1.80 | 0.87 | 0.76 |

Table 15: Standard deviation of accuracies across 16 tests for each reasoning LLM at the high level.

| | Average | en | zh | ar | bn | de | es | fr | id | it | ja | ko | ms | pt | ru | sw | te | th | vi |
|---|---|---|---|---|---|---|---|---|---|---|---|---|---|---|---|---|---|---|---|
| Deepseek-R1-671B | 1.08 | 0.87 | 1.36 | 0.70 | 0.89 | 1.44 | 1.74 | 1.61 | 1.91 | 1.31 | 1.20 | 0.70 | 1.32 | 0.95 | 0.76 | 0.57 | 0.66 | 0.76 | 0.76 |
| Qwen-QwQ-32B | 0.54 | 0.17 | 0.52 | 0.34 | 0.25 | 0.19 | 0.42 | 0.50 | 0.60 | 0.37 | 0.48 | 0.89 | 0.26 | 0.84 | 1.18 | 0.91 | 0.19 | 0.82 | 0.82 |
| Qwen-3-235B-A22B-Thinking | 0.57 | 0.26 | 0.45 | 0.30 | 0.20 | 0.55 | 0.84 | 0.67 | 0.52 | 0.79 | 0.63 | 0.28 | 0.60 | 0.42 | 0.91 | 1.02 | 0.33 | 0.73 | 0.73 |
| Claude-3.7-sonnet-thinking | 0.82 | 1.93 | 0.46 | 0.51 | 1.20 | 0.98 | 1.15 | 0.76 | 0.89 | 0.52 | 0.60 | 0.70 | 0.69 | 1.13 | 0.39 | 0.76 | 0.66 | 0.69 | 0.69 |
| Gemini-2.0-flash-thinking | 1.63 | 1.01 | 2.60 | 2.78 | 2.05 | 0.98 | 1.61 | 2.08 | 2.14 | 0.51 | 1.64 | 1.61 | 1.01 | 0.51 | 1.83 | 1.86 | 1.80 | 1.70 | 1.70 |
| Gemini-2.5-pro | 0.72 | 0.45 | 0.87 | 0.60 | 0.52 | 0.35 | 1.14 | 0.37 | 0.74 | 0.60 | 0.53 | 1.07 | 0.59 | 1.74 | 0.41 | 0.78 | 0.35 | 0.90 | 0.90 |
| OpenAI-o1-mini | 1.48 | 2.54 | 0.89 | 0.89 | 1.32 | 2.18 | 0.60 | 1.67 | 2.42 | 0.70 | 0.52 | 1.18 | 0.66 | 2.30 | 0.80 | 1.16 | 2.46 | 2.22 | 2.22 |
| OpenAI-o3-mini-medium | 1.03 | 0.61 | 1.46 | 1.27 | 0.69 | 1.01 | 0.98 | 1.41 | 1.24 | 1.37 | 0.95 | 1.37 | 0.57 | 1.18 | 1.27 | 0.69 | 0.69 | 0.87 | 0.87 |

Table 16: Standard deviation of accuracies across 16 tests for each reasoning LLM at the top level.

| | Average | en | zh | ar | bn | de | es | fr | id | it | ja | ko | ms | pt | ru | sw | te | th | vi |
|---|---|---|---|---|---|---|---|---|---|---|---|---|---|---|---|---|---|---|---|
| Deepseek-R1-671B | 1.04 | 0.70 | 0.83 | 1.04 | 0.76 | 0.69 | 1.20 | 0.90 | 0.89 | 0.33 | 1.97 | 0.52 | 1.76 | 0.60 | 1.81 | 1.36 | 1.78 | 0.60 | 1.12 |
| Qwen-QwQ-32B | 0.93 | 0.76 | 0.70 | 1.18 | 1.18 | 0.51 | 0.66 | 0.77 | 1.56 | 0.77 | 0.87 | 0.70 | 0.77 | 1.51 | 0.82 | 0.83 | 0.89 | 1.36 | 0.98 |
| Qwen-3-235B-A22B-Thinking | 0.97 | 0.79 | 0.85 | 1.12 | 1.25 | 0.58 | 0.92 | 1.00 | 1.43 | 0.69 | 0.73 | 0.97 | 0.88 | 1.38 | 0.77 | 0.95 | 0.81 | 1.31 | 1.22 |
| Claude-3.7-sonnet-thinking | 0.95 | 0.57 | 0.51 | 0.89 | 1.32 | 0.33 | 1.52 | 1.49 | 0.57 | 0.69 | 1.10 | 0.39 | 0.57 | 1.48 | 1.58 | 1.25 | 1.20 | 0.73 | 1.34 |
| Gemini-2.0-flash-thinking | 0.56 | 0.30 | 0.20 | 0.64 | 1.07 | 0.49 | 0.28 | 0.60 | 0.60 | 0.62 | 0.58 | 0.25 | 0.62 | 0.46 | 0.68 | 0.55 | 0.71 | 0.91 | 0.66 |
| Gemini-2.5-pro | 0.67 | 0.25 | 0.66 | 0.81 | 0.94 | 0.34 | 1.18 | 0.67 | 0.81 | 0.78 | 0.41 | 0.53 | 0.62 | 0.77 | 0.28 | 0.84 | 0.77 | 0.78 | 0.73 |
| OpenAI-o1-mini | 1.12 | 1.32 | 1.04 | 0.73 | 2.13 | 0.94 | 0.69 | 2.12 | 0.87 | 0.52 | 1.24 | 1.00 | 0.73 | 0.95 | 1.36 | 0.76 | 1.35 | 1.36 | 1.09 |
| OpenAI-o3-mini-medium | 1.02 | 1.32 | 0.40 | 1.15 | 1.18 | 0.20 | 1.05 | 1.25 | 0.39 | 1.06 | 1.48 | 0.57 | 0.52 | 1.54 | 0.83 | 1.62 | 0.80 | 2.00 | 1.45 |

# F  Further Analysis

## F.1  Input-Output Language Consistency (Continued)

Table 17: The average and standard deviation of **input-output language consistency** across all languages, under different LLMs and levels. We also present the **language-level correlation (Pearson correlation coefficient) between consistency and reasoning accuracy**.

| Model | Average Consistency Across Languages | | | | Correlation with Accuracy | | | |
|---|---|---|---|---|---|---|---|---|
| | low | medium | high | top | low | medium | high | top |
| **Non-Reasoning LLMs** | | | | | | | | |
| Qwen-2.5-Max | $99.1_{3.4}$ | $99.0_{3.3}$ | $99.6_{1.0}$ | $99.1_{3.3}$ | 0.10 | 0.24 | -0.45 | 0.63 |
| ChatGPT-4o-latest | $98.9_{3.4}$ | $97.2_{5.2}$ | $96.9_{5.8}$ | $97.8_{5.0}$ | 0.03 | 0.20 | 0.17 | 0.04 |
| GPT-4.5-Preview | $98.1_{5.2}$ | $99.5_{1.2}$ | $99.5_{1.2}$ | $99.4_{2.1}$ | -0.18 | -0.23 | 0.16 | -0.28 |
| **Reasoning LLMs** | | | | | | | | |
| Qwen-QwQ-32B (Thinking) | $38.0_{42.8}$ | $37.0_{41.2}$ | $36.9_{45.0}$ | $37.2_{44.5}$ | 0.21 | -0.32 | -0.75 | -0.59 |
| Deepseek-R1-671B (Thinking) | $40.4_{43.9}$ | $37.5_{43.9}$ | $36.5_{42.7}$ | $34.5_{42.3}$ | -0.03 | -0.56 | -0.67 | -0.66 |
| Claude-3.7-sonnet-thinking (Thinking) | $43.0_{28.6}$ | $41.6_{28.3}$ | $40.5_{30.4}$ | $40.4_{29.6}$ | 0.30 | -0.35 | -0.54 | 0.48 |
| Qwen-QwQ-32B (Answer) | $57.6_{29.6}$ | $38.6_{40.5}$ | $38.2_{41.3}$ | $38.0_{39.2}$ | 0.29 | -0.53 | -0.72 | -0.54 |
| Deepseek-R1-671B (Answer) | $67.6_{28.6}$ | $39.8_{42.1}$ | $38.5_{42.7}$ | $34.5_{42.3}$ | 0.23 | -0.56 | -0.68 | -0.66 |
| Claude-3.7-sonnet-thinking (Answer) | $94.0_{10.2}$ | $95.3_{5.8}$ | $94.8_{6.8}$ | $87.9_{18.1}$ | 0.24 | -0.33 | -0.41 | -0.60 |

**Correlation Between LC and Performance.**    We also examine whether there is a certain correlation between LC and ACC. The results are shown on the right side of Table 17, where we use the language-level Pearson correlation coefficient for measurement.

- For non-reasoning LLMs, LC and ACC show no significant correlation, likely because LLMs achieve near-perfect LC under all languages, leaving little room for variation.

- In contrast, for reasoning LLMs, LC shows a more noticeable correlation with level progression. For Qwen-QwQ-32B and Deepseek-R1-671B, LC and ACC exhibit a strong negative correlation, suggesting that lower language consistency is associated with better reasoning performance. Interestingly, we also observe that when these reasoning LLMs face language inconsistency, their response languages are predominantly English or Chinese. This is likely because their slow-thinking abilities are mainly developed in English and Chinese contexts, making slow-thinking dominant in these languages. Consequently, when these LLMs understand other languages but respond in English or Chinese, it may better evoke their reasoning abilities, resulting in improved performance.

These results indicate that ***the language used by the LLMs to think and answer can influence their reasoning performance to some extent***.

## F.2  Language Control (Continued)

**Instruction-following Degree Under Control.**    Furthermore, we examine how these LLMs follow our language control instructions. Table 18 shows the input-output language consistency when there is no language control and when LLMs are forced to respond in the query language. Table 19 shows the proportion of outputs in English when there is no language control and when LLMs are forced to respond in English. Our findings are as follows:

- When reasoning LLMs are forced to respond in English, they generally follow the instructions well, with over 90% of the thinking parts and nearly 100% of the answer parts. This enables them to effectively engage in English slow-thinking capabilities, boosting their reasoning performance. Despite this, the thinking part still shows a low proportion of English for certain languages, such as Chinese (zh, 33.2%) and Russian (ru, 36.2%) on Qwen-QwQ-32B. By coincidence, in these language contexts, the input-output language consistency is high even without language control (see Figure 3). This suggests that when an LLM has strong language consistency in a particular language, it becomes more difficult to switch its output language.

Table 18: The input-output language consistency (%) when LLMs are not forced (no any control) and forced to respond in the query language.

(a) **No Language Control**

| | avg. | en | zh | ar | bn | de | es | fr | id | it | ja | ko | pt | ru | sw | te | th | vi |
|---|---|---|---|---|---|---|---|---|---|---|---|---|---|---|---|---|---|---|
| **Thinking Part** | | | | | | | | | | | | | | | | | | |
| Qwen-QwQ-32B | 37.3 | 100.0 | 90.6 | 87.2 | 8.2 | 0.6 | 0.0 | 0.0 | 0.2 | 0.8 | 91.8 | 87.6 | 0.6 | 98.8 | 1.8 | 8.8 | 51.4 | 5.6 |
| Deepseek-R1-671B | 35.6 | 99.8 | 90.2 | 99.8 | 36.0 | 9.4 | 0.0 | 0.4 | 0.0 | 0.0 | 87.0 | 84.2 | 0.0 | 95.0 | 0.2 | 0.0 | 4.2 | 0.0 |
| Claude-3.7-sonnet-thinking | 41.4 | 99.0 | 36.2 | 10.4 | 2.4 | 49.8 | 66.2 | 73.2 | 56.2 | 48.4 | 72.8 | 34.2 | 77.0 | 28.0 | 1.0 | 1.6 | 16.8 | 30.0 |
| **Answer Part** | | | | | | | | | | | | | | | | | | |
| Qwen-QwQ-32B | 42.6 | 100.0 | 92.2 | 77.0 | 15.2 | 12.4 | 13.8 | 14.4 | 11.8 | 10.2 | 89.6 | 90.8 | 10.0 | 95.8 | 5.8 | 17.2 | 52.0 | 15.8 |
| Deepseek-R1-671B | 45.1 | 100.0 | 90.6 | 94.8 | 39.2 | 24.2 | 12.2 | 15.4 | 26.0 | 10.0 | 93.4 | 95.4 | 17.6 | 95.0 | 3.8 | 2.8 | 20.6 | 25.8 |
| Claude-3.7-sonnet-thinking | 94.8 | 99.6 | 76.8 | 96.0 | 83.0 | 99.2 | 100.0 | 100.0 | 98.6 | 99.0 | 98.2 | 95.6 | 98.0 | 99.0 | 91.0 | 83.4 | 98.0 | 96.8 |

(b) **Forcing Output in Query-Language**

| | avg. | en | zh | ar | bn | de | es | fr | id | it | ja | ko | pt | ru | sw | te | th | vi |
|---|---|---|---|---|---|---|---|---|---|---|---|---|---|---|---|---|---|---|
| **Thinking Part** | | | | | | | | | | | | | | | | | | |
| Qwen-QwQ-32B | 62.2 | 100.0 | 92.8 | 96.2 | 56.2 | 80.2 | 3.2 | 8.4 | 9.4 | 97.6 | 97.2 | 92.6 | 35.2 | 99.6 | 55.2 | 27.6 | 62.0 | 43.2 |
| Deepseek-R1-671B | 45.0 | 99.6 | 91.4 | 99.2 | 59.8 | 39.2 | 1.0 | 0.4 | 2.0 | 12.6 | 91.4 | 92.8 | 6.0 | 99.8 | 10.4 | 0.6 | 48.1 | 10.8 |
| Claude-3.7-sonnet-thinking | 85.4 | 99.6 | 84.1 | 95.4 | 50.4 | 98.1 | 98.4 | 98.4 | 97.8 | 98.2 | 97.1 | 89.1 | 96.4 | 94.4 | 35.9 | 23.6 | 97.3 | 98.4 |
| **Answer Part** | | | | | | | | | | | | | | | | | | |
| Qwen-QwQ-32B | 61.9 | 100.0 | 90.6 | 96.0 | 56.2 | 80.4 | 3.2 | 8.4 | 9.6 | 97.2 | 97.0 | 91.8 | 35.2 | 99.6 | 55.0 | 27.2 | 61.8 | 43.0 |
| Deepseek-R1-671B | 74.6 | 100.0 | 89.6 | 99.6 | 85.0 | 77.6 | 39.6 | 43.0 | 76.0 | 59.6 | 98.0 | 99.6 | 55.5 | 99.4 | 49.5 | 28.8 | 89.6 | 78.6 |
| Claude-3.7-sonnet-thinking | 98.4 | 100.0 | 79.4 | 100.0 | 99.4 | 99.8 | 99.6 | 100.0 | 99.4 | 100.0 | 99.8 | 99.0 | 98.2 | 99.8 | 99.2 | 99.8 | 100.0 | 100.0 |

Table 19: The proportion of outputs in English (%) when LLMs are not forced (no any control) and are forced to respond in English.

(a) **No Language Control**

| | avg. | en | zh | ar | bn | de | es | fr | id | it | ja | ko | pt | ru | sw | te | th | vi |
|---|---|---|---|---|---|---|---|---|---|---|---|---|---|---|---|---|---|---|
| **Thinking Part** | | | | | | | | | | | | | | | | | | |
| Qwen-QwQ-32B | 66.0 | 100.0 | 0.2 | 12.0 | 91.8 | 99.4 | 100.0 | 100.0 | 99.8 | 99.2 | 5.6 | 5.8 | 99.4 | 1.0 | 73.8 | 91.2 | 48.6 | 94.4 |
| Deepseek-R1-671B | 68.5 | 100.0 | 0.2 | 0.0 | 64.0 | 90.6 | 100.0 | 100.0 | 100.0 | 100.0 | 0.0 | 9.2 | 100.0 | 5.0 | 99.8 | 100.0 | 95.8 | 100.0 |
| Claude-3.7-sonnet-thinking | 63.5 | 100.0 | 55.6 | 89.8 | 97.6 | 49.2 | 34.4 | 26.6 | 42.4 | 51.2 | 26.4 | 65.0 | 20.8 | 72.0 | 99.0 | 97.8 | 83.0 | 69.4 |
| **Answer Part** | | | | | | | | | | | | | | | | | | |
| Qwen-QwQ-32B | 63.0 | 100.0 | 0.4 | 24.6 | 87.2 | 90.8 | 89.8 | 87.6 | 90.4 | 91.6 | 9.4 | 9.0 | 91.4 | 4.0 | 71.8 | 87.6 | 49.8 | 86.2 |
| Deepseek-R1-671B | 59.9 | 100.0 | 0.0 | 5.2 | 61.0 | 75.4 | 87.8 | 84.6 | 74.0 | 89.8 | 2.2 | 4.4 | 82.4 | 5.0 | 96.2 | 97.2 | 79.4 | 74.2 |
| Claude-3.7-sonnet-thinking | 8.8 | 100.0 | 1.8 | 0.2 | 13.2 | 0.8 | 0.0 | 0.0 | 0.6 | 0.0 | 1.0 | 3.6 | 0.0 | 0.4 | 8.2 | 15.8 | 1.0 | 2.2 |

(b) **Forcing Output in English**

| | avg. | en | zh | ar | bn | de | es | fr | id | it | ja | ko | pt | ru | sw | te | th | vi |
|---|---|---|---|---|---|---|---|---|---|---|---|---|---|---|---|---|---|---|
| **Thinking Part** | | | | | | | | | | | | | | | | | | |
| Qwen-QwQ-32B | 91.4 | 100.0 | 33.2 | 95.2 | 99.6 | 100.0 | 100.0 | 100.0 | 99.8 | 100.0 | 92.2 | 99.8 | 100.0 | 36.2 | 100.0 | 99.2 | 99.0 | 99.4 |
| Deepseek-R1-671B | 93.5 | 100.0 | 20.4 | 73.0 | 100.0 | 100.0 | 100.0 | 100.0 | 100.0 | 100.0 | 99.6 | 100.0 | 100.0 | 96.0 | 100.0 | 100.0 | 100.0 | 100.0 |
| Claude-3.7-sonnet-thinking | 99.9 | 99.8 | 99.8 | 100.0 | 99.8 | 100.0 | 99.8 | 100.0 | 100.0 | 100.0 | 99.8 | 99.8 | 99.8 | 100.0 | 99.8 | 100.0 | 100.0 | 100.0 |
| **Answer Part** | | | | | | | | | | | | | | | | | | |
| Qwen-QwQ-32B | 99.2 | 100.0 | 93.8 | 99.8 | 99.8 | 100.0 | 100.0 | 100.0 | 99.8 | 99.8 | 99.8 | 99.8 | 100.0 | 93.2 | 100.0 | 99.8 | 100.0 | 100.0 |
| Deepseek-R1-671B | 97.8 | 100.0 | 95.0 | 74.8 | 99.8 | 99.4 | 99.4 | 99.4 | 99.8 | 100.0 | 100.0 | 99.2 | 99.4 | 97.6 | 100.0 | 99.8 | 100.0 | 99.6 |
| Claude-3.7-sonnet-thinking | 99.3 | 100.0 | 100.0 | 98.8 | 99.8 | 99.6 | 99.4 | 98.3 | 99.8 | 99.8 | 99.8 | 99.6 | 99.6 | 99.8 | 99.1 | 94.6 | 99.8 | 100.0 |

- In contrast, when reasoning LLMs are forced to respond with the query language, they often struggle to follow the instructions properly, especially models like Qwen-QwQ-32B and Deepseek-R1-671B with inherently poor input-output consistency (see Figure 3). For languages that exhibit relatively better instruction following (e.g., in Qwen-QwQ-32B, input-output consistency for thinking improved: German (de): 0.6 → 80.2, Swahili (sw): 1.8 → 55.2, Italian (it): 0.8 → 97.6), reasoning performance often drops (de: 50.7 → 43.5, sw: 40.0 → 28.0, it: 49.6 → 46.5). This leads to an overall decline in reasoning ability (see Table 6), suggesting that ***forcing a response in a language with low consistency may push the model into a weaker slow-thinking pattern, ultimately harming performance***.

## F.3 Thinking Length Across Languages

Reasoning efficiency has become a central topic in the era of reasoning LLMs [25], often measured by output token length [15]. While prior work has explored issues hindering efficient thinking [7, 31] and proposed some solutions [32, 28] in monolingual settings, multilingual reasoning efficiency remains underexplored. This means that *how LLMs adjust their thinking length across languages is*

*still not well understood.* Therefore, in this part, we provide a preliminary analysis of LLM thinking length in multilingual contexts.

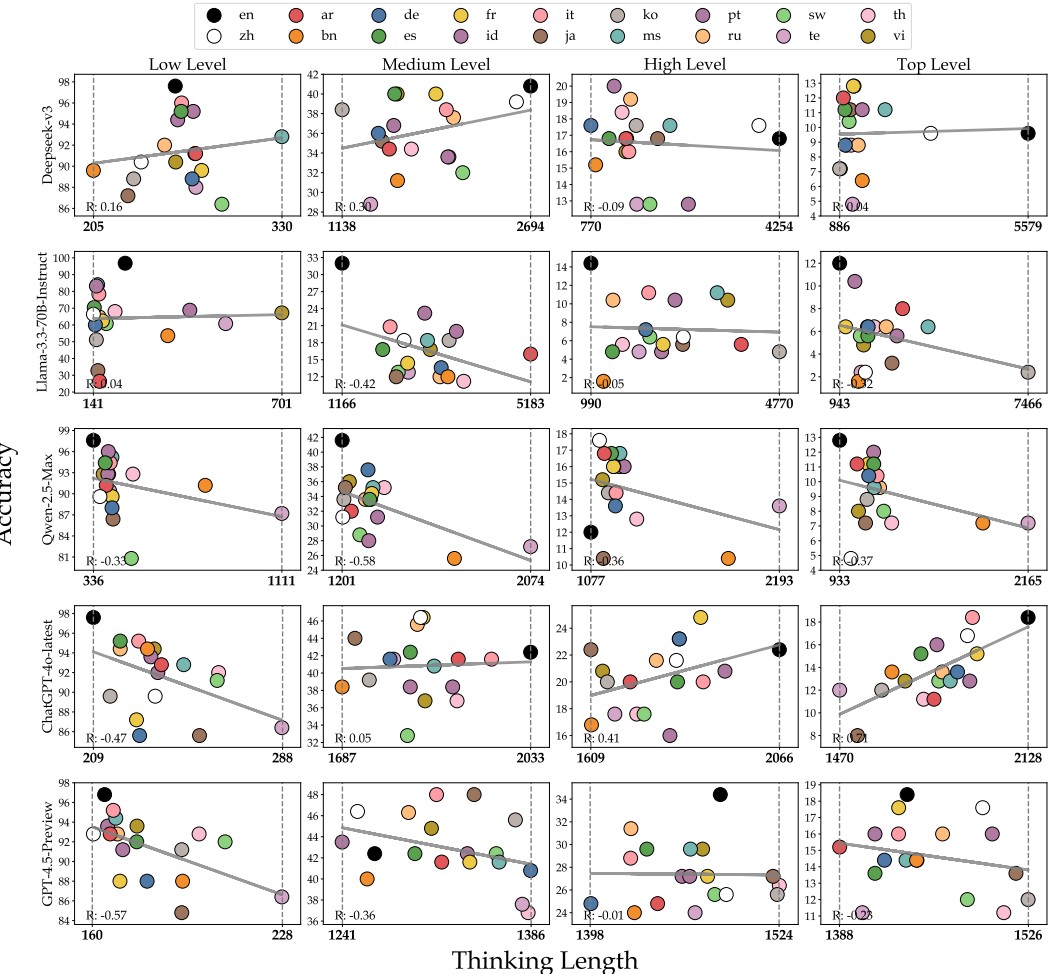

Figure 8: **Thinking lengths** of **Non-Reasoning LLMs** at each language context and level, and its language-level correlation (Pearson correlation coefficient) with reasoning accuracy.

**Overall Thinking Length.** The $x$-axes of Figures 8 and 9 present the thinking lengths across all languages for various non-reasoning and reasoning LLMs at each level. Our findings are as follows:

- The thinking lengths of non-reasoning LLMs across languages tend to stabilize as difficulty level increases. Apart from a few outliers, most models show similar thinking lengths, typically between 1k and 2k tokens in the last three levels.

- In contrast, reasoning LLMs show a continuous increase in thinking length with higher levels, and there are large variations between models. For example, at the top level, OpenAI-o1-mini and o3-mini typically generate 6k-8k tokens, Qwen-QwQ-32B and Deepseek-R1-671B generate 10k-20k tokens, while Claude-3.7-sonnet-thinking reaches as high as 20k–30k tokens, indicating that different reasoning LLMs exhibit distinct slow-thinking patterns.

**Thinking Length Under Different Language Contexts.** When we examine the thinking lengths of different LLMs across various language contexts, we observe distinct patterns between reasoning LLMs and non-reasoning LLMs:

- Although reasoning LLMs tend to have longer absolute thinking lengths, the differences in length between languages are relatively small. At the top level, the maximum thinking lengths for the

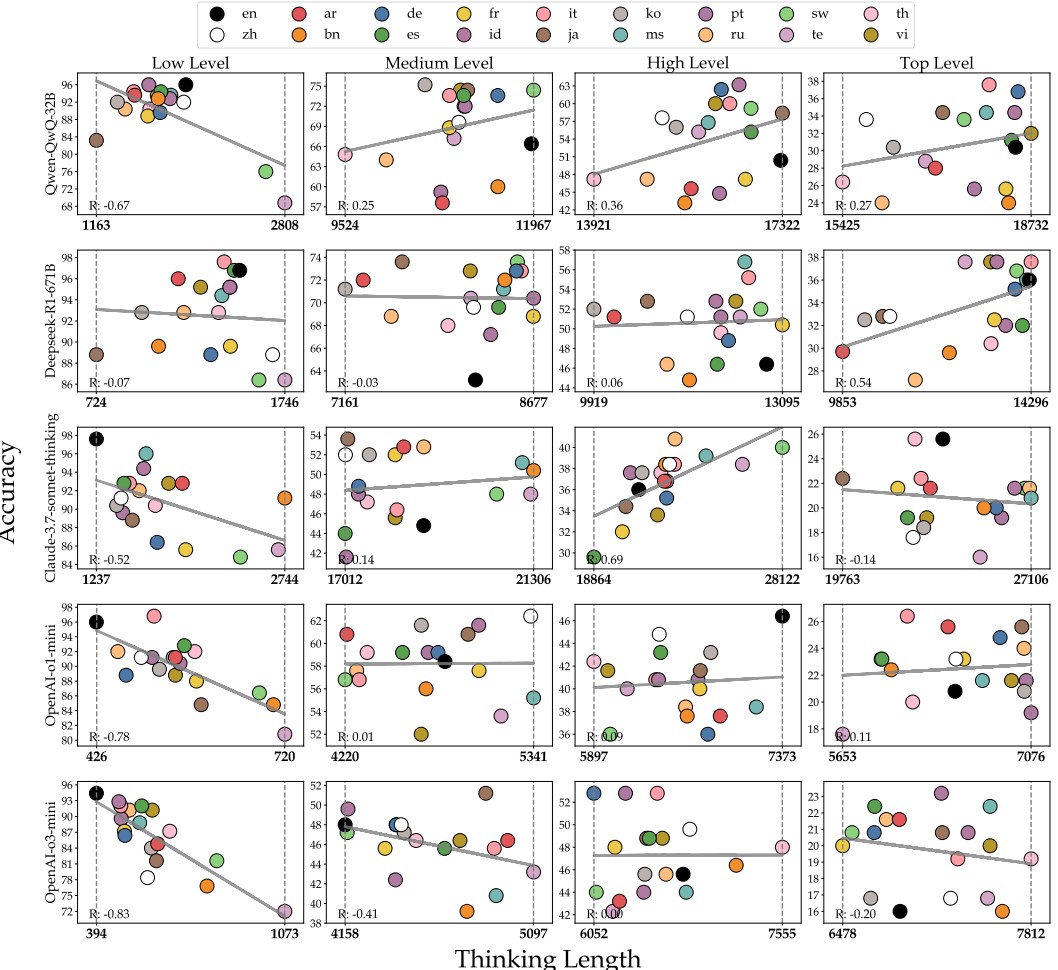

Figure 9: **Thinking lengths** of **Reasoning LLMs** at each language context and level, and its language-level correlation (Pearson correlation coefficient) with reasoning accuracy.

five reasoning LLMs across all languages are only 1.21, 1.45, 1.37, 1.25, and 1.21 times the respective minimums. Considering that the same text can vary slightly in length across different language versions, these differences in thinking length are not particularly significant.

- In contrast, most non-reasoning LLMs exhibit much larger differences in thinking length across different language contexts. For example, at the top level, the maximum thinking lengths for Deepseek-v3, Llama3.3-70B-Instruct, and Qwen-2.5-Max across all languages are 6.30, 7.92, and 2.32 times their respective minimums. More importantly, these extremely long values are almost always dominated by just one or two specific languages: English (en) and Chinese (zh) for Deepseek-v3, Korean (ko) for Llama3.3-70B-Instruct, Telugu (te) and Bengali (bn) for Qwen-2.5-Max, and English (en) for ChatGPT-4o-latest.

These results suggest that ***reasoning LLMs maintain more stable slow-thinking behavior across languages, while non-reasoning LLMs may struggle with cross-lingual behavior alignment***. We speculate two possible reasons: (1) When extremely long values occur in English — such as in Deepseek-v3 and ChatGPT-4o-latest — it is likely that these models have developed preliminary slow-thinking capabilities in English, while such capabilities remain underdeveloped in other languages. (2) When extremely long values appear in low-resource languages — such as in Llama-3.3-70B-Instruct and Qwen-2.5-Max — it may be due to a relative scarcity of training data for those languages, resulting in weaker overall language competence and consequently more complex or verbose outputs.

**Correlation Between Thinking Length and Performance.**  also show the correlation between thinking length and performance across languages.

- At the low level, both reasoning and non-reasoning LLMs tend to exhibit degraded performance with increased thinking length, which aligns with the overthinking conclusions on simple tasks as observed by [7].

- As difficulty level increases, the correlation between thinking length and reasoning performance becomes less evident. In monolingual studies (mainly in English), existing work suggested that longer thinking lengths can improve performance on hard problems [20], indicating a positive correlation. However, across languages, this pattern is inconsistent. Among reasoning LLMs, only Deepseek-R1-671B and Qwen-QwQ-32B show a relatively strong positive correlation. Most others show little or no correlation. For non-reasoning LLMs, only ChatGPT-4o-latest shows a clear positive trend, while models like Llama-3.3-70B-Instruct and Qwen-2.5-Max tend to show negative correlations. These findings suggest that *for the same problem, switching to a different language context that leads to longer thinking lengths does not always result in better reasoning performance*.

# G  Case Study

## G.1  Consistent Input-Output Language

---

**GPT-4.5-Preview   (Output Length: 1110)**

● **Problem: Swahili**    ● **Answer: Swahili**

**Problem**

Ikiwa $f(x) = (x+a)(x+b)$ ambapo $a, b > 0$. Kwa namba halisi yoyote $x_1, x_2, \ldots, x_n \geqslant 0$ zinazotimiza sharti $x_1 + x_2 + \ldots + x_n = 1$, tafuta thamani kubwa zaidi ya $F = \sum_{1 \leqslant i < j \leqslant n} \min\{f(x_i), f(x_j)\}$.

Kumbuka: Tafadhali weka jibu la mwisho katika $\boxed{}$.

**Answer**

### Hatua ya 1: Uchambuzi wa tatizo
Tumepewa:
$$f(x) = (x + a)(x + b), \quad a, b > 0$$

Kwa hiyo, kwa kuwa $a, b > 0$, tunaona kwamba $f(x)$ ni kazi inayoongezeka kwa $x \geqslant 0$. Hii ni kwa sababu, kwa $x \geqslant 0$, tunayo:

$$f(x) = (x + a)(x + b) = x^2 + (a + b)x + ab$$

Kwa kuwa $a, b > 0$, basi $f(x)$ ni kazi inayoongezeka kwa $x \geqslant 0$. (Kwa usahihi zaidi, tuangalie derivative: $f'(x) = 2x + (a + b)$, ambayo ni dhahiri chanya kwa $x \geqslant 0$, hivyo $f(x)$ ni strictly increasing kwa $x \geqslant 0$.

Kwa hiyo, kwa $0 \leqslant x_i \leqslant x_j$, tunapata:
$$\min\{f(x_i), f(x_j)\} = f(x_i)$$

Kwa hivyo, tunaweza kuandika tena tatizo letu kama ifuatavyo:

### Hatua ya 2: Kurahisisha tatizo
Tunataka thamani kubwa zaidi ya:

$$F = \sum_{1 \leqslant i < j \leqslant n} \min\{f(x_i), f(x_j)\}$$

Kwa kuwa $f(x)$ ni strictly increasing kwa $x \geqslant 0$, ili kuongeza thamani ya $F$, tunahitaji kuongeza kadiri iwezekanavyo idadi ya jozi ambazo zina thamani kubwa ya $\min\{f(x_i), f(x_j)\}$. Hii ina maana kwamba, ili kuongeza thamani ya $F$, tunahitaji kuchagua thamani zote za $x_i$ ziwe sawa, yaani:

$$x_1 = x_2 = \cdots = x_n = \frac{1}{n}$$

Hii ni kwa sababu, kama kuna tofauti yoyote kati ya $x_i$, basi baadhi ya jozi zitakuwa na thamani ndogo zaidi ya $\min\{f(x_i), f(x_j)\}$, hivyo kupunguza jumla.

### Hatua ya 3: Kupata thamani kubwa zaidi Kwa hivyo, hali ya juu kabisa inapatikana pale ambapo:

$$x_1 = x_2 = \cdots = x_n = \frac{1}{n}$$

Katika hali hii, tuna jumla ya jozi $\frac{n(n-1)}{2}$, na kila jozi ina thamani sawa ya:

$$\min\{f(\frac{1}{n}), f(\frac{1}{n})\} = f\left(\frac{1}{n}\right)$$

---

Kwa hiyo, thamani kubwa zaidi ya $F$ ni:

$$F_{\max} = \frac{n(n-1)}{2} \cdot f\left(\frac{1}{n}\right)$$

### Hatua ya 4: Kurahisisha zaidi Tuna:

$$f\left(\frac{1}{n}\right) = \left(\frac{1}{n}+a\right)\left(\frac{1}{n}+b\right) = ab + \frac{a+b}{n} + \frac{1}{n^2}$$

Kwa hiyo, thamani kubwa zaidi ya $F$ ni:

$$F_{\max} = \frac{n(n-1)}{2}\left(ab + \frac{a+b}{n} + \frac{1}{n^2}\right)$$

Tunaweza kurahisisha kidogo zaidi:

$$F_{\max} = \frac{n(n-1)}{2}(ab) + \frac{n(n-1)}{2}\cdot\frac{a+b}{n} + \frac{n(n-1)}{2}\cdot\frac{1}{n^2}$$

Hii inakuwa:

$$F_{\max} = \frac{n(n-1)}{2}ab + \frac{(n-1)(a+b)}{2} + \frac{(n-1)}{2n}$$

Kwa hivyo, jibu lililorahisishwa vizuri ni:

$$F_{\max} = \frac{n(n-1)}{2}ab + \frac{(n-1)(a+b)}{2} + \frac{(n-1)}{2n}$$

### Jibu la mwisho:
Kwa hiyo, thamani kubwa zaidi ya $F$ ni:

$$\boxed{\frac{n(n-1)}{2}ab + \frac{(n-1)(a+b)}{2} + \frac{(n-1)}{2n}}$$

**Ground Truth**

$\frac{n-1}{2}\left(\frac{1}{n} + a + b + nab\right)$

---

## ChatGPT-4o-latest  (Output Length: 1517)

● **Problem: Chinese**    ● **Answer: Chinese**

### Problem

设 S 为一个 K3 曲面，C 为一个亏格为 2 的复曲线。此外，设 $\rho$ 是 S 的一个非辛对合，$\psi$ 是 C 的一个对合。$\rho$ 和 $\psi$ 共同定义了一个笛卡尔积 $S \times C$ 上的对合 $\rho \times \psi$。$S \times C$ 被该对合整除得到的商是一个复轨形。设 M 是通过爆破奇异轨迹得到的光滑流形。通过这种构造可以得到的 Hodge 数 $h^{1,1}$ 的最大值是多少？

注意：请将最终答案放在 $\boxed{}$ 中。

### Answer

我们被要求计算通过某种构造方式得到的复三维流形 $M$ 的 Hodge 数 $h^{1,1}$ 的最大值。这个构造是从一个 K3 曲面 $S$ 和一个亏格为 2 的复曲线 $C$ 出发，考虑它们的乘积 $S \times C$，然后对其施加一个对合 $\rho \times \psi$，其中：

- $\rho$ 是 $S$ 上的一个非辛对合（non-symplectic involution），
- $\psi$ 是 $C$ 上的一个对合。

然后我们考虑商空间 $(S \times C)/(\rho \times \psi)$，并对其奇点进行爆破（blow-up）得到一个光滑的复三维流形 $M$。我们要找的是通过这种构造方式可以得到的 $h^{1,1}(M)$ 的最大值。

—

### 第一步：理解构造

我们从 $S \times C$ 出发，维数是 $2 + 1 = 3$，所以我们最终得到的是一个复三维流形。

我们考虑对合 $\rho \times \psi$，它的作用是：

$$(s, c) \mapsto (\rho(s), \psi(c))$$

我们将 $S \times C$ 按照这个对合取商，得到一个有奇点的复三维轨形（orbifold），然后对其奇点进行爆破得到光滑流形 $M$。

我们要计算的是 $h^{1,1}(M)$，即 $M$ 的 Hodge 数之一。

—

### 第二步：K3 曲面上的非辛对合

K3 曲面 $S$ 是一个常见的 Calabi–Yau 二维流形，具有如下 Hodge 数：

$$h^{1,1}(S) = 20, \quad h^{2,0}(S) = 1$$

一个非辛对合 $\rho$ 是指它在 $H^{2,0}(S)$ 上的作用是乘以 $-1$，即它不保留全纯 2-形式。

Nikulin 分类了 K3 曲面上的非辛对合。根据 Nikulin 的结果，一个非辛对合 $\rho$ 的不动点集由若干个曲线组成，其拓扑类型由两个整数 $(r, a)$ 决定，其中：

- $r$ 是 $\rho$ 在 $H^2(S, \mathbb{Z})$ 上的不变子空间的秩，
- $a$ 是某个与不变量有关的整数。

对于非辛对合，$r$ 的取值范围是 $1 \le r \le 20$，而 $h^{1,1}(S) = 20$，所以 $r$ 是 $\rho$ 不变的 $(1,1)$ 类的数量。

—

### 第三步：亏格为 2 的曲线上的对合

设 $C$ 是亏格为 2 的复曲线。我们考虑其上的对合 $\psi$。一个亏格为 2 的曲线是一个紧 Riemann 曲面，其 Hodge 数为：

$$h^{1,0}(C) = 2, \quad h^{0,1}(C) = 2$$

一个对合 $\psi$ 在 $C$ 上的作用可以有不同的不动点数。我们记其不动点数为 $2k$，其中 $0 \le k \le 2$。对合的商 $C/\psi$ 是一个亏格为 $g'$ 的曲线，满足：

$$2g' - 2 = 2(g-1) - \#\text{不动点}$$

设 $g = 2$，则：

$$2g' - 2 = 2(2-1) - 2k = 2 - 2k \Rightarrow g' = 1 - k + 1 = 2 - k$$

所以 $k = 0, 1, 2$，对应 $g' = 2, 1, 0$

我们关心的是 $\psi$ 的不动点数，因为它影响 $\rho \times \psi$ 的不动点集的维数和个数，从而影响爆破后 $h^{1,1}$ 的值。

—

### 第四步：计算 $h^{1,1}(M)$
我们考虑如下公式（见 Borcea–Voisin 构造）：

Borcea–Voisin 构造是从一个 K3 曲面 $S$ 和一个亏格为 $g$ 的曲线 $C$，分别带有非辛对合 $\rho$ 和对合 $\psi$，构造出一个 Calabi–Yau 三维流形 $M$。其 Hodge 数为：

$$h^{1,1}(M) = 11 + 5n + n'$$
$$h^{2,1}(M) = 11 + 5m + m'$$

其中：

- $n$ 是 $\rho$ 的不动点集中的曲线的个数，
- $n'$ 是 $\psi$ 的不动点数，
- $m$ 是 $\rho$ 的不动点集中的曲线的亏格之和，
- $m'$ 是 $\psi$ 的不动点数。

我们只关心 $h^{1,1}(M) = 11 + 5n + n'$

所以我们要最大化 $h^{1,1}(M)$，即最大化 $5n + n'$

—

### 第五步：最大化 $5n + n'$

根据 Nikulin 的分类，K3 曲面上的非辛对合最多可以有：

- $n = 10$ 条不动曲线（包括一个亏格为 10 的曲线和 9 个 $\mathbb{P}^1$），
- 此时 $r = 10$

而对于亏格为 2 的曲线 $C$，对合 $\psi$ 最多可以有 6 个不动点（即 $n' = 6$）

所以最大值是：

$$h^{1,1}(M) = 11 + 5 \cdot 10 + 6 = 11 + 50 + 6 = \boxed{67}$$

—

### 最终答案：

$$\boxed{67}$$

**Ground Truth**

## Deepseek-v3   (Output Length: 1071)

● **Problem: Korean**   ● **Answer: Korean**

### Problem

소수 $p$에 대하여, $\mathbb{F}_p$는 모듈 $p$로 표시하고, $\mathbb{F}_p[x]$는 $\mathbb{F}_p$에서 계수를 가지는 다항식들의 집합을 나타낸다고 하자. 모든 정수 $k$에 대해 $P(\ell) \equiv k \pmod{p}$를 만족하는 어떤 정수 $\ell$가 존재하는 4차 다항식 $P(x) \in \mathbb{F}_p[x]$가 존재하는 모든 $p$를 구하시오. ($\mathbb{F}_p[x]$에는 총 $p^4(p-1)$개의 4차 다항식이 존재함을 주목하시오.)

참고: 최종 답안을 \$\boxed{}\$ 안에 넣어 주세요.

### Answer

주어진 문제는 모든 정수 $k$에 대해 $P(\ell) \equiv k \pmod{p}$를 만족하는 정수 $\ell$가 존재하는 4차 다항식 $P(x) \in \mathbb{F}_p[x]$가 존재하는 모든 소수 $p$를 찾는 것입니다.

### 문제 분석

1. **다항식의 성질**:
- $P(x)$는 4차 다항식이므로, $P(x)$는 최대 4개의 근을 가질 수 있습니다.
- $P(x)$는 $\mathbb{F}_p[x]$에 속하므로, 계수는 $\mathbb{F}_p$에서 취해집니다.

2. **조건 해석**:
- 모든 정수 $k$에 대해 $P(\ell) \equiv k \pmod{p}$를 만족하는 $\ell$이 존재해야 합니다.
- 이는 $P(x)$가 $\mathbb{F}_p$에서 모든 값을 취할 수 있어야 함을 의미합니다. 즉, $P(x)$는 \*\*전사 함수(surjective function)\*\*여야 합니다.

3. **전사 함수 조건**:
- $P(x)$가 전사 함수가 되려면, $P(x)$는 $\mathbb{F}_p$에서 모든 값을 적어도 한 번씩 취해야 합니다. - 4차 다항식이 전사 함수가 되기 위해서는 $p$가 특정 조건을 만족해야 합니다.

### 전사 함수 조건 분석

- $p = 2$:
- $\mathbb{F}_2$는 원소가 $0$과 $1$뿐입니다.
- 4차 다항식 $P(x)$가 모든 값을 취할 수 있는지 확인해야 합니다. - 예를 들어, $P(x) = x^4 + x$는 $P(0) = 0, P(1) = 1 + 1 = 0$이므로 전사 함수가 아닙니다.
- 따라서 $p = 2$는 조건을 만족하지 않습니다.

- $p = 3$:
- $\mathbb{F}_3$는 원소가 $0, 1, 2$입니다.
- 4차 다항식 $P(x)$가 모든 값을 취할 수 있는지 확인해야 합니다.
- 예를 들어, $P(x) = x^4 + x$는 $P(0) = 0, P(1) = 1 + 1 = 2, P(2) = 16 + 2 = 18 \equiv 0 \pmod{3}$이므로 전사 함수가 아닙니다.
- 따라서 $p = 3$는 조건을 만족하지 않습니다.

- $p = 5$:
- $\mathbb{F}_5$는 원소가 $0, 1, 2, 3, 4$입니다.
- 4차 다항식 $P(x)$가 모든 값을 취할 수 있는지 확인해야 합니다.
- 예를 들어, $P(x) = x^4 + x$는 $P(0) = 0, P(1) = 1 + 1 = 2, P(2) = 16 + 2 = 18 \equiv 3 \pmod{5}, P(3) = 81 + 3 = 84 \equiv 4 \pmod{5}, P(4) = 256 + 4 = 260 \equiv 0 \pmod{5}$이므로 전사 함수가 아닙니다.
- 따라서 $p = 5$는 조건을 만족하지 않습니다.

- 일반적인 $p$:
- 4차 다항식 $P(x)$가 전사 함수가 되기 위해서는 $p$가 4차 다항식의 전사성을 보장할 수 있는 특별한 조건을 만족해야 합니다.
- 일반적으로, $p$가 충분히 크면 4차 다항식이 전사 함수가 될 가능성이 높아집니다.
- 그러나 $p$가 작은 경우에는 전사 함수가 되기 어렵습니다.

### 결론

위의 분석을 통해, $p = 2, 3, 5$는 조건을 만족하지 않음을 확인했습니다. 더 큰 소수 $p$에 대해서는 4차 다항식 $P(x)$가 전사 함수가 될 가능성이 있지만, 일반적으로 모든 소수 $p$에 대해 조건을 만족하는지는 명확하지 않습니다.

따라서, 문제의 조건을 만족하는 소수 $p$는 **없습니다**.

$$\boxed{\emptyset}$$

### Ground Truth

$p = 2, 3, 7$

## Qwen-2.5-Max   (Output Length: 1314)

● **Problem: French**   ● **Answer: French**

### Problem

Soit $M(t)$ une fonction mesurable et localement bornée, c'est-à-dire,

$$M(t) \leq C_{a,b}, \quad \forall 0 \leq a \leq t \leq b < \infty,$$

avec une certaine constante $C_{a,b}$, de $[0, \infty)$ à $[0, \infty)$ telle que

$$M(t) \leq 1 + \int_0^t M(t-s)(1+t)^{-1} s^{-1/2} ds, \quad \forall t \geq 0.$$

Trouver le maximum de $M(t)$ lorsque $t \geq 0$.

Remarque: Veuillez mettre la réponse finale dans le $\boxed{}$

## Answer

Nous cherchons à déterminer le maximum de la fonction $M(t)$ sous les hypothèses données. Voici une analyse détaillée du problème.

—

### 1. Hypothèses et inégalité principale La fonction $M(t)$ satisfait l'inégalité suivante pour tout $t \geq 0$ :

$$M(t) \leq 1 + \int_0^t M(t-s) \cdot (1+t)^{-1} \cdot s^{-1/2} ds.$$

De plus, $M(t)$ est localement bornée, c'est-à-dire qu'il existe une constante $C_{a,b}$ telle que $M(t) \leq C_{a,b}$ pour tout $t \in [a, b]$ avec $0 \leq a \leq b < \infty$.

—

### 2. Analyse de l'inégalité
L'inégalité peut être réécrite en introduisant un changement de variable dans l'intégrale. Posons $u = t - s$, ce qui donne $s = t - u$ et $ds = -du$. Lorsque $s$ varie de 0 à $t$, $u$ varie de $t$ à 0. Ainsi, l'intégrale devient :

$$\int_0^t M(t-s) \cdot (1+t)^{-1} \cdot s^{-1/2} ds = \int_t^0 M(u) \cdot (1+t)^{-1} \cdot (t-u)^{-1/2} (-du).$$

En inversant les bornes d'intégration, nous obtenons :

$$\int_0^t M(t-s) \cdot (1+t)^{-1} \cdot s^{-1/2} ds = \int_0^t M(u) \cdot (1+t)^{-1} \cdot (t-u)^{-1/2} du.$$

Ainsi, l'inégalité devient :

$$M(t) \leq 1 + \frac{1}{1+t} \int_0^t M(u) \cdot (t-u)^{-1/2} du.$$

—

### 3. Majoration de $M(t)$
Pour simplifier l'analyse, supposons que $M(t)$ atteigne son maximum en un certain point $t_0 \geq 0$. Alors, pour tout $t \geq 0$, on a $M(t) \leq M(t_0)$. En substituant cette majoration dans l'inégalité, nous obtenons :

$$M(t_0) \leq 1 + \frac{1}{1+t_0} \int_0^{t_0} M(t_0) \cdot (t_0 - u)^{-1/2} du.$$

Factorisons $M(t_0)$ hors de l'intégrale :

$$M(t_0) \leq 1 + \frac{M(t_0)}{1+t_0} \int_0^{t_0} (t_0 - u)^{-1/2} du.$$

Calculons l'intégrale :

$$\int_0^{t_0} (t_0 - u)^{-1/2} du = \int_0^{t_0} v^{-1/2} dv \quad \text{(en posant } v = t_0 - u\text{)}.$$

Cette intégrale est standard et vaut :

$$\int_0^{t_0} v^{-1/2} dv = 2\sqrt{t_0}.$$

Ainsi, l'inégalité devient :

$$M(t_0) \leq 1 + \frac{M(t_0)}{1+t_0} \cdot 2\sqrt{t_0}.$$

—

### 4. Résolution de l'inégalité
Réorganisons l'inégalité pour isoler $M(t_0)$ :

$$M(t_0) \left( 1 - \frac{2\sqrt{t_0}}{1+t_0} \right) \leq 1.$$

Posons :

$$C(t_0) = 1 - \frac{2\sqrt{t_0}}{1+t_0}.$$

Alors :

$$M(t_0) \leq \frac{1}{C(t_0)}.$$

—

### 5. Comportement de $C(t_0)$
Étudions $C(t_0)$ pour comprendre son comportement :

$$C(t_0) = 1 - \frac{2\sqrt{t_0}}{1+t_0}.$$

- Lorsque $t_0 = 0$, $C(0) = 1$.
- Lorsque $t_0 \to \infty$, $\frac{2\sqrt{t_0}}{1+t_0} \to 0$, donc $C(t_0) \to 1$.

Ainsi, $C(t_0) > 0$ pour tout $t_0 \geq 0$, et $C(t_0)$ reste proche de 1. Par conséquent, $M(t_0)$ est borné.

—

### 6. Conclusion
Le maximum de $M(t)$ est atteint lorsque $t_0 = 0$, où $M(0) \leq 1$. En vérifiant l'inégalité pour $t = 0$, on trouve que $M(0) = 1$ est une solution valide.

$$\boxed{1}$$

**Ground Truth**

$10 + 2\sqrt{5}$

**Qwen-QwQ-32B   (Output Length: 7037)**

● **Problem: Japanese**    ● **Thinking: Japanese**    ● **Answer: Japanese**

**Problem**

ターボというカタツムリが、2024行と2023列のボード上でゲームをします。2022個のセルには隠れたモンスターが存在します。ターボは最初、モンスターの位置を知りませんが、各行には第1行目と最終行を除いて正確に1つのモンスターが存在し、各列には最大1つのモンスターが含まれることを知っています。

ターボは第1行目から最終行まで行こうとする一連の試みを行います。各試みで、彼は第1行目の任意のセルから始め、続いて隣接するセルに側を共有する形で移動し続けます（以前に訪れたセルに戻ることもできます）。もしモンスターがいるセルに到達した場合、その試みは終了し、彼は新しい試みを開始するために第1行目に戻されます。モンスターは動かず、ターボは訪れた各セルにモンスターがいるかどうかを記憶します。もし彼が最終行の任意のセルに到達した場合、その試みは終了し、ゲームも終了します。

ターボがどのようなモンスターの位置の場合でも、$n$回目以内に最終行に到達することを保証する戦略を持てるような、$n$の最小値を求めなさい。

注意：最終的な答えを $\boxed{}$ に入れてください。

**Thinking**

まず、問題を整理してみます。ターボというカタツムリが、2024行と2023列のボード上でゲームをします。このボードには2022個のモンスターが隠れています。モンスターの配置の条件は、各行（1行目と最終行を除く）にちょうど1つずつ、各列には最大1つしかモンスターが存在しないということです。ターボは、1行目の任意のセルから始めて、隣接するセル（上下左右）を移動し続けます。モンスターのいるセルに到達すると、その試みは終了し、ターボは1行目に戻されます。また、ターボは訪れたセルにモンスターがいるかどうかを記憶しています。最終行に到達するとゲームが終了します。

目標は、どのようなモンスターの配置に対しても、n回目以内に最終行に到達する戦略を持てるようなnの最小値を求めることです。

まず、モンスターの配置条件を確認します。各行（1行目と2024行目を除く2022行）にちょうど1つずつ、各列には最大1つ。つまり、各列にはモンスターが0個または1個で、全体で2022個あるので、2023列のうち2022列には1個ずつ、残りの1列にはモンスターがないということになります。

ターボの戦略は、モンスターの配置を考慮しつつ、最終行に到達するまでに必要な試行回数の最小値を求めることです。つまり、最悪の場合の試行回数の最小値を求める必要があります。

まず、ターボが最終行に到達するためには、モンスターがその経路上に存在しないか、あるいはモンスターに到達する前に最終行に到達する必要があります。ただし、モンスターが経路上にある場合、ターボはそのセルで試みが終了し、1行目に戻されます。

モンスターの配置は各行に1つずつ、列ごとに最大1つなので、各列には0または1つのモンスターがあります。つまり、モンスターの配置は、各行に1つずつ配置され、列が重複しないように配置されていると考えられます。つまり、モンスターの位置は、各行と各列のペアのうち、2022列分が1行ずつに配置されていると考えられます。

ここで、ターボが最終行に到達するためには、モンスターがその経路を遮断していない必要があります。しかし、モンスターの配置は未知なので、ターボは最悪の場合を想定して、モンスターが経路をすべて遮断する可能性がある場合に対処する必要があります。

まず、各試みでターボが取る経路を選ぶ必要があります。この経路を選ぶ戦略によって、必要な試行回数が異なります。最適な戦略を見つけ、そのときの最悪ケースの試行回数を求めます。

まず、ターボが最終行に到達するためには、最終行のどのセルでも構わないため、最終行のどのセルに到達するかを自由に選べます。ただし、モンスターがその経路に存在する場合、途中で試みが終了します。

ターボの戦略は、各試みで異なる経路を試すことで、モンスターの位置を特定し、最終行に到達する経路を見つけ出す必要があります。ただし、モンスターの位置を記憶できるので、前回の試みでモンスターが存在したセルは避けることができます。

まず、モンスターの配置は各行に1つずつ、各列に最大1つなので、モンスターの位置は各行と各列のペアで、列が重複しないように配置されています。つまり、モンスターの位置は、各行の列番号がすべて異なるように配置されています。つまり、各行のモンスターの列は、2023列から選ばれ、2022列が選ばれています（1列は未使用）。この未使用の列は、どの行にもモンスターがない列です。

ターボが最終行に到達するためには、最終行のどの列でも到達可能であれば良いので、最終行にモンスターがない列を選べば良いかもしれません。ただし、モンスターの配置では、最終行（2024行目）にはモンスターは存在しないので、最終行自体はモンスターの配置対象外です。したがって、最終行のどのセルにもモンスターはないため、最終行のどのセルに到達する経路を選ぶかが問題です。

ただし、各列にモンスターが存在するかどうかは、各行のモンスターがその列にあるか否かで決まります。つまり、ある列がモンスターの配置に含まれていない場合（すなわち、どの行にもその列のモンスターがない場合）、その列の最終行のセルは、モンスターが存在しないので、その列を選ぶと最終行に到達できる可能性があります。

しかし、モンスターの配置では、2022列がモンスターの配置に使われ、残り1列は使われていないので、その1列を選べば最終行に到達でききます。ただし、その1列がどこかは不明です。

したがって、ターボの戦略は、その1列を見つけるまでに必要な試行回数を最小化する必要があります。

では、ターボがどのようにその1列を見つけるか？

各試みで、ターボは1行目から始めて、最終行を目指す経路を選ぶ必要があります。もし、その経路がモンスターの列を避けていれば、最終行に到達できます。しかし、モンスターの列を避けるには、モンスターの位置を知る必要があります。

しかし、ターボはモンスターの位置を知らずに、各試みで経路を選ぶ必要があります。ただし、前回の試みでモンスターに当たったセルを記憶できるので、そのセルを避けることができます。

ここで、モンスターが配置されている列は2022列、未使用の列は1列です。ターボが最終行の未使用の列に到達するためには、その列を選ぶ必要があります。しかし、その列がどこかは不明です。

各試みで、ターボは経路を選ぶ際、未使用の列を避けるか、それとも試す必要があります。ただし、未使用の列は1つしかないので、それを試すまでに、他の列を試す必要があります。

ただし、各試みで、ターボが経路を選ぶ方法によって、モンスターに当たる可能性があります。例えば、ターボが1列目から最終行を目指すと、その列にモンスターがあれば、そのモンスターの行で試みが終了します。もし、その列が未使用であれば、最終行に到達できます。

したがって、ターボが各列を順番に試す戦略をとれば、1列ずつ試すことで、最終的に未使用の列を見つけることができます。ただし、最悪の場合、未使用の列が最後に試される場合、2022回の試みが必要になるかもしれません。

しかし、モンスターの配置では、各列にモンスターが存在するか否かは、2022列には存在し、1列だけ存在しないので、各列を試すたびに、モンスターが存在する列を試すと、その試みは途中で終了します。未使用の列を試すと、試みは成功してゲーム終了です。

したがって、最悪の場合、ターボは2022回の失敗（モンスターのある列を試した）を経て、2023回目に未使用の列を試し、成功するということになります。しかし、2023列のうち2022列がモンスターの列なので、未使用の列は1列だけ存在します。したがって、各試みで異なる列を選ぶ場合、2022回の試みでモンスターのある列を試し、2023回目に未使用の列を選ぶ必要があります。しかし、ターボは各試みで同じ列を試す必要はありません。例えば、各試みで異なる列を選ぶようにすれば、2023回目までに未使用の列にたどり着きます。しかし、2023列のうち、モンスターが存在する列は2022列あるので、そのうちの1列だけ未使用。したがって、2022回の試みでモンスターのある列を試し、2023回目に未使用の列を選ぶ必要があります。

しかし、ターボは各試みで経路を選ぶ際に、1行目のどこから始めるか自由に選べます。つまり、各試みで異なる列を選ぶことで、未使用の列を見つけることができます。したがって、最悪の場合、2022回の失敗（モンスターのある列を試す）を経て、2023回目に未使用の列を選ぶと成功します。したがって、必要な試行回数は2023回。

しかし、この戦略では、各試みで1列ずつ試すので、2023回目が必要ですが、問題は各試みで経路を選ぶ際に、モンスターのある列を試すと、その試みは途中で終了します。つまり、各試みで、ターボが選んだ列がモンスターの列であれば、その列のモンスターの行で試みが終了します。しかし、ターボはその列のモンスターの位置を知らないので、その列を選ぶと、必ずその列のモンスターの行で試みが終了します。つまり、各試みで選んだ列がモンスターの列であれば、その試みは失敗し、次に別の列を選ぶ必要があります。

したがって、ターボが各試みで異なる列を選ぶ場合、最悪の場合、2022回の失敗を経て、2023回目に未使用の列を選ぶ必要があります。したがって、必要なnは2023。

しかし、問題は、各試みでターボがどのように経路を選ぶかによって、より効率的な戦略があるかもしれません。

例えば、各試みで、異なる列を選ぶのではなく、各列を複数回試すことで、モンスターの位置を特定する方法があるかもしれません。

ただし、ターボは訪れたセルにモンスターがいるかどうかを記憶するので、一度モンスターに当たったセルは避けることができます。

ここで、モンスターの配置は各行に1つずつ、各列に最大1つなので、モンスターの位置は各行の列がすべて異なる。つまり、モンスターの配置は、各行に対応する列がすべて異なるので、これは各行の列を示すような関数と考えることができます。つまり、各行の列番号はすべて異なるので、それは各行の列を1つずつ選んだ列の集合で、各列は1回しか使われていない。

したがって、モンスターの配置は、各行に対応する列が全て異なるので、各列のモンスターがあるかどうかは、その列が選ばれているかどうかです。

つまり、未使用の列は1列だけ存在し、他の列は各1行に1つずつモンスターが配置されています。

ターボの戦略としては、各列を試すことで、モンスターの有無を確認し、未使用の列を見つける必要があります。

ただし、各試みでターボが選んだ列がモンスターのある列の場合、その列のモンスターの行で試みが終了します。その際、ターボはそのセル（列、行）にモンスターが存在することを知るので、そのセルを避けることができます。

しかし、モンスターの配置は各行に1つずつあるので、同じ列を選ぶと、その列のモンスターが存在する行を知ることができます。つまり、列lを選ぶと、モンスターが存在する行kを知ることができます。次回以降、列lを選ぶ際は、その行kを避けることで、モンスターを避けて進むことができます。

ただし、ターボは各試みで、1行目から出発し、隣接するセルを移動する必要があります。つまり、経路を選ぶ際、モンスターの行を避けるために、その列のモンスターの行を迂回する必要があります。

しかし、モンスターの行が分かった場合、その列のモンスターの行を迂回する経路を選ぶことで、その列の最終行に到達できるかもしれません。例えば、モンスターが行kの列lにある場合、ターボは1行目から出発し、列lの行kを避けて進み、最終行まで行けるか？

しかし、列lにはモンスターが存在するので、その列のモンスターの行以外のセルを通過することは可能でしょうか？

モンスターが存在するセル（k,l）に到達すると、試みが終了するので、そのセルを避けて進む必要があります。

例えば、列lを選ぶ場合、ターボは1行目から出発し、列lの行kの上側を通って進むか、下側を通って進む必要があります。ただし、モンスターの行kがその経路の途中にあるかどうかによって、到達できるかどうかが決まります。

しかし、モンスターの行kが最終行（2024行目ではない）なので、列lの最終行のセル（2024,l）はモンスターが存在しないので、到達可能かもしれません。

ただし、ターボが列lのモンスターの行kを迂回できるかどうかは、経路の選択次第です。

例えば、列lのモンスターが行kにある場合、ターボは1行目から出発し、列lの上側を進み、行kの上を通過し、モンスターの行を避けて列lの下側を通って最終行に到達できるかもしれません。ただし、隣接するセルしか移動できないので、行kの上側を通る場合、行k-1の列lのセルを通過する必要があります。しかし、そのセルはモンスターではないので、通過可能です。同様に、行k+1以降を進むこともできます。

つまり、モンスターの行kを迂回して最終行に到達できる可能性があります。しかし、モンスターの行kを知っている場合、迂回経路を取ることで、その列lの最終行に到達できるかもしれません。

しかし、モンスターの行kが不明の場合は、その列lを選ぶと、モンスターの行kに到達し、試みが終了します。したがって、列lを選ぶと、モンスターの行kを知ることができます。その後、次回以降、列lを選ぶ際は、その行kを避けて迂回する経路を選ぶことで、最終行に到達できるかもしれません。

しかし、ターボは、各試みで経路を選ぶ自由があるので、モンスターの位置を知った後は、その列を迂回して最終行に到達できる可能性があります。しかし、この場合、最初の試みで列lを選ぶと、モンスターの行kを知り、次回以降はその列を選ぶ際に、迂回して最終行に到達できるかもしれません。したがって、列lを試す最初の試みは失敗しますが、その後の試みでは成功するかもしれません。

しかし、問題は、各試みでは、ターボが最終行に到達できればゲーム終了なので、ターボがモンスターのある列lを試した後、その列lを迂回して最終行に到達できるなら、その列lで成功する可能性があります。

しかし、モンスターのある列lを選ぶ場合、最初の試みでは、モンスターの行kに到達し、失敗します。しかし、その後、同じ列lを選ぶ際に、モンスターの行kを迂回して最終行を目指せば、成功するかもしれません。

しかし、ターボは各試みで、1行目から出発しなければならないので、列lを選ぶ際には、1行目から出発し、列lのモンスターの行kを避けて進み、最終行に到達できる経路を選ぶ必要があります。

つまり、列lを選ぶ場合、最初の試みでは失敗するが、2回目以降は迂回可能かもしれません。

この場合、各列lのモンスターの位置を特定するためには、最初の試みで列lを選ぶことで、その列のモンスターの行kを特定でき、次回以降はその列lを選んで迂回経路を選ぶことで、最終行に到達できるかもしれません。

しかし、この場合、各列lに対して、2回の試みが必要になるかもしれません。最初の試みで失敗し、2回目で成功する場合、各列lについて2回の試みが必要です。しかし、未使用の列は1つなので、未使用の列を選ぶと最初から成功します。

しかし、この戦略では、各列l（モンスターがある2022列）に対して、2回の試みが必要となり、未使用の列は1回で成功するので、総試行回数は2022×2 +1 =4045回が必要になるかもしれません。

しかし、これは最悪の場合の試行回数であり、より効率的な戦略があるかもしれません。

あるいは、モンスターのある列を選ぶと、その列のモンスターの位置を特定できるので、次回以降、その列を迂回して最終行に到達できる場合、各列lについて、最初の試みで失敗し、2回目で成功するので、各列lで2回の試行が必要ですが、未使用の列は1回で成功します。

しかし、未使用の列を選ぶと最初から成功するので、その列を選ぶと、1回で成功します。

したがって、最悪の場合、ターボはモンスターのある列をすべて試すまでに、各列に対して2回の試行が必要で、未使用の列を最後に試す場合、2022×2 +1 =4045回が必要になります。

しかし、この計算は、各列を2回試す必要があると仮定しているが、実際には、一度試した列のモンスターの位置を知っているので、2回目以降はその列を迂回して最終行に到達できるかもしれません。したがって、各モンスターのある列に対して、2回の試行が必要になるかもしれません。

しかし、このような戦略では、試行回数が多くなるので、より効率的な戦略を考える必要があります。

別のアプローチとして、ターボが各列を試すのではなく、各列を試す際に、モンスターの位置を特定しながら、最終行に到達する経路を見つける方法を考える。

例えば、ターボが列lを選ぶ際、1行目から出発し、列lの最終行を目指す経路を選ぶ。この場合、モンスターが存在する行kが経路上にあるかどうか。もし、存在する場合、その行で試みが終了し、モンスターの行kを特定できる。次回以降、列lを選ぶ際には、その行kを迂回する経路を選ぶことで、最終行に到達できるかもしれない。

ただし、迂回経路を選ぶには、経路の選択が可能かどうかが問題です。

例えば、列lのモンスターが行kにある場合、ターボは行kの上側を通って進むか、下側を通って進む必要があります。ただし、行kが最終行の直前であれば、迂回ができないかもしれません。例えば、行2023の列lにモンスターがある場合、その列の最終行は2024行目なので、モンスターの行2023を迂回する必要はありません。1行目から出発し、列lを進み、行2023でモンスターに当たる前に、最終行2024に到達できます。ただし、行2023の列lのセルにモンスターがあるので、行2023まで進むと試みが終了します。したがって、列lを選ぶ場合、モンスターが最終行の1つ手前にある場合、最終行に到達できないかもしれません。

つまり、モンスターが行kにある場合、最終行に到達するためには、行kより下の行（最終行側）を進む必要がありますが、モンスターが行kにあるため、その行で試みが終了します。したがって、列lのモンスターの行kが最終行の直前（2023行目）にある場合、列lを選ぶと、行2023で試みが終了し、最終行には到達できません。この場合、列lを選ぶと、モンスターの行kを知り、次回以降、列lを選ぶ際に、モンスターの行kを避ける必要があるが、その列lの最終行に到達するには、モンスターの行kを迂回する必要がありますが、最終行はモンスターの行kの下にあるので、迂回できないかもしれません。

したがって、モンスターが列lの行kにある場合、最終行に到達するためには、モンスターの行kより下の行を進む必要がありますが、その行kで試みが終了するため、不可能です。したがって、列lを選ぶと、モンスターの行kを知り、その列lは最終行に到達できないことが判明します。その場合、列lは避けて、他の列を選ぶ必要があります。

したがって、列lを選ぶと、モンスターの行kを知り、その列lが未使用の列ではないことが分かり、次回以降は他の列を選ぶ必要があります。未使用の列を選ぶと、その列にはモンスターがないため、最終行に到達できます。

したがって、各列lを選ぶと、モンスターが存在するかどうかを知ることができます。モンスターがある場合、その列は最終行に到達できないため、次回以降は別の列を選ぶ必要があります。未使用の列を選ぶと、成功です。

したがって、ターボの戦略は、各列を1つずつ試すことで、モンスターの有無を確認し、未使用の列を見つけるまでに必要な試行回数を最小化する必要があります。

最悪の場合、未使用の列が最後に試される場合、2022回の失敗（モンスターのある列を試す）を経て、2023回目に未使用の列を選ぶと成功します。したがって、必要な試行回数は2023回。

しかし、ターボが各列を順番に試す場合、各試みで異なる列を選ぶことができるので、2023回目の試みで未使用の列を選ぶことができます。ただし、各試みで、ターボはその列を選ぶと、モンスターがあるかどうかを知るので、モンスターがある列を選ぶと試みは失敗しますが、その列を記録して、次回以降選ばないようにすれば、2023回目で未使用の列を選ぶことができます。

したがって、最悪の場合、必要な試行回数は2023回。

しかし、問題は、ターボが各試みで、モンスターのある列を選ぶと、その試みは失敗しますが、その列がモンスターがあることが分かり、次回以降はその列を避けることができます。したがって、各列を1回ずつ試すことで、未使用の列を見つけるため、最悪の場合、2023回目の試みが必要です。したがって、nの最小値は2023。

しかし、問題の条件に注意が必要です。各行（1行目と最終行を除く）に正確に1つのモンスターがあるということです。つまり、2024行のうち、1行目と2024行目を除く2022行に各1つずつ、合計2022個のモンスターがある。各列には最大1つなので、各列には0または1つ。全体で2022個なので、2023列のうち2022列が1つずつ、残り1列が0個。

したがって、未使用の列は1列だけ存在し、その列を選ぶと、モンスターが存在しないため、最終行に到達できます。

したがって、ターボが各列を順番に試す場合、最悪の場合、2022回の失敗を経て、2023回目に未使用の列を選ぶと成功します。したがって、必要なnは2023。

しかし、ここで、ターボが各試みで、列を選ぶ際、同じ列を複数回選ぶ可能性があるが、効率的な戦略では、各列を1回ずつ試すことで、未使用の列を見つけることができます。したがって、最適な戦略では、2023回目の試みで未使用の列を選ぶことができ、これが最悪の場合の試行回数になります。

しかし、問題文の注意事項で、最終行に到達した場合、ゲームが終了するので、ターボが未使用の列を選ぶと、その試みで成功します。したがって、未使用の列を選ぶ試みは成功し、ゲーム終了です。

したがって、最悪の場合、2023回の試みが必要です。したがって、nの最小値は2023。

しかし、ここで、ターボが各試みで、1列を選ぶのではなく、複数列を試す戦略があるかもしれません。例えば、各試みで、複数の列を通過する経路を選ぶことで、複数の列のモンスターの位置を特定できるかもしれません。しかし、モンスターが存在する列を通過すると、試みは終了するので、複数列を試すことはできません。つまり、各試みでは、最初に当たるモンスターの位置で終了します。

したがって、各試みで、1列を選ぶか、複数列を選ぶ経路を選ぶが、最初のモンスターに当たるとそこで終了するので、複数列を選ぶことは効果的ではないかもしれません。

したがって、各試みで、1列を選ぶことが最適で、各列を1回ずつ試す戦略が最適です。

したがって、nの最小値は2023。

しかし、問題の条件で、2024行と2023列のボードで、モンスターの数が2022個で、各行（1行目と最終行を除く）に1つずつあるので、各列に0または1つのモンスター。各列のモンスターの数の合計が2022個なので、2023列のうち、2022列が1つ、1列が0個。

したがって、未使用の列は1列だけ存在します。ターボがその列を選ぶと成功です。

最悪の場合、ターボがその未使用の列を最後に試すので、2022回の失敗を経て、2023回目に成功します。したがって、答えは2023。

しかし、ここで、ターボが各試みで、列を選ぶ際に、未使用の列を選ぶ可能性があるが、モンスターの配置が最悪の場合、未使用の列が最後に試される可能性があるため、2023回目が必要。

したがって、nの最小値は2023。

しかし、この問題の答えが2023かどうか、あるいは別の値かを確認する必要があります。

例えば、各列を選ぶ際、モンスターが存在する列を選ぶと、その試みは失敗しますが、その列のモンスターの位置を知ることができます。その後、他の列を選ぶことで、未使用の列を見つけることができます。しかし、未使用の列を選ぶと、その試みで成功します。

しかし、モンスターの配置が最悪の場合、ターボが未使用の列を選ぶまでに、2022回の失敗が必要になります。

したがって、最悪の場合の試行回数は2023回。

しかし、問題の条件では、ターボは「各行には第1行目と最終行を除いて正確に1つのモンスターが存在し、各列には最大1つのモンスターが含まれる」という条件を知っています。したがって、ターボは、各列がモンスターを含むか否かを、試行を繰り返すことで記録し、未使用の列を見つけることができます。したがって、各列を1回ずつ試すことで、未使用の列を見つけるため、2023回が必要です。

しかし、ここで、ターボが各試みで、異なる列を選ぶ必要があるが、最初の2022回の試みで、モンスターのある列をすべて試すことができ、2023回目に未使用の列を選ぶことができます。したがって、2023回目が最悪の場合の試行回数です。

したがって、答えは2023。

しかし、問題の解答例を見ると、2023ではなく、2023-1=2022になる可能性もあります。なぜなら、各列を試す際に、2022列のうち、2022回で全てのモンスターのある列を試すと、残りの1列が未使用であることが確定するため、2022回で未使用の列が特定でき、2023回目は必要ないかもしれません。

例えば、ターボが2022回の試みで、モンスターのある列を全て試すと、未使用の列が残る1列であることが確定します。したがって、2023回目にその列を選ぶと成功します。したがって、2023回目が必要です。したがって、2023が正解です。

しかし、問題の解答が2023ではなく、2022+1=2023なので、答えは2023です。

しかし、もう少し考えます。ターボが各試みで、1列を選ぶ場合、各試みで1列を選ぶので、2022列のモンスターのある列を全て試すには、2022回の試みが必要です。このとき、未使用の列は残り1列です。したがって、2023回目はその未使用の列を選ぶことで成功します。したがって、2023回が必要です。

したがって、答えは2023。

しかし、問題文では、ターボは「最終行の任意のセルに到達した場合、ゲーム終了」とあります。したがって、未使用の列を選ぶと、その試みで成功します。したがって、最悪の場合、2022回の失敗（モンスターのある列を選ぶ）を経て、2023回目に未使用の列を選ぶことで成功します。

したがって、答えは2023。

しかし、ここで、各試みで、ターボが異なる列を選ぶ場合、2022回の試みで、モンスターのある列を全て試すことができます。したがって、2023回目に未使用の列を選ぶことができます。したがって、nの最小値は2023です。

したがって、答えは 2023 です。

しかし、ここで、ターボが各試みで、同じ列を選ぶ必要がないため、2022列のモンスターのある列を全て試すには、2022回の試みが必要です。そして、2023回目に未使用の列を選ぶ必要があります。したがって、最悪の場合、2023回が必要です。

しかし、もう一つの考え方として、モンスターの配置は各行の列がすべて異なるので、各列のモンスターの行はすべて異なる。したがって、未使用の列を除く各列のモンスターの行は、各行に対応しています。したがって、ターボが最終行に到達するためには、未使用の列を選ぶ必要があります。

各試みで、ターボが選んだ列がモンスターの列であれば、その試みは途中で終了します。未使用の列を選ぶと成功します。したがって、最悪の場合、未使用の列が最後に試されるので、2023回目が必要です。

したがって、答えは2023です。

しかし、ここで、ターボがモンスターのある列を選ぶと、その列のモンスターの行を知るので、次回以降、その列を迂回して最終行に到達できる可能性があるかもしれません。例えば、列1を選ぶと、モンスターが行kにあることが分かり、次回以降、列1を選ぶ際に、行kを迂回して最終行に到達できるかもしれません。しかし、モンスターが列1の行kにある場合、列1の最終行にはモンスターはないので、列1を選ぶ場合、モンスターの行kを迂回して最終行に到達できるでしょうか？

例えば、モンスターが行kにある場合、ターボは1行目から出発し、列1を進みますが、行kで試みが終了します。しかし、次回、列1を選ぶ際、モンスターの行kを避ける経路を選ぶことができます。例えば、列1の上側を通らず、他の列を通って最終行を目指す経路を選ぶかもしれませんが、ターボは列1を選ぶので、列1を通らなければならないかもしれません。

ただし、ターボは、1行目から出発し、隣接するセルを移動し続ける必要があります。したがって、列1を選ぶ場合、1行目の列1のセルから出発し、列1のセルを進む必要があります。したがって、モンスターの行kを避けることはできません。つまり、列1を選ぶと、必ずモンスターの行kに到達し、試みが終了します。

したがって、モンスターのある列1を選ぶ場合、どの経路をとっても、モンスターの行kに到達するため、試みは失敗します。したがって、モンスターのある列1を選ぶことは、失敗を意味し、次回以降はその列1を避ける必要があります。

したがって、各モンスターのある列1を選ぶと、失敗し、その列1を記録します。未使用の列を選ぶと成功します。したがって、最悪の場合、2023回目の試みが必要です。

したがって、答えは2023です。

しかし、もう一度考えると、ターボが各試みで、異なる列を選ぶ場合、2022回の試みで、モンスターのある列を全て試すことができます。この時、未使用の列が残るので、2023回目にその列を選ぶことで成功します。したがって、nの最小値は2023です。

したがって、答えは 2023 です。

**Answer**

ターボが2024行2023列のボード上で最終行に到達するためには、モンスターが配置されていない1列を見つける必要があります。モンスターの配置条件から、各行（1行目と最終行を除く2022行）に1つずつ、各列には最大1つのモンスターがあり、全体で2022個のモンスターがあるため、2023列のうち1列だけはモンスターが存在しません。

ターボが各試みで異なる列を選ぶ場合、最悪の場合、モンスターのある2022列を全て試す必要があります。この場合、2022回の試みで全てのモンスターのある列を試した後、2023回目にモンスターがない未使用の列を選ぶことで、最終行に到達できます。したがって、必要な試行回数の最小値は2023です。

2023

**Ground Truth**

3

## G.2 Inconsistent Input-Output Language

**Qwen-QwQ-32B  (Output Length: 2996)**

● **Problem: Japanese**  ● **Thinking: Chinese**  ● **Answer: Japanese**

**Problem**

無限次元の $\mathbb{C}$-ベクトル空間 $M = \bigoplus_{i \in \mathbb{Z}} \mathbb{C}e_i$ を考える。$\text{End}(M)$ を $M$ の $\mathbb{C}$-線形自己準同型からなる $\mathbb{C}$-代数とする。$A$ と $B$ を $\text{End}(M)$ の2つの可換な元とし、次の条件を満たすとする：整数 $m \le n < 0 < p \le q$ が存在し、$\gcd(-m, p) = \gcd(-n, q) = 1$ を満たし、すべての $j \in \mathbb{Z}$ に対して

$$Ae_j = \sum_{i=j+m}^{j+n} a_{i,j}e_i, \quad \text{with } a_{i,j} \in \mathbb{C}, a_{j+m,j}a_{j+n,j} \neq 0,$$

$$Be_j = \sum_{i=j+p}^{j+q} b_{i,j}e_i, \quad \text{with } b_{i,j} \in \mathbb{C}, b_{j+p,j}b_{j+q,j} \neq 0.$$

が成り立つ。$R \subset \text{End}(M)$ を $A$ と $B$ で生成される $\mathbb{C}$-部分代数とする。$R$ が可換環であり、$M$ は $R$ の加群とみなすことができることに注意してください。$R$ の分数体を $K$ とし、$M \otimes_R K$ は $K$ 上の $h$ 次元ベクトル空間であるとする。$h$ の値は何か？

注意：最終的な答えを $\boxed{}$ に入れてください。

## Thinking

好，我现在要解决这个问题。首先，我需要仔细理解题目中的各个条件，然后一步步分析，找出答案 $h$ 的值。题目涉及到无限维的向量空间 $M$，由整数下标的基础向量 $e_j$ 组成，直和结构。然后考虑 $\text{End}(M)$ 中的两个可交换的元素 $A$ 和 $B$，它们生成一个交换子代数 $R$，然后考虑 $M$ 作为 $R$ 模的张量积与分数体 $K$，求其维度 $h$。

首先，题目给出 $A$ 和 $B$ 的具体形式。对于每个 $j$，$Ae_j$ 是从 $i = j+m$ 到 $j+n$ 的和，其中 $m \leq n < 0 < p \leq q$，并且 $\gcd(-m,p) = 1$，$\gcd(-n,q) = 1$。同时，系数 $a_{j+m,j}$ 和 $a_{j+n,j}$ 不为零，同样 $B$ 的情况类似，但范围是从 $j+p$ 到 $j+q$，且 $b_{j+p,j}$ 和 $b_{j+q,j}$ 不为零。

接下来，$R$ 是由 $A$ 和 $B$ 生成的交换 $\mathbb{C}$-子代数，$M$ 作为 $R$ 模，分数体 $K$ 是 $R$ 的分数体，那么 $M \otimes_R K$ 作为 $K$ 向量空间的维度 $h$ 是多少？

首先，我需要理解分数体 $K$ 的性质。当 $R$ 是交换环时，分数体 $K$ 是 $R$ 的局部化，即所有非零元素的逆元存在后的环。然后，$M \otimes_R K$ 应该是一个 $K$ 向量空间，其维度 $h$ 即为所求。

接下来，我需要分析 $M$ 作为 $R$ 模的结构，以及当张量积到 $K$ 之后，这个结构如何变化。通常，张量积到分数体 $K$ 之后，相当于将 $R$ 模 $M$ 转换为 $K$ 向量空间，其维度等于 $M$ 在 $R$ 上的秩，或者说，当 $M$ 是自由 $R$ 模时，其秩即为 $h$。但这里 $M$ 是无限维的，所以可能需要其他方法。

不过题目中说 $M \otimes_R K$ 是 $h$ 维的，所以需要找到这个 $h$ 的值。可能的思路是找到 $R$ 的作用在 $M$ 上的某种循环性质，或者通过分析 $A$ 和 $B$ 的作用，找到它们的公共特征向量，从而确定 $h$。

首先，考虑 $A$ 和 $B$ 是交换的，即 $AB = BA$。因此，它们可以同时被对角化，或者在某种意义上有共同的特征向量。不过，由于 $M$ 是无限维的，可能不存在通常意义上的基，但或许可以考虑某种形式的结构。

接下来，考虑 $A$ 和 $B$ 的具体形式。$A$ 的作用是将 $e_j$ 映射到从 $j+m$ 到 $j+n$ 的线性组合，而 $B$ 的作用是将 $e_j$ 映射到从 $j+p$ 到 $j+q$ 的线性组合。这里 $m \leq n < 0$，而 $p$ 和 $q$ 都是正数，并且满足 $\gcd(-m,p) = 1$，$\gcd(-n,q) = 1$。

例如，假设 $m = -k$，其中 $k$ 是正整数，且 $\gcd(k,p) = 1$。同样，$n = -l$，其中 $l$ 是正整数，且 $\gcd(l,q) = 1$。不过题目中的条件是 $m \leq n < 0$ 所以 $m$ 和 $n$ 都是负数，而 $p$ 和 $q$ 是正数。

接下来，我需要考虑 $A$ 和 $B$ 的作用如何影响 $e_j$。例如，$A$ 将 $e_j$ 移动到更低的索引（因为 $m$ 是负数，所以 $j+m$ 可能比 $j$ 小），而 $B$ 将 $e_j$ 移动到更高的索引。由于 $A$ 和 $B$ 可交换，它们的组合作用应该满足某种交换性条件。

或许可以考虑 $A$ 和 $B$ 生成了一个格，或者某种周期性的结构，从而在分数体 $K$ 的作用下，每个轨道生成一个一维的 $K$ 向量空间，但需要更具体的分析。

另一个可能的思路是，考虑每个 $e_j$ 在 $R$ 的作用下的轨道。由于 $A$ 和 $B$ 可以移动索引，可能每个轨道对应某个格点，而分数体 $K$ 允许这些移动的逆操作，因此每个轨道生成一个一维的 $K$ 向量空间。但需要确定这样的轨道是否独立，或者是否存在多个轨道。

不过，题目中的条件 $\gcd(-m,p) = 1$ 和 $\gcd(-n,q) = 1$ 可能在这里起到关键作用。例如，假设 $m = -1$，$n = -1$，那么 $A$ 的作用是每个 $e_j$ 被映射到 $e_{j-1}$ 和可能的其他项，但题目中的 $A$ 是有限带宽的，但每个 $Ae_j$ 只涉及从 $j+m$ 到 $j+n$ 的项，这里 $m \leq n < 0$ 所以比如 $m = -2$，$n = -1$，那么 $Ae_j = a_{j-2,j}e_{j-2} + a_{j-1,j}e_{j-1}$ 并且 $a_{j-2,j}$ 和 $a_{j-1,j}$ 不为零。类似地，$B$ 的作用是每个 $e_j$ 被映射到从 $j+p$ 到 $j+q$ 的项，比如 $p = 1$，$q = 2$，那么 $Be_j = b_{j+1,j}e_{j+1} + b_{j+2,j}e_{j+2}$，同样非零系数。

由于 $A$ 和 $B$ 可交换，它们的组合作用必须满足一定的相容性条件。例如，$ABe_j = BAe_j$，这可能导致系数之间的某种递归关系。但具体如何呢？

或许可以考虑 $A$ 和 $B$ 作为移位算子的变形。例如，如果 $A$ 是向下的移位，$B$ 是向上的移位，并且它们的组合满足交换性，那么可能需要某种周期性的条件。但这里 $A$ 和 $B$ 的移动步数可能由 $m$ 和 $p$ 决定，而 $\gcd(-m,p) = 1$ 的条件可能意味着它们的移动步数是互质的，从而生成整个整数格，因此每个轨道在分数体 $K$ 下可能生成一维空间，从而 $h = 1$。但需要验证这个猜测是否正确。

例如，假设 $m = -1$，$p = 1$，$\gcd(1,1) = 1$，满足条件。那么 $A$ 的作用是 $e_j$ 映射到 $e_{j-1}$ 和可能的其他项，而 $B$ 的作用是将 $e_j$ 映射到 $e_{j+1}$ 和可能的其他项。如果 $A$ 和 $B$ 是单纯的移位算子，即 $Ae_j = e_{j-1}$，$Be_j = e_{j+1}$，那么 $AB = BA$ 因为移位是交换的。此时，分数体 $K$ 应该包含 $A^{-1}$ 和 $B^{-1}$，即允许向上和向下移位。此时，$M \otimes_R K$ 中的每个元素可以表示为某个 $e_j$ 的标量倍数，因为通过 $A$ 和 $B$ 的逆操作，可以到达任何 $e_k$，因此张量积后的空间可能是一维的，即 $h = 1$。

但题目中的 $A$ 和 $B$ 并不是单纯的移位，而是有多个项的线性组合，比如 $Ae_j$ 可能包含从 $j+m$ 到 $j+n$ 的多个项，而不仅仅是单一步长。但题目中给出的条件是非零的首项和末项系数，即 $a_{j+m,j}$ 和 $a_{j+n,j}$ 非零，同样对于 $B$ 也是如此。这可能意味着 $A$ 和 $B$ 的作用类似于带有最高和最低项的差分或微分算子，但这里是在无限维空间上。

不过，由于 $A$ 和 $B$ 是交换的，它们的组合作用可能需要满足某种多项式关系，或者它们的系数满足某种递归关系。例如，考虑 $A$ 和 $B$ 作为多项式环中的生成元，但这里可能更复杂。

另一个角度是考虑 $R$ 的分数体 $K$ 是一个域，而 $M \otimes_R K$ 作为 $K$ 向量空间的维度 $h$，等于 $M$ 作为 $R$ 模的秩。如果 $M$ 是一个秩为 $h$ 的自由 $R$ 模，那么张量积后的维度就是 $h$。但这里 $M$ 是无限维的，所以可能需要通过其他方式确定这个秩。

或许可以考虑每个 $e_j$ 在 $R$ 的作用下的轨道是否生成整个 $M$，或者是否存在多个生成元。例如，如果通过 $A$ 和 $B$ 的作用，可以从一个 $e_j$ 生成所有其他 $e_k$，那么 $M$ 作为 $R$ 模是循环的，即秩 1，因此 $h = 1$。否则，可能需要更多的生成元。

这里的关键可能在于 $A$ 和 $B$ 的移动步数是否互质。例如，假设 $A$ 的移动步数是 $m' = -m$，$B$ 的移动步数是 $p$，且 $\gcd(m',p) = 1$，那么根据贝祖定理，存在整数 $a$ 和 $b$ 使得 $a*m' + b*p = 1$，这意味着通过组合 $A$ 和 $B$ 的作用，可以移动 1 步，从而生成所有位置的 $e_j$。因此，这种情况下，可能 $M$ 作为 $R$ 模是循环的，即由单个生成元生成，因此 $h = 1$。

但题目中的条件不仅仅是关于步数的，还有生成元 $A$ 和 $B$ 的具体形式。例如，$A$ 移动范围是从 $j+m$ 到 $j+n$，其中 $m \leq n < 0$，即可能覆盖多个向下的步数，而 $B$ 则是向上的多个步数。但题目中的 $\gcd$ 条件可能确保这些步数的组合可以生成任意整数步，从而允许通

过组合 $A$ 和 $B$ 的作用到达任何位置，从而使得 $M$ 作为 $R$ 模是循环的，因此 $h = 1$。

例如，假设 $m = -2$，$n = -1$，$p = 3$，$q = 4$，那么 $\gcd(2, 3) = 1$，$\gcd(1, 4) = 1$。这时候，$A$ 的作用可以向下移动至少 1 或 2 步，而 $B$ 的作用可以向上移动 3 或 4 步。由于 $\gcd(2, 3) = 1$ 可以通过组合 $A$ 和 $B$ 的移动得到任意整数步。例如，使用贝祖定理，存在整数 $a, b$ 使得 $2a + 3b = 1$，比如 $a = 2$，$b = -1$，因为 $2 * 2 + 3 * (-1) = 4 - 3 = 1$。因此，通过 $A^2$ 和 $B^{-1}$ 的组合，可以得到移动 1 步的效果。这可能意味着，通过组合 $A$ 和 $B$ 的作用，可以生成任何位置的 $e_j$，从而使得 $M$ 作为 $R$ 模是循环的，即由单个生成元生成，因此张量积后的维度 $h = 1$。

不过，这里需要注意的是，虽然理论上存在这样的组合，但实际中 $A$ 和 $B$ 可能不是单纯的移位算子，而是有多个项的线性组合，因此可能需要更仔细的分析。例如，$A$ 可能是一个下移加上其他项，但系数非零，所以可能通过线性组合来消除其他项的影响，从而得到纯粹的移位效果。不过这可能涉及到分数体 $K$ 中的元素，即允许使用 $A$ 和 $B$ 的逆来构造这些移位。

例如，假设 $A$ 的作用是 $Ae_j = a_{j-2,j}e_{j-2} + a_{j-1,j}e_{j-1}$，而 $B$ 的作用是 $Be_j = b_{j+3,j}e_{j+3} + b_{j+4,j}e_{j+4}$。由于 $A$ 和 $B$ 可交换，它们的组合必须满足 $AB = BA$。这可能对系数 $a$ 和 $b$ 有严格的要求，使得它们的组合可以交换。例如，可能需要系数满足某种递归关系，从而允许交换性。

不过，这里的关键可能不在于具体的系数，而在于移动的步数 $m$ 和 $p$ 的互质性。因为题目中给出的条件是 $\gcd(-m, p) = 1$ 和 $\gcd(-n, q) = 1$，这可能暗示着通过组合 $A$ 和 $B$ 的移动步数，可以生成整个整数格，从而使得每个轨道在分数体 $K$ 下生成一个一维空间，因此 $h = 1$。

因此，综合以上分析，可能的答案是 $h = 1$ 即 $M \otimes_R K$ 是一维的 $K$ 向量空间，因此 $h = 1$。所以答案应该是

$$\boxed{1}$$

## Answer

定无限次元の複素ベクトル空間 $M = \bigoplus_{i \in \mathbb{Z}} \mathbb{C}e_i$ と、その線形自己準同型環 $\text{End}(M)$ の可換な元 $A$ と $B$ で生成される部分代数 $R$ について考える。$A$ と $B$ はそれぞれ、整数 $m \le n < 0 < p \le q$ と条件 $\gcd(-m, p) = \gcd(-n, q) = 1$ を満たす範囲で基底ベクトル $e_j$ を線形結合に写す。ここで、分数体 $K$ における $M \otimes_R K$ の次元 $h$ を求める。

分析の要点:
1. **$A$ と $B$ の作用**: $A$ は負の方向、$B$ は正の方向に基底ベクトルを移動させる。特に、$\gcd(-m, p) = 1$ と $\gcd(-n, q) = 1$ より、ベズーの等式からこれらの移動ステップの組み合わせで任意の整数ステップを生成できる。
2. **可換性**: $A$ と $B$ の可換性により、これらの作用を組み合わせて任意の位置の基底ベクトルを生成可能。
3. **分数体 $K$ の性質**: $K$ は $R$ の分数体であり、$A$ と $B$ の逆元を含む。これにより、任意の基底ベクトル $e_j$ から他の基底ベクトルを生成できる。
4. **次元 $h$**: $M \otimes_R K$ は $K$ 上のベクトル空間であり、基底ベクトル間の生成関係から1次元となる。

結論: $R$ の作用により、任意の基底ベクトル $e_j$ から他の全ての基底ベクトルを生成できるため、$M \otimes_R K$ は1次元 $K$-ベクトル空間となる。したがって、$h = 1$。

$$\boxed{1}$$

## Ground Truth

1

---

# Deepseek-R1-671B   (Output Length: 6269)

● **Problem: Spanish**   ● **Thinking: English**   ● **Answer: English**

## Problem

Hoy, Iván el Confesor prefiere funciones continuas $f : [0, 1] \to \mathbb{R}$ que satisfacen $f(x) + f(y) \ge |x - y|$ para todos los pares $x, y \in [0, 1]$. Encuentra el mínimo de $\int_0^1 f$ sobre todas las funciones preferidas.

Nota: Por favor, coloque la respuesta final en el $\boxed{}$.

## Thinking

Okay, so I need to find the minimum of the integral from 0 to 1 of $f(x)\, dx$, where $f$ is a continuous function that satisfies $f(x) + f(y) \ge |x - y|$ for all $x, y \in [0, 1]$. Hmm, let's start by understanding the problem step by step.

First, the function $f$ has to be continuous on $[0, 1]$, and for every pair of points $x$ and $y$ in this interval, the sum $f(x) + f(y)$ must be at least the distance between $x$ and $y$. The goal is to find the function $f$ that meets these conditions and has the smallest possible integral. That integral is essentially the area under the curve of $f$ from 0 to 1, so we want to make this area as small as possible while still satisfying the inequality condition.

Let me think about how to approach this. Maybe I can start by analyzing the inequality $f(x) + f(y) \ge |x - y|$. Since this has to hold for all $x$ and $y$, perhaps I can fix one variable and see what constraints that imposes on $f$.

For example, if I fix $y$ and let $x$ vary, then for each $y$, $f(x)$ must be at least $|x - y| - f(y)$. But since this has to hold for all $x$, maybe I can find a lower bound for $f(x)$ by considering all possible $y$. Alternatively, if I fix $x$ and let $y$ vary, then $f(y) \ge |x - y| - f(x)$ for all $y$. So for each $x$, $f(y)$ must be at least $|x - y| - f(x)$ for every $y$. But since this must hold for all $y$, perhaps the maximum of these lower bounds over $y$ gives a constraint on $f(x)$.

Wait, maybe we can rephrase the inequality. For all $x, y$, $f(x) + f(y) \ge |x - y|$. Let's fix $x$ and consider the minimum value of $f(y)$ over $y$. But since the inequality must hold for all $y$, the minimal value of $f(y)$ would be when $y$ is such that $|x - y|$ is maximized. But I'm not sure if that's helpful.

Alternatively, for each $x$, the minimal value of $f(x)$ can be found by considering the maximum over $y$ of $(|x - y| - f(y))$. But since the inequality must hold for all $y$, we have $f(x) \geq \sup_{y \in [0,1]} (|x - y| - f(y))$. This seems like a recursive kind of condition. Maybe this is related to the concept of conjugates or Fenchel transforms in convex analysis? Not sure.

Alternatively, maybe we can consider specific points. For example, take $y = x$. Then the inequality becomes $f(x) + f(x) \geq |x - x| = 0$, which is always true since $f$ is real-valued. So that doesn't give us any new information.

What if we take $y = 0$? Then for any $x$, $f(x) + f(0) \geq |x - 0| = x$. Similarly, taking $y = 1$, we get $f(x) + f(1) \geq |x - 1| = 1 - x$. So combining these two, for each $x$, $f(x) \geq \max\{x - f(0), (1 - x) - f(1)\}$.

Hmm, that's interesting. So for each $x$, $f(x)$ is at least the maximum of $x - f(0)$ and $(1 - x) - f(1)$. Let me write that down:

$$f(x) \geq \max\{x - f(0), (1 - x) - f(1)\} \text{ for all } x \in [0, 1].$$

But since $f$ is continuous, these lower bounds must be satisfied everywhere. Also, since we want to minimize the integral of $f$, we might guess that the minimal $f$ is exactly equal to this lower bound. That is, $f(x) = \max\{x - f(0), (1 - x) - f(1)\}$ for all $x$. If that's the case, then perhaps we can set up equations for $f(0)$ and $f(1)$.

Let me denote $a = f(0)$ and $b = f(1)$. Then, the function becomes $f(x) = \max\{x - a, (1 - x) - b\}$.

But since $f(0) = a$, let's compute $f(0)$. At $x = 0$, $f(0) = \max\{0 - a, (1 - 0) - b\} = \max\{-a, 1 - b\}$.

But $f(0) = a$, so we have $a = \max\{-a, 1 - b\}$.

Similarly, at $x = 1$, $f(1) = \max\{1 - a, (1 - 1) - b\} = \max\{1 - a, -b\}$.

But $f(1) = b$, so $b = \max\{1 - a, -b\}$.

So now we have two equations:

$$a = \max\{-a, 1 - b\} \tag{4}$$

$$b = \max\{1 - a, -b\} \tag{5}$$

We need to solve these equations for $a$ and $b$. Let's analyze the first equation: $a = \max\{-a, 1 - b\}$

This implies that $a$ is equal to the larger of $-a$ and $1 - b$. Similarly for the second equation.

Let's consider possible cases.

Case 1: For equation 1, suppose that $-a \geq 1 - b$. Then $a = -a \Rightarrow 2a = 0 \Rightarrow a = 0$. But if $a = 0$, then from $-a \geq 1 - b \Rightarrow 0 \geq 1 - b \Rightarrow b \geq 1$.

But from equation 2, $b = \max\{1 - a, -b\} = \max\{1 - 0, -b\} = \max\{1, -b\}$. Since $b \geq 1$, then $\max\{1, -b\} = 1$ if $b \geq 1$. But $b = \max\{1, -b\}$ implies that if $b \geq 1$, then $b = 1$. But we had $b \geq 1$, so $b = 1$.

So in this case, $a = 0$ and $b = 1$. Let's check if this satisfies the original assumption that $-a \geq 1 - b$. $-0 = 0 \geq 1 - 1 = 0$. So equality holds. So this is a possible solution.

Case 2: For equation 1, suppose that $1 - b \geq -a$. Then $a = 1 - b$. Similarly, for equation 2, let's see.

From equation 2: $b = \max\{1 - a, -b\}$

If we substitute $a = 1 - b$ into equation 2, we get:

$$b = \max\{1 - (1 - b), -b\} = \max\{b, -b\}$$

So $b = \max\{b, -b\}$, which is always true because $\max\{b, -b\} = |b|$. But since $b$ is a real number, we have that $b = |b|$, which implies that $b \geq 0$.

So in this case, $a = 1 - b$ and $b \geq 0$.

But we need to ensure that the assumption for case 2 holds, which is $1 - b \geq -a$. Since $a = 1 - b$, this becomes $1 - b \geq -(1 - b) \Rightarrow 1 - b \geq -1 + b \Rightarrow 1 - b \geq -1 + b \Rightarrow 2 \geq 2b \Rightarrow b \leq 1$.

So combining with $b \geq 0$ from equation 2, we have $0 \leq b \leq 1$.

Therefore, in case 2, we have $a = 1 - b$ and $0 \leq b \leq 1$.

But we need to check if these values of $a$ and $b$ satisfy the original function definition. Let's recall that $f(x) = \max\{x - a, (1 - x) - b\}$.

Substituting $a = 1 - b$, we get

$$f(x) = \max\{x - (1 - b), (1 - x) - b\} = \max\{x - 1 + b, 1 - x - b\}.$$

So $f(x) = \max\{(x - 1) + b, (1 - x) - b\}$.

Hmm, perhaps we can analyze where each term is larger. Let's set the two expressions equal to each other:

$$(x - 1) + b = (1 - x) - b$$

Solving for $x$:

$$x - 1 + b = 1 - x - b$$

Bring variables to one side:

$$x + x = 1 + 1 - b - b$$

$$2x = 2 - 2b$$

$$x = 1 - b$$

So the two expressions are equal at $x = 1 - b$. Therefore, for $x \leq 1 - b$, which is the point where the two expressions cross, which one is larger?

Let's take $x < 1 - b$. Then, let's compute $(x - 1) + b$ versus $(1 - x) - b$.

$$(x - 1) + b = x - 1 + b$$

$$(1 - x) - b = 1 - x - b$$

Compare these two:

$$x - 1 + b \text{ vs } 1 - x - b$$

Which is larger?

Subtract the two:

$$(x - 1 + b) - (1 - x - b) = x - 1 + b - 1 + x + b = 2x + 2b - 2$$

If $x < 1 - b$, then $2x + 2b - 2 < 2(1 - b) + 2b - 2 = 2 - 2b + 2b - 2 = 0$. Therefore, $(x - 1 + b) - (1 - x - b) < 0$, so $(x - 1 + b) < (1 - x - b)$. Therefore, for $x < 1 - b$, the maximum is $1 - x - b$. For $x \geq 1 - b$, the maximum is $x - 1 + b$. Therefore, the function $f(x)$ can be written as:

$$f(x) = \begin{cases} 1 - x - b, & \text{if } x \leq 1 - b \\ x - 1 + b, & \text{if } x \geq 1 - b \end{cases}$$

But since $b$ is between 0 and 1 (from case 2), $1 - b$ is between 0 and 1. So this defines a piecewise linear function with a V-shape at $x = 1 - b$. Now, since $f$ is continuous (as given in the problem), this piecewise function is continuous as long as the two pieces meet at $x = 1 - b$. Let's check the value at $x = 1 - b$:

Left limit:

$$1 - (1 - b) - b = 1 - 1 + b - b = 0$$

.

Right limit:

$$(1 - b) - 1 + b = -b + b = 0$$

So they meet at 0. Wait, that's strange. Wait, but if $f(x)$ is defined as

$$\max\{(x - 1) + b, (1 - x) - b\}$$

then at $x = 1 - b$, both terms are equal to

$$(1 - b - 1) + b = -b + b = 0$$

and

$$(1 - (1 - b)) - b = b - b = 0.$$

So $f(1 - b) = 0$. But since $f$ is supposed to be continuous, this is okay. However, the function $f(x)$ reaches zero at $x = 1 - b$. But since $f(x)$ must satisfy

$$f(x) + f(y) \geq |x - y|$$

for all $x$, $y$, having $f(x) = 0$ at some point might be problematic. Let's check.
Suppose $x = 1 - b$ and $y = 1 - b$. Then

$$f(x) + f(y) = 0 + 0 = 0,$$

but

$$|x - y| = 0,$$

so

$$0 \geq 0,$$

which is okay. For other points, say $x = 1 - b$ and $y = 0$. Then

$$f(x) + f(y) = 0 + a = 0 + (1 - b) \quad \text{(since } a = 1 - b\text{)}.$$

So we have

$$1 - b \geq |(1 - b) - 0| = 1 - b.$$

So equality holds. Similarly, for $x = 1 - b$ and $y = 1$,

$$f(x) + f(y) = 0 + b \geq |(1 - b) - 1| = b.$$

So

$$0 + b \geq b,$$

which is equality. So that's okay.

But wait, if $f(x)$ is zero at $x = 1 - b$, then when we take another point $y$, we need $f(y)$ to compensate for the distance from $y$ to $1 - b$. For example, take $y = 1 - b + \epsilon$. Then

$$f(y) = (y - 1) + b = (1 - b + \epsilon - 1) + b = (-b + \epsilon) + b = \epsilon.$$

Then $f(y) = \epsilon$. Then

$$f(x) + f(y) = 0 + \epsilon = \epsilon,$$

which must be

$$\geq |x - y| = \epsilon.$$

So equality holds. Similarly, if $y = 1 - b - \epsilon$, then

$$f(y) = 1 - y - b = 1 - (1 - b - \epsilon) - b = 1 - 1 + b + \epsilon - b = \epsilon.$$

Then

$$f(x) + f(y) = 0 + \epsilon = \epsilon \geq |x - y| = \epsilon.$$

So equality again. So it seems that this function $f(x)$ meets the condition with equality along the lines where one variable is $1 - b$.

Therefore, this function is tight at certain points, which is a good sign that it might be the minimal function.

Now, we need to compute the integral of $f(x)$ from 0 to 1. Let's compute the integral of the piecewise function.

The function $f(x)$ is:

- For $x$ from 0 to $1 - b$:

$$f(x) = 1 - x - b$$

- For $x$ from $1 - b$ to 1:

$$f(x) = x - 1 + b$$

Therefore, the integral is the sum of two integrals:

$$\int_0^{1-b} (1 - x - b) \, dx + \int_{1-b}^1 (x - 1 + b) \, dx$$

Let's compute the first integral:

$$\int_0^{1-b} (1 - x - b) \, dx = \int_0^{1-b} (1 - b - x) \, dx$$

This is the integral of a linear function. The antiderivative is

$$(1 - b)x - \frac{1}{2}x^2$$

evaluated from 0 to $1 - b$.
At upper limit:

$$(1 - b)(1 - b) - \frac{1}{2}(1 - b)^2 = (1 - b)^2 - \frac{1}{2}(1 - b)^2 = \frac{1}{2}(1 - b)^2$$

At lower limit:

$$0 - 0 = 0$$

So the first integral is

$$\frac{1}{2}(1 - b)^2$$

Now the second integral:

$$\int_{1-b}^1 (x - 1 + b) \, dx = \int_{1-b}^1 (x - (1 - b)) \, dx$$

Again, a linear function. The antiderivative is

$$\frac{1}{2}x^2 - (1 - b)x$$

evaluated from $1 - b$ to 1.
At upper limit (x=1):

$$\frac{1}{2}(1)^2 - (1 - b)(1) = \frac{1}{2} - (1 - b) = -\frac{1}{2} + b$$

At lower limit (x=1 - b):

$$\frac{1}{2}(1 - b)^2 - (1 - b)(1 - b) = \frac{1}{2}(1 - b)^2 - (1 - b)^2 = -\frac{1}{2}(1 - b)^2$$

Therefore, the integral is

$$\left[-\frac{1}{2} + b\right] - \left[-\frac{1}{2}(1 - b)^2\right] = -\frac{1}{2} + b + \frac{1}{2}(1 - b)^2$$

So combining both integrals:

$$\text{Total integral} = \frac{1}{2}(1 - b)^2 + \left(-\frac{1}{2} + b + \frac{1}{2}(1 - b)^2\right)$$

Simplify: First term:

$$\frac{1}{2}(1 - b)^2$$

Second term:

$$-\frac{1}{2} + b + \frac{1}{2}(1 - b)^2$$

So total integral =

$$\frac{1}{2}(1 - b)^2 + \left(-\frac{1}{2} + b\right) + \frac{1}{2}(1 - b)^2$$

Combine like terms:

$$= \left(\frac{1}{2} + \frac{1}{2}\right)(1 - b)^2 + \left(-\frac{1}{2} + b\right)$$

$$= (1)(1 - b)^2 + \left(-\frac{1}{2} + b\right)$$

Expand $(1 - b)^2$:

$$= 1 - 2b + b^2 + \left(-\frac{1}{2} + b\right)$$

Combine terms:

$$1 - 2b + b^2 - \frac{1}{2} + b$$

$$= \left(1 - \frac{1}{2}\right) + (-2b + b) + b^2$$

$$= \frac{1}{2} - b + b^2$$

Therefore, the integral is

$$b^2 - b + \frac{1}{2}.$$

So the integral of $f(x)$ is equal to

$$b^2 - b + \frac{1}{2}.$$

Now, since in case 2, we have $0 \le b \le 1$, we need to find the value of $b$ in $[0, 1]$ that minimizes the quadratic expression $b^2 - b + \frac{1}{2}$.
To find the minimum of a quadratic function $f(b) = b^2 - b + \frac{1}{2}$, we can complete the square or take the derivative.
Taking derivative:

$$f'(b) = 2b - 1.$$

Setting this equal to zero gives

$$2b - 1 = 0 \Rightarrow b = \frac{1}{2}.$$

Since the coefficient of $b^2$ is positive, this critical point is a minimum. Therefore, the minimal value occurs at $b = \frac{1}{2}$.
Therefore, substituting $b = \frac{1}{2}$ into the integral expression:

$$\text{Integral} = \left(\frac{1}{2}\right)^2 - \left(\frac{1}{2}\right) + \frac{1}{2} = \frac{1}{4} - \frac{1}{2} + \frac{1}{2} = \frac{1}{4}.$$

So the minimal integral is

$$\frac{1}{4}.$$

But wait, let's check if this is correct. If $b = \frac{1}{2}$, then $a = 1 - b = 1 - \frac{1}{2} = \frac{1}{2}$. So $a = b = \frac{1}{2}$.
Therefore, the function $f(x)$ is: For $x \le 1 - \frac{1}{2} = \frac{1}{2}$:

$$f(x) = 1 - x - \frac{1}{2} = \left(1 - \frac{1}{2}\right) - x = \frac{1}{2} - x$$

For $x \ge \frac{1}{2}$:

$$f(x) = x - 1 + \frac{1}{2} = x - \frac{1}{2}$$

So $f(x)$ is a V-shaped function that decreases from $x = 0$ to $x = \frac{1}{2}$ with slope -1, reaching 0 at $x = \frac{1}{2}$, then increases from $x = \frac{1}{2}$ to $x = 1$ with slope +1.
But wait, at $x = 0$:

$$f(0) = \frac{1}{2} - 0 = \frac{1}{2},$$

which matches $a = \frac{1}{2}$. At $x = 1$:

$$f(1) = 1 - \frac{1}{2} = \frac{1}{2},$$

which matches $b = \frac{1}{2}$.
So the function is symmetric around $x = \frac{1}{2}$. Let's check if this function satisfies the original inequality

$$f(x) + f(y) \ge |x - y|.$$

Take any $x$ and $y$. Suppose $x \le \frac{1}{2}$ and $y \le \frac{1}{2}$. Then

$$f(x) = \frac{1}{2} - x$$

and

$$f(y) = \frac{1}{2} - y.$$

So

$$f(x) + f(y) = 1 - x - y.$$

We need to check if

$$1 - x - y \ge |x - y|.$$

But since $x$ and $y$ are both $\le \frac{1}{2}$, the maximum possible $|x - y|$ is $\frac{1}{2}$ (if one is 0 and the other is $\frac{1}{2}$). Let's see:

$$1 - x - y \ge |x - y|$$

Let me consider two cases: $x \ge y$ and $x < y$.
Case 1: $x \ge y$. Then $|x - y| = x - y$. So inequality becomes

$$1 - x - y \ge x - y \Rightarrow 1 - x - y \ge x - y \Rightarrow 1 - x \ge x \Rightarrow 1 \ge 2x \Rightarrow x \le \frac{1}{2}.$$

Which is true since $x \le \frac{1}{2}$.
Case 2: $x < y$. Then $|x - y| = y - x$. Inequality becomes

$$1 - x - y \ge y - x \Rightarrow 1 - x - y \ge y - x \Rightarrow 1 \ge 2y \Rightarrow y \le \frac{1}{2}.$$

Which is also true since $y \le \frac{1}{2}$.
Therefore, in this case, the inequality holds.
Now, suppose $x \le \frac{1}{2}$ and $y \ge \frac{1}{2}$. Then

$$f(x) = \frac{1}{2} - x$$

and

$$f(y) = y - \frac{1}{2}.$$

So

$$f(x) + f(y) = \left(\frac{1}{2} - x\right) + \left(y - \frac{1}{2}\right) = y - x.$$

But

$$|x - y| = y - x \quad \text{since } y \geq x.$$

So

$$f(x) + f(y) = y - x = |x - y|.$$

So equality holds here.
Similarly, if $x \geq \frac{1}{2}$ and $y \leq \frac{1}{2}$, then

$$f(x) + f(y) = (x - \frac{1}{2}) + (\frac{1}{2} - y) = x - y = |x - y|.$$

Finally, if both $x$ and $y$ are $\geq \frac{1}{2}$, then

$$f(x) = x - \frac{1}{2}$$

and

$$f(y) = y - \frac{1}{2}.$$

So

$$f(x) + f(y) = x + y - 1.$$

We need to check if

$$x + y - 1 \geq |x - y|.$$

Again, consider two cases: $x \geq y$ and $x < y$.
Case 1: $x \geq y$. Then $|x - y| = x - y$. So inequality becomes

$$x + y - 1 \geq x - y \Rightarrow y - 1 \geq -y \Rightarrow 2y \geq 1 \Rightarrow y \geq \frac{1}{2}.$$

Which is true since $y \geq \frac{1}{2}$.
Case 2: $x < y$. Then $|x - y| = y - x$. Inequality becomes

$$x + y - 1 \geq y - x \Rightarrow x - 1 \geq -x \Rightarrow 2x \geq 1 \Rightarrow x \geq \frac{1}{2}.$$

Which is true since $x \geq \frac{1}{2}$.
Therefore, in all cases, the inequality holds, and in some cases, it holds with equality. Therefore, this function $f(x)$ satisfies the required condition and has an integral of

$$\frac{1}{4}.$$

But wait, earlier when we considered case 1 where $a = 0$ and $b = 1$, let's check what the integral would be. If $a = 0$ and $b = 1$, then

$$f(x) = \max\{x - 0, (1 - x) - 1\} = \max\{x, -x\}.$$

But $\max\{x, -x\}$ is just $|x|$. Wait, but $f(x) = |x|$? Wait, no. Wait, when $a = 0$ and $b = 1$, then

$$f(x) = \max\{x - a, (1 - x) - b\} = \max\{x, (1 - x) - 1\} = \max\{x, -x\}.$$

But $\max\{x, -x\}$ is indeed $|x|$. So $f(x) = |x|$. But integrating $|x|$ from 0 to 1 would give

$$\frac{1}{2},$$

since the integral of $x$ from 0 to 1 is $\frac{1}{2}$. But in case 2, we found a function with integral $\frac{1}{4}$, which is smaller. Therefore, case 1 gives a larger integral, so the minimal must come from case 2.
Therefore, the minimal integral is

$$\frac{1}{4},$$

achieved by the function $f(x)$ which is

$$\frac{1}{2} - x$$

for $x \leq \frac{1}{2}$ and

$$x - \frac{1}{2}$$

for $x \geq \frac{1}{2}$.
But let me just verify once more. Suppose we take another function that maybe is not symmetric. For example, suppose we set $b$ to something else. If $b$ is not $\frac{1}{2}$, say $b = 0$. Then $a = 1 - 0 = 1$. Then

$$f(x) = \max\{x - 1, 1 - x - 0\} = \max\{x - 1, 1 - x\}.$$

But $x - 1$ is negative for $x$ in [0,1), so

$$f(x) = 1 - x.$$

Then the integral of $1 - x$ from 0 to 1 is $1 - \frac{1}{2} = \frac{1}{2}$, which is larger than $\frac{1}{4}$. Similarly, if $b = 1$, then $a = 0$, and

$$f(x) = \max\{x, -x\},$$

which is $|x|$, integral $\frac{1}{2}$. So indeed, the minimal is achieved at $b = \frac{1}{2}$.
Therefore, the minimal value of the integral is

$$\frac{1}{4}.$$

## Answer

To find the minimum of the integral $\int_0^1 f(x)\, dx$ over all continuous functions $f : [0, 1] \rightarrow \mathbb{R}$ that satisfy $f(x) + f(y) \geq |x - y|$ for all $x, y \in [0, 1]$, we start by analyzing the given inequality.

We consider the constraints imposed by fixing $y$ at the endpoints 0 and 1, leading to:
- $f(x) + f(0) \geq x$
- $f(x) + f(1) \geq 1 - x$

This implies $f(x) \geq \max\{x - f(0), (1 - x) - f(1)\}$. We denote $a = f(0)$ and $b = f(1)$, leading to the function $f(x) = \max\{x - a, (1 - x) - b\}$.

By solving the equations for $a$ and $b$ derived from the conditions at $x = 0$ and $x = 1$, we find that $a = 1 - b$ and $b = 1 - a$. This leads to the symmetric solution $a = b = \frac{1}{2}$.

The function $f(x)$ then becomes:
- $f(x) = \frac{1}{2} - x$ for $x \leq \frac{1}{2}$
- $f(x) = x - \frac{1}{2}$ for $x \geq \frac{1}{2}$

This function is continuous and satisfies the given inequality. The integral of this function is calculated as follows:
- The integral from 0 to $\frac{1}{2}$ of $\frac{1}{2} - x\, dx$
- The integral from $\frac{1}{2}$ to 1 of $x - \frac{1}{2}\, dx$

Both integrals evaluate to $\frac{1}{8}$, summing to a total integral of $\frac{1}{4}$.

Thus, the minimum value of the integral is $\boxed{\dfrac{1}{4}}$.

## Ground Truth

$$\frac{1}{4}$$

## Claude-3.7-sonnet-thinking   (Output Length: 8157)

### ● Problem: Korean   ● Thinking: **English**   ● Answer: Korean

### Problem

다항식 $x^3 - 5x^2 + 8x - 13$의 근의 $k$번째 거듭제곱의 합을 $s_k$라고 합시다. 특히, $s_0 = 3$, $s_1 = 5$, 그리고 $s_2 = 9$입니다. $a$, $b$, 그리고 $c$가 실수일 때, $s_{k+1} = a\, s_k + b\, s_{k-1} + c\, s_{k-2}$이며, $k = 2, 3, \ldots$ 입니다. $a + b + c$의 값을 구하십시오.

참고: 최종 답안을 $\boxed{}$ 안에 넣어 주세요.

### Thinking

Let's denote the roots of the polynomial $x^3 - 5x^2 + 8x - 13$ as $r_1$, $r_2$, $r_3$. Then, $s_k = r_1^k + r_2^k + r_3^k$ is the sum of the $k$-th powers of the roots.

We know that:
- $s_0 = r_1^0 + r_2^0 + r_3^0 = 1 + 1 + 1 = 3$
- $s_1 = r_1^1 + r_2^1 + r_3^1$ which is the sum of the roots.
- $s_2 = r_1^2 + r_2^2 + r_3^2$ which is the sum of the squares of the roots.

For a polynomial $x^3 + bx^2 + cx + d$, Vieta's formulas tell us that:
- sum of roots = $-b$
- sum of products of roots taken two at a time = $c$
- product of roots = $-d$

But our polynomial is $x^3 - 5x^2 + 8x - 13$. So, $b = -5$, $c = 8$, and $d = -13$.

Therefore:
- sum of roots = $-b = -(-5) = 5$, which matches the given $s_1 = 5$.
- sum of products of roots taken two at a time = $c = 8$. Let's denote this as $e_2 = r_1 r_2 + r_1 r_3 + r_2 r_3 = 8$.
- product of roots = $-d = -(-13) = 13$. Let's denote this as $e_3 = r_1 r_2 r_3 = 13$.

I'm given that:
- $s_0 = 3$
- $s_1 = 5$
- $s_2 = 9$

And I need to find $a$, $b$, and $c$ such that $s_{k+1} = as_k + bs_{k-1} + cs_{k-2}$ for $k \geq 2$.

Let me try to find a recurrence relation for $s_k$. I know that $s_1 = r_1 + r_2 + r_3$ and I'm trying to relate $s_{k+1}$ to $s_k$, $s_{k-1}$, and $s_{k-2}$.

Let's use the fact that the roots $r_1$, $r_2$, $r_3$ satisfy the equation $x^3 - 5x^2 + 8x - 13 = 0$. This means for each root $r_i$:

$$r_i^3 - 5r_i^2 + 8r_i - 13 = 0$$

Solving for $r_i^3$, I get:

$$r_i^3 = 5r_i^2 - 8r_i + 13$$

Now, for $k \geq 3$, I can multiply the above equation by $r_i^{k-3}$ to get:

$$r_i^k = 5r_i^{k-1} - 8r_i^{k-2} + 13r_i^{k-3}$$

Summing over all roots:

$$r_1^k + r_2^k + r_3^k = 5(r_1^{k-1} + r_2^{k-1} + r_3^{k-1}) - 8(r_1^{k-2} + r_2^{k-2} + r_3^{k-2}) + 13(r_1^{k-3} + r_2^{k-3} + r_3^{k-3})$$

Which means:

$$s_k = 5s_{k-1} - 8s_{k-2} + 13s_{k-3}$$

But, the formula asked for in the problem is:

$$s_{k+1} = as_k + bs_{k-1} + cs_{k-2}$$

Substituting $k$ with $k + 1$ in my recurrence relation:

$$s_{k+1} = 5s_k - 8s_{k-1} + 13s_{k-2}$$

So, $a = 5$, $b = -8$, and $c = 13$.

Therefore:

$$a + b + c = 5 + (-8) + 13 = 10$$

The answer is 10.

Actually, let me double-check this. We're given that:
$s_0 = 3$
$s_1 = 5$
$s_2 = 9$

And with my recurrence,

$$s_3 = 5s_2 - 8s_1 + 13s_0 = 5 \times 9 - 8 \times 5 + 13 \times 3 = 45 - 40 + 39 = 44$$

Let's see if this is correct by directly computing $s_3$ from the roots. We have:

$$s_3 = r_1^3 + r_2^3 + r_3^3$$

Using the fact that the roots satisfy $x^3 - 5x^2 + 8x - 13 = 0$, for each root $r_i$:

$r_i^3 = 5r_i^2 - 8r_i + 13$

So:

$s_3 = r_1^3 + r_2^3 + r_3^3 = 5(r_1^2 + r_2^2 + r_3^2) - 8(r_1 + r_2 + r_3) + 13 \times 3 = 5s_2 - 8s_1 + 39 = 5 \times 9 - 8 \times 5 + 39 = 45 - 40 + 39 = 44$

That checks out. So, our $a = 5$, $b = -8$, $c = 13$, and $a + b + c = 10$ are correct.

Actually I realize there's an issue here because I'm being asked for the recurrence relation for all $k$, but the relation I derived is specifically for $k \geq 3$. Let me try another approach.

In general, for a cubic polynomial $p(x) = x^3 + px^2 + qx + r$ with roots $r_1$, $r_2$, $r_3$, the Newton-Girard identities provide a way to relate $s_k$ to the coefficients of the polynomial and previous values of $s_j$.

In particular, for a cubic polynomial, we have:
$s_1 + p = 0$
$s_2 + p * s_1 + 2q = 0$
$s_3 + p * s_2 + q * s_1 + 3r = 0$
$s_4 + p * s_3 + q * s_2 + r * s_1 = 0$
$s_k + p * s_{k-1} + q * s_{k-2} + r * s_{k-3} = 0$ for $k \geq 4$

From the last equation, I can solve for $s_k$:
$s_k = -p * s_{k-1} - q * s_{k-2} - r * s_{k-3}$ for $k \geq 4$

For our polynomial $x^3 - 5x^2 + 8x - 13$, we have $p = -5$, $q = 8$, and $r = -13$. So:
$s_k = 5 * s_{k-1} - 8 * s_{k-2} + 13 * s_{k-3}$ for $k \geq 4$

But this is for $k$
$geq 4$. I need to check if this same relation holds for $k = 3$ as well.

Let's compute $s_3$ directly using the given values:
$s_3 = 5 * s_2 - 8 * s_1 + 13 * s_0 = 5 * 9 - 8 * 5 + 13 * 3 = 45 - 40 + 39 = 44$

Let me also compute $s_3$ using the Newton-Girard identities.
$s_3 + p * s_2 + q * s_1 + 3r = 0$
$s_3 + (-5) * 9 + 8 * 5 + 3 * (-13) = 0$
$s_3 - 45 + 40 - 39 = 0$
$s_3 = 45 - 40 + 39 = 44$

This matches our previous calculation, which is good.

Now let me determine if the same formula works for $k = 2$.

$s_2 = 9$
$s_2 \overset{?}{=} 5 * s_1 - 8 * s_0 + 13 * s_{-1}$
$9 \overset{?}{=} 5 * 5 - 8 * 3 + 13 * s_{-1}$
$9 \overset{?}{=} 25 - 24 + 13 * s_{-1}$
$9 \overset{?}{=} 1 + 13 * s_{-1}$
$8 \overset{?}{=} 13 * s_{-1}$
$s_{-1} \overset{?}{=}$

*frac*813

We need a term $s_{-1}$. Even though we can compute it, the recurrence relation given in the problem starts with $k = 2$, so I think the recurrence might be different for this specific case. Let's try to determine it.

I want to find $a$, $b$, and $c$ such that:
$s_{k+1} = a * s_k + b * s_{k-1} + c * s_{k-2}$ for $k \geq 2$

For $k = 2$:
$s_3 = a * s_2 + b * s_1 + c * s_0$
$44 = a * 9 + b * 5 + c * 3$

For $k = 3$:
$s_4 = a * s_3 + b * s_2 + c * s_1$
$s_4 = a * 44 + b * 9 + c * 5$

But I don't know $s_4$ yet. Let me compute it.

From our earlier recurrence relation for $k \geq 4$:
$s_4 = 5 * s_3 - 8 * s_2 + 13 * s_1 = 5 * 44 - 8 * 9 + 13 * 5 = 220 - 72 + 65 = 213$.

Now:
$213 = a * 44 + b * 9 + c * 5$

For $k = 4$:
$s_5 = a * s_4 + b * s_3 + c * s_2$
$s_5 = a * 213 + b * 44 + c * 9$

Let me compute $s_5$ using the recurrence relation for $k \geq 4$:
$s_5 = 5 * s_4 - 8 * s_3 + 13 * s_2 = 5 * 213 - 8 * 44 + 13 * 9 = 1065 - 352 + 117 = 830$.

So:
$830 = a * 213 + b * 44 + c * 9$

I now have three equations and three unknowns:
$44 = a * 9 + b * 5 + c * 3$
$213 = a * 44 + b * 9 + c * 5$
$830 = a * 213 + b * 44 + c * 9$
Let's use the first equation to create a constraint:
$44 = a * 9 + b * 5 + c * 3$
$\frac{44}{9} = a + \frac{5b}{9} + \frac{3c}{9}$
$a = \frac{44}{9} - \frac{5b}{9} - \frac{c}{3}$

And from the Newton-Girard identity, we know:
$s_4 + p * s_3 + q * s_2 + r * s_1 = 0$ for $k = 4$
$s_4 + (-5) * 44 + 8 * 9 + (-13) * 5 = 0$
$s_4 - 220 + 72 - 65 = 0$
$s_4 = 220 - 72 + 65 = 213$

And:
$s_5 + p * s_4 + q * s_3 + r * s_2 = 0$ for $k = 5$
$s_5 + (-5) * 213 + 8 * 44 + (-13) * 9 = 0$
$s_5 - 1065 + 352 - 117 = 0$
$s_5 = 1065 - 352 + 117 = 830$

Actually, I want to approach this slightly differently. Let's substitute our knowledge of the recurrence relation back into the general formula I'm looking for.

We know that the roots satisfy: $r_i^3 = 5r_i^2 - 8r_i + 13$, so $r_i^{k+1} = 5r_i^k - 8r_i^{k-1} + 13r_i^{k-2}$ for $k \geq 2$.

This means:
$s_{k+1} = 5s_k - 8s_{k-1} + 13s_{k-2}$ for $k \geq 2$.

So, $a = 5$, $b = -8$, and $c = 13$.

Therefore, $a + b + c = 5 + (-8) + 13 = 10$.

But wait, let me be completely sure by checking this recurrence relation for $k = 2$ and $k = 3$ using the given values:
For $k = 2$:
$s_{2+1} = 5s_2 - 8s_1 + 13s_0$
$s_3 = 5 \times 9 - 8 \times 5 + 13 \times 3$
$s_3 = 45 - 40 + 39 = 44$

For $k = 3$:

$s_{3+1} = 5s_3 - 8s_2 + 13s_1$
$s_4 = 5 \times 44 - 8 \times 9 + 13 \times 5$
$s_4 = 220 - 72 + 65 = 213$

These match the values I computed earlier using the Newton-Girard identities, so I'm confident that the recurrence relation is correct.

Actually, I need to double-check our working more carefully. Let's go back to the polynomial $x^3 - 5x^2 + 8x - 13$ and its roots.

For a polynomial $ax^3 + bx^2 + cx + d$, the Vieta formulas are:

$r_1 + r_2 + r_3 = -\frac{b}{a}$
$r_1 r_2 + r_1 r_3 + r_2 r_3 = \frac{c}{a}$
$r_1 r_2 r_3 = -\frac{d}{a}$

For the polynomial $x^3 - 5x^2 + 8x - 13$, we have $a = 1$, $b = -5$, $c = 8$, $d = -13$. So:

$r_1 + r_2 + r_3 = -\frac{-5}{1} = 5$
$r_1r_2 + r_1r_3 + r_2r_3 = \frac{8}{1} = 8$
$r_1r_2r_3 = -\frac{-13}{1} = 13$

So $s_1 = r_1 + r_2 + r_3 = 5$, which matches what we know.

Now, $s_0 = 3$ means that there are 3 roots (which is what I'd expect for a cubic polynomial).
Let's compute $s_2 = r_1^2 + r_2^2 + r_3^2$. We can use the identity:

$$(r_1 + r_2 + r_3)^2 = r_1^2 + r_2^2 + r_3^2 + 2(r_1r_2 + r_1r_3 + r_2r_3)$$

So, $s_1^2 = s_2 + 2 \cdot (r_1r_2 + r_1r_3 + r_2r_3)$, and thus:

$$s_2 = s_1^2 - 2 \cdot (r_1r_2 + r_1r_3 + r_2r_3) = 5^2 - 2 \cdot 8 = 25 - 16 = 9$$

This confirms that $s_2 = 9$.

Okay so now I want to find the recurrence relation. If the roots $r_i$ satisfy the polynomial $x^3 - 5x^2 + 8x - 13 = 0$, then for each $r_i$:

$$r_i^3 - 5r_i^2 + 8r_i - 13 = 0$$

Solving for $r_i^3$:

$$r_i^3 = 5r_i^2 - 8r_i + 13$$

Multiplying by $r_i^{k-3}$:

$$r_i^k = 5r_i^{k-1} - 8r_i^{k-2} + 13r_i^{k-3}$$

Taking the sum over all roots:

$$s_k = 5s_{k-1} - 8s_{k-2} + 13s_{k-3}$$

For $k = 3$, this gives:

$$s_3 = 5s_2 - 8s_1 + 13s_0 = 5 \times 9 - 8 \times 5 + 13 \times 3 = 45 - 40 + 39 = 44$$

But the problem states that $s_{k+1} = as_k + bs_{k-1} + cs_{k-2}$ for $k \geq 2$. Substituting $k + 1$ for $k$ in our recurrence:

$$s_{k+1} = 5s_k - 8s_{k-1} + 13s_{k-2} \text{ for } k \geq 2.$$

So, $a = 5$, $b = -8$, and $c = 13$. Therefore, $a + b + c = 5 + (-8) + 13 = 10$.

Let me double check this with a few values to be sure:

For $k = 2$:
$s_{2+1} = 5s_2 - 8s_1 + 13s_0$
$s_3 = 5 \times 9 - 8 \times 5 + 13 \times 3 = 45 - 40 + 39 = 44$

For $k = 3$:
$s_{3+1} = 5s_3 - 8s_2 + 13s_1$
$s_4 = 5 \times 44 - 8 \times 9 + 13 \times 5 = 220 - 72 + 65 = 213$

So, $a + b + c = 10$.

Actually, let me go back and try a slightly different approach, just to be sure.

For a cubic polynomial with roots $r_1$, $r_2$, $r_3$, the Newton-Girard identities relate the sums of powers of the roots to the coefficients of the polynomial:

For the polynomial $x^3 + px^2 + qx + r$ (not yet matching our specific polynomial):

$s_1 + p = 0$
$s_2 + ps_1 + 2q = 0$
$s_3 + ps_2 + qs_1 + 3r = 0$

Additionally, for $k \geq 1$:

$$s_k + ps_{k-1} + qs_{k-2} + rs_{k-3} = 0 \text{ for } k \geq 3$$

Our polynomial is $x^3 - 5x^2 + 8x - 13$, so $p = -5$, $q = 8$, $r = -13$. This means, for $k \geq 3$:

$s_k + (-5)s_{k-1} + 8s_{k-2} + (-13)s_{k-3} = 0$
$s_k - 5s_{k-1} + 8s_{k-2} - 13s_{k-3} = 0$

Solving for $s_k$:

$$s_k = 5s_{k-1} - 8s_{k-2} + 13s_{k-3}$$

But, the problem asks for a relation of the form: $s_{k+1} = as_k + bs_{k-1} + cs_{k-2}$.

Replacing $k$ with $k + 1$ in the above:

$$s_{k+1} = 5s_k - 8s_{k-1} + 13s_{k-2} \text{ for } k \geq 2$$

So, $a = 5$, $b = -8$, and $c = 13$. Therefore, $a + b + c = 10$.

Let me double-check this result:

For $k = 2$:
$s_3 = 5s_2 - 8s_1 + 13s_0 = 5 \times 9 - 8 \times 5 + 13 \times 3 = 45 - 40 + 39 = 44$

For $k = 3$:
$s_4 = 5s_3 - 8s_2 + 13s_1 = 5 \times 44 - 8 \times 9 + 13 \times 5 = 220 - 72 + 65 = 213$

So I'm confident that $a = 5$, $b = -8$, and $c = 13$, giving $a + b + c = 10$.

Actually, wait, there's a subtlety in how I set up the polynomial. Let me get the correct polynomial coefficients.

We have the polynomial $x^3 - 5x^2 + 8x - 13$. This is in the form $x^3 + bx^2 + cx + d$ with $b = -5$, $c = 8$, $d = -13$. The corresponding monic polynomial in the form $x^3 + px^2 + qx + r$ that I used for the Newton-Girard identities has $p = -5$, $q = 8$, $r = -13$.

Using the Newton-Girard identity for cubic polynomials:
$s_k + ps_{k-1} + qs_{k-2} + rs_{k-3} = 0$ for $k \geq 3$

For our polynomial:
$s_k - 5s_{k-1} + 8s_{k-2} - 13s_{k-3} = 0$ for $k \geq 3$

Solving for $s_k$:
$s_k = 5s_{k-1} - 8s_{k-2} + 13s_{k-3}$ for $k \geq 3$

Now, replacing $k$ with $k + 1$:
$s_{k+1} = 5s_k - 8s_{k-1} + 13s_{k-2}$ for $k \geq 2$

So, $a = 5$, $b = -8$, $c = 13$, and $a + b + c = 5 + (-8) + 13 = 10$. This is likely the correct answer.

I'm just going to double-check this again to rule out any errors. If the Newton-Girard identity for cubic polynomials is:
$s_k + ps_{k-1} + qs_{k-2} + rs_{k-3} = 0$ for $k \geq 3$

And our polynomial has coefficients $p = -5$, $q = 8$, $r = -13$, then:
$s_k - 5s_{k-1} + 8s_{k-2} - 13s_{k-3} = 0$ for $k \geq 3$
$s_k = 5s_{k-1} - 8s_{k-2} + 13s_{k-3}$ for $k \geq 3$

Let me verify this recurrence using the given values:

For $k = 3$:
$s_3 = 5s_2 - 8s_1 + 13s_0 = 5 \times 9 - 8 \times 5 + 13 \times 3 = 45 - 40 + 39 = 44$

Actually, I think I need to double check the Newton-Girard identity that I am using. Let's verify it by using direct computation.

Let's derive the Newton-Girard identities for a cubic polynomial. If $p(x) = x^3 + px^2 + qx + r$ and $p(x) = (x - r_1)(x - r_2)(x - r_3)$, then:

$s_1 = r_1 + r_2 + r_3 = -p$
$s_2 = r_1^2 + r_2^2 + r_3^2$
$s_3 = r_1^3 + r_2^3 + r_3^3$
and so on.

Now, $r_1$, $r_2$, and $r_3$ all satisfy the equation $x^3 + px^2 + qx + r = 0$. So, for each $i$:
$r_i^3 + pr_i^2 + qr_i + r = 0$
$r_i^3 = -pr_i^2 - qr_i - r$

Summing over all $i$:
$\sum_{i=1}^3 r_i^3 = -p\sum_{i=1}^3 r_i^2 - q\sum_{i=1}^3 r_i - 3r$
$s_3 = -ps_2 - qs_1 - 3r$
$s_3 + ps_2 + qs_1 + 3r = 0$

For $k \geq 4$, we can derive the recurrence by multiplying the relation $r_i^3 = -pr_i^2 - qr_i - r$ by $r_i^{k-3}$:
$r_i^k = -pr_i^{k-1} - qr_i^{k-2} - rr_i^{k-3}$

Summing over all $i$:
$\sum_{i=1}^3 r_i^k = -p\sum_{i=1}^3 r_i^{k-1} - q\sum_{i=1}^3 r_i^{k-2} - r\sum_{i=1}^3 r_i^{k-3}$
$s_k = -ps_{k-1} - qs_{k-2} - rs_{k-3}$
$s_k + ps_{k-1} + qs_{k-2} + rs_{k-3} = 0$

For our polynomial, $p = -5$, $q = 8$, $r = -13$. Thus:
$s_k + (-5)s_{k-1} + 8s_{k-2} + (-13)s_{k-3} = 0$ for $k \geq 3$
$s_k - 5s_{k-1} + 8s_{k-2} - 13s_{k-3} = 0$ for $k \geq 3$

Solving for $s_k$:
$s_k = 5s_{k-1} - 8s_{k-2} + 13s_{k-3}$ for $k \geq 3$

Now, to get the form $s_{k+1} = as_k + bs_{k-1} + cs_{k-2}$, I replace $k$ with $k + 1$:
$s_{k+1} = 5s_k - 8s_{k-1} + 13s_{k-2}$ for $k \geq 2$

So, $a = 5$, $b = -8$, and $c = 13$. Therefore, $a + b + c = 5 + (-8) + 13 = 10$.

Let's verify this recurrence using the given values:

For $k = 2$:
$s_3 = 5s_2 - 8s_1 + 13s_0 = 5 \times 9 - 8 \times 5 + 13 \times 3 = 45 - 40 + 39 = 44$

Let me already double check to be sure our computed values are correct. We have:
$s_0 = 3$ (given)
$s_1 = 5$ (given)
$s_2 = 9$ (given)

$s_3 = 44$ (computed)

To verify $s_3$ is correct,

method 1: let's compute $s_3$ using our polynomial identity.
$s_k + ps_{k-1} + qs_{k-2} + rs_{k-3} = 0$ for $k \geq 3$

For $k = 3$:
$s_3 + (-5)s_2 + 8s_1 + (-13)s_0 = 0$
$s_3 - 5 \times 9 + 8 \times 5 - 13 \times 3 = 0$
$s_3 - 45 + 40 - 39 = 0$
$s_3 = 45 - 40 + 39 = 44$

So we get the same value of $s_3 = 44$, which confirms our computation.

method 2: I note that the roots satisfy $r_i^3 = 5r_i^2 - 8r_i + 13$ and I can directly compute
$s_3 = \sum_{i=1}^{3} r_i^3 = 5 \sum_{i=1}^{3} r_i^2 - 8 \sum_{i=1}^{3} r_i + 13 \cdot 3 = 5s_2 - 8s_1 + 13s_0 = 5 \times 9 - 8 \times 5 + 13 \times 3 = 45 - 40 + 39 = 44$

So this again confirms that my result is correct.

So my final answer is $a + b + c = 10$.

## Answer

# 다항식의 근의 거듭제곱 합 관계식 구하기

주어진 다항식 $x^3 - 5x^2 + 8x - 13$의 근들을 $r_1, r_2, r_3$라고 하고, 이 근들의 $k$번째 거듭제곱의 합을 $s_k = r_1^k + r_2^k + r_3^k$라고 정의합니다.

문제에서는 $s_0 = 3, s_1 = 5, s_2 = 9$라고 주어졌습니다.

## 접근 방법

먼저 $s_k$가 만족하는 점화식을 찾아보겠습니다. 이를 위해 근들이 만족하는 방정식을 활용합니다.

다항식의 근들은 $x^3 - 5x^2 + 8x - 13 = 0$을 만족합니다. 따라서 각 근 $r_i$에 대해:
$r_i^3 - 5r_i^2 + 8r_i - 13 = 0$

이를 $r_i^3$에 대해 정리하면:
$r_i^3 = 5r_i^2 - 8r_i + 13$

위 식에 $r_i^{k-3}$을 곱하면:
$r_i^k = 5r_i^{k-1} - 8r_i^{k-2} + 13r_i^{k-3}$

모든 근에 대해 합하면:
$s_k = 5s_{k-1} - 8s_{k-2} + 13s_{k-3}$ ($k \geq 3$일 때)

## 관계식 확인

문제에서 요구하는 형태의 관계식은 $s_{k+1} = as_k + bs_{k-1} + cs_{k-2}$ ($k \geq 2$)입니다.

위 식에서 $k$를 $k + 1$로 대체하면:
$s_{k+1} = 5s_k - 8s_{k-1} + 13s_{k-2}$ ($k \geq 2$)

이를 문제에서 요구하는 형태와 비교하면:
$a = 5, b = -8, c = 13$

## 검증

$k = 2$일 때:
$s_3 = 5s_2 - 8s_1 + 13s_0 = 5 \times 9 - 8 \times 5 + 13 \times 3 = 45 - 40 + 39 = 44$

따라서 $a + b + c = 5 + (-8) + 13 = 10$입니다.

$\boxed{10}$

## Ground Truth

