# OpenReview forum: "PolyMath: Evaluating Mathematical Reasoning in Multilingual Contexts"
_NeurIPS.cc/2025/Datasets_and_Benchmarks_Track — NeurIPS 2025 Datasets and Benchmarks Track poster_

### Official Review · Reviewer_xMfm · 2025-06-19

**Rating:** 5
**Confidence:** 4

**Summary:**

This paper proposes a multilingual mathematical reasoning benchmark, denoted as PolyMath, which encompasses 18 languages and 4 difficulty levels. Based on this benchmark, the authors conduct a comprehensive evaluation on 8 non-reasoning LLMs and 8 reasoning LLMs, and find that even the best-performing model achieves only a 54.6 benchmark score. Moreover, they perform further analyses and reveal several key challenges, e.g., input-output language inconsistency.

**Dataset Code Accessibility:**

Yes

**Ethical Considerations:**

No, there are no or only very minor ethics concerns

**Limitations Weaknesses:**

- It will be better to provide some error analyses, e.g., the error type and the error difference between non-reasoning and reasoning LLMs.
- In Table 5, the reasoning LLMs can usually achieve better performance than the English task on non-English tasks, while the non-reasoning model does not exhibit this phenomenon. I suggest to provide some explanation.
- In section 3.4, I suggest to illustrate the input-output language consistency (as done in Figure 2) after adding the language control.
- Some typos. For example, in line 187, "significantly.." should be "significantly."

**Strengths Contributions:**

- The dataset is novel and could promote more related research in the math reasoning field.
- The empirical findings are insightful.
- The paper is well-written and easy to follow.

---

> ### Author Rebuttal · Authors · 2025-07-28
>
> Thank you for your recognition of our work, especially its novelty, contribution to the field, insightful findings, and clear writing. Regarding the weaknesses you raised (W1–W4), we address them one by one as follows:
>
> ---
>
> > W1: It will be better to provide some error analyses, e.g., the error type and the error difference between non-reasoning and reasoning LLMs.
>
>
> Thank you for pointing this out. We conducted an error analysis of PolyMath with reference to the error taxonomy in [1]. Specifically, we selected four models (two non-reasoning and two reasoning models) and sampled 60 examples (15 from each difficulty level), reporting the average results across English(en), Chinese(zh), and Japanese(ja).
>
>
>
> ||Qwen-2.5-Max (Non-Reasoning)|ChatGPT-4o-latest (Non-Reasoning)|Deepseek-R1-671B (Reasoning)|OpenAI-o3-mini-medium (Reasoning)|
> |-|-|-|-|-|
> |Logical Reasoning / Induction Error|52%|58%|76%|78%
> |Insufficient Classification Discussions|28%|23%|16%|13%|
> |Calculation Error|8%|6%|0%|1%|
> |Question Misunderstanding|10%|9%|2%|2%|
> |Unfinished Answering|0%|0%|5%|6%|
> |Others|2%|4%|1%|0%|
>
> *Key Insights:*
>
> * Firstly, logical errors and insufficient classification discussions account for the majority of mistakes (over 80%), which aligns with the common types of errors humans encounter when solving math problems.
>
> * Secondly, reasoning models exhibit a noticeably lower proportion of “elementary errors”, such as misunderstanding of the problem or calculation errors. Instead, their errors are more concentrated in complex logical reasoning, indicating a genuine improvement in their reasoning capabilities. On the other hand, we also observed that reasoning models have a higher rate of incomplete answers compared to non-reasoning models. This is often due to excessively long CoT that prevent the model from completing the reasoning within the maximum length limit, or due to step repetition and looping.
>
> ---
>
> > W2: In Table 5, the reasoning LLMs can usually achieve better performance than the English task on non-English tasks, while the non-reasoning model does not exhibit this phenomenon. I suggest to provide some explanation.
>
> Thank you for raising this important point of discussion. We are pleased to share our preliminary thoughts on this observation:
>
> * First, non-reasoning models generally perform better on English tasks is relatively easy to explain. These models are not trained with reinforcement learning algorithms and essentially operate as token-level fitting. Both reasoning and non-reasoning tasks are treated as surface-level sequence modeling problems. As such, their performance is highly sensitive to the distribution of training data across languages. Since high-resource languages like English are overrepresented in pretraining corpora, these models tend to acquire richer information in English, leading to relatively stronger performance on English tasks compared to other languages.
>
> * Compared to non-reasoning models, which are more likely to be influenced primarily by their language capabilities, reasoning models are jointly affected by both language and slow-thinking capabilities (learned by RL). As a result, reasoning in multilingual settings may exhibit complex interactions where language ability and reasoning capability can either reinforce or interfere with each other. In certain non-English scenarios (particularly some European languages) where performance exceeds that in English, we speculate that there may be a phenomenon of language-facilitated generalization of slow-thinking. However, we are currently unable to assert a definitive cause for this effect. We believe that PolyMath provides a valuable foundation for future work to investigate this intriguing phenomenon further.
>
> We will include these discussions in the next version of the paper.
>
> ---
>
>
> > W3: In section 3.4, I suggest to illustrate the input-output language consistency (as done in Figure 2) after adding the language control.
>
> Thanks for pointing this out. Regarding input-output language consistency under language control, **we have already provided the results in Table 18 and 19 of Appendix F.2**.
>
> Overall, when the model is forced to output in English, its compliance is nearly perfect. However, when the model is instructed to follow the input language, the input-output language consistency improves significantly but still falls short of perfection.
>
> We also conducted an in-depth analysis of the **correlation between language consistency and performance gains under language control**. For details, please refer to Appendix F.2.
>
>
> ---
>
> > W4: Some typos. For example, in line 187, "significantly.." should be "significantly."
>
> Thanks for your careful observation. We will thoroughly check the typos and correct them in the next version.
>
> ---
>
> We hope that these explanations and additional results address your concerns. We look forward to your positive feedback.
>
> [1] OlympiadBench: A Challenging Benchmark for Promoting AGI with Olympiad-Level Bilingual Multimodal Scientific Problems. https://arxiv.org/abs/2402.14008

---

> > ### Author Response · Authors · 2025-08-07
> >
> > Dear Reviewer xMfm,
> >
> > Thank you for your insightful comments. In response to the limitations you raised, we have included detailed experiments and analyses to address them.
> >
> > With that in mind, we would greatly appreciate knowing whether our responses have fully resolved your concerns and whether they might influence your final decision, as this would help facilitate convergence in the reviewer-author discussion phase.
> >
> > Thank you again for your time!
> >
> > Best,
> >
> > Authors

---

### Official Review · Reviewer_v5eL · 2025-06-30

**Rating:** 4
**Confidence:** 4

**Summary:**

This article introduces PolyMath, a multilingual mathematical reasoning benchmark that covers 18 languages and 4 difficulty distinctions. The article focuses on building benchmarks for multi-difficulty and multi-language math problems, ensuring high-quality and all-round translation among multiple languages. From the perspective of language, the article reveals several key challenges for LLMS in multilingual reasoning: (1) The reasoning performance of current LLMS varies greatly among different languages; (2) It is inferred that the consistency of the input and output languages of llm is relatively low, which may be related to performance; (3) Currently, the thinking length of LLMS varies significantly due to language. Furthermore, we have demonstrated that the output language in the control instructions may affect the inference performance, especially for some minor languages. This provides a promising direction for improving the multilingual capabilities in LLMS.

**Additional Feedback:**

1. Clarification Needed on the Role of Human Intervention in Difficulty Labeling

Line 179 mentions that semantic visualization via T-SNE “underscores the importance of human intervention in data collection and difficulty labeling.” However, the precise role of “human intervention” is not clearly defined. Does this refer to the validation of multilingual translations?

2. Further Details Needed on Accuracy Assurance in Mathematical Translation

Section 2.3 describes a multi-stage translation process involving LLM-based pre-translation, key term extraction, and expert calibration. However, it is unclear how language experts ensure accuracy and consistency of mathematical term across languages.

3. Ambiguity in the Definition of “Standard Accuracy” for Complex Answers

PolyMath includes a variety of answer types (e.g., Numeric, Expression, Equation, Interval, Set, Tuple), yet the paper states that evaluation is based on “standard accuracy” without further elaboration. For answer types such as expressions and equations, are algebraically equivalent forms considered correct?

**Dataset Code Accessibility:**

Yes

**Dataset Code Comments:**

On the homepage of the github website, the corresponding data content can be clearly found, and the accompanying instructions can also clearly introduce the format and distribution of the data set

**Ethical Considerations:**

No, there are no or only very minor ethics concerns

**Final Justification:**

Most of my concerns have been solved by the rebuttal. The workload of the paper is large however the experimental design seems not easy and lack of deep analysis.

**Limitations Weaknesses:**

1.	Unfair Evaluation Setup Between Reasoning and Non-Reasoning Models

The paper states that non-reasoning LLMs are evaluated using greedy decoding, whereas reasoning models are evaluated using sampling with multiple runs (16 times) followed by averaging (Lines 213–215). As demonstrated in prior work (e.g., Wang et al., ICLR 2023), self-consistency via sampling significantly improves performance. This evaluation asymmetry introduces bias in favor of reasoning models. To ensure fair comparison, I recommend adding an ablation study where both model types are evaluated using the same decoding strategy (e.g., either greedy or self-consistency) to isolate the impact of reasoning abilities.

2.	Insufficient Statistical Support and Explanation for Language Gap Claims

In Line 247, the authors claim that “at the same difficulty level, stronger reasoning LLMs tend to exhibit larger language gaps.” However, as shown in Table 5, models like Llama-3.3-70B-Instruct exhibit particularly high standard deviation and range values, especially on the PolyMath-Medium and PolyMath-Low subsets. This undermines the persuasiveness of the claim, as the performance difference between reasoning and non-reasoning models appears less significant in some cases. Furthermore, while Line 240 highlights that ACC differences (measured by std. and range) are used to reveal language gaps, the paper lacks deeper analysis of the underlying causes. For example, all models are said to exhibit substantial language gaps.

3.	Inconsistencies and Missing Explanations in Thinking Length Analysis

Figures 8 and 9 in the appendix (Line 622) suggest that, at the low difficulty level, both reasoning and non-reasoning LLMs show degraded performance as the thinking length increases. However, models such as Deepseek-v3 and Llama-3.3-70B-Instruct do not follow this trend. Additionally, the average thinking length of English inputs shows inconsistent behavior across difficulty levels. For instance, ChatGPT-4o-latest sometimes has the longest responses, and sometimes the shortest, depending on the level (Figure 8). These inconsistencies require further explanation. The authors are encouraged to offer more detailed analysis of such exceptions.

Wang, Xuezhi, et al. Self-consistency improves chain of thought reasoning in language models. ICLR 2023

**Strengths Contributions:**

The article presents a detailed and comprehensive benchmark from a new perspective. With high-quality translation, complete mathematical problems that are in line with the context have been constructed. The three-stage translation has well achieved the combination of mathematical language and natural language, and also reduced the occurrence of language ambiguity during the translation process. The difficulty levels of the math problems in the article are excellent. The four levels of math problems contain all the problems from easy to difficult, thus effectively improving the deficiency of high-difficulty math problems. The article has conducted thorough experiments and analyses on the performance differences and distinctions of large models among different languages, which can well demonstrate the logic and content of the research and lay a foundation for further exploration in the future.

---

> ### Author Rebuttal · Authors · 2025-07-31
>
> Thanks for your constructive feedback. Below, we respond to each of the three weaknesses (W1 - W3) and three feedback (F1 - F3) you raised, one by one:
>
> ---
>
>
> > W1: Unfair Evaluation Setup Between Reasoning and Non-Reasoning Models
>
> Thank you for pointing this out. Regarding your claim that *self-consistency (SC) can improve performance but introduces evaluation asymmetry*, we completely agree. However, **we would like to clarify that SC was not used in our evaluation, and sampling average is fundamentally different from SC. --- Our evaluation does not introduce any asymmetry.**
>
>
> * Our $N$-way sampling is performed *independently*, with the goal of reducing randomness.
> * In contrast, SC requires *joint selection* after $N$-way sampling, which introduces **external intervention (i.e., majority voting), this is the true source of evaluation asymmetry**.
>
> ***We use different sampling hyperparameters for different models with the goal of aligning with their recommended configurations, so as to activate their true reasoning capabilities from their model distributions for fair evaluation [1,2]. Greedy decoding is actually a deterministic form of sampling (temperature = 0).***
>
> To fully address your concerns, we additionally re-evaluated the open-source non-reasoning model originally decoded with the greedy strategy (Llama-3.3-70B-Instruct, Qwen-2.5-72B-Instruct, Qwen-2.5-Math-72B-Instruct, and Deepseek-v3) using the same sampling decoding as reasoning models and closed-source non-reasoning models. The ablation results are shown below:
>
>
> ||*average*|en|zh|ar|bn|de|es|fr|id|it|ja|ko|ms|pt|ru|sw|te|th|vi|
> |-|-|-|-|-|-|-|-|-|-|-|-|-|-|-|-|-|-|-|-|
> |Qwen2.5-Math-72B-Instruct (Greedy)|**21.0**|21.2|22.0|22.5|20.9|22.0|21.8|23.6|19.4|22.0|20.2|20.6|21.8|22.0|19.7|17.5|17.9|20.9|21.3|
> |Qwen2.5-Math-72B-Instruct (Sampling)|**20.4**|20.6|21.3|21.9|21.0|21.9|21.8|22.3|17.9|21.0|19.5|21.0|21.2|22.5|17.8|16.9|17.9|20.3|21.0|
> |Qwen2.5-72B-Instruct (Greedy)|**16.9**|19.8|18.1|15.6|17.7|16.3|19.6|14.7|19.3|18.8|15.7|16.6|16.7|17.7|17.7|11.3|13.3|17.4|18.6|
> |Qwen2.5-72B-Instruct (Sampling)|**16.0**|19.0|16.6|15.2|15.8|14.8|17.7|15.1|19.8|18.0|14.3|15.8|16.9|16.7|16.1|10.4|12.1|15.6|17.8|
>
>
> Due to computational resource constraints, we are currently unable to report the all models' results in the rebuttal phase. The ablation results for Llama-3.3-70B-Instruct and Deepseek-v3 will be included directly in the next version.
>
> The results from Qwen-2.5-72B-Instruct and Qwen-2.5-Math-72B-Instruct show that **for non-reasoning models, switching from greedy to sampling typically causes a slight performance drop, but the impact is modest.**
>
> We will include these ablation results along with the corresponding discussion in the next version.
>
>
>
> ---
>
> > W2: Insufficient Statistical Support and Explanation for Language Gap Claims
>
> Thank you for pointing it out. We will elaborate on our perspective in detail.
>
> > *(1) ... This undermines the persuasiveness of the claim, as the performance difference between reasoning and non-reasoning models appears less significant in some cases.*
>
> **We would like to clarify a misunderstanding: in this context (**lines 247–255**), our comparison is conducted within reasoning models only** --- What we mean is that **stronger reasoning models tend to exhibit larger language gaps compared to weaker reasoning models**. We did not perform any direct comparison between reasoning and non-reasoning models, as such a comparison would be inappropriate due to the large gap in their absolute performance. In lines 247–255, we did not include any non-reasoning models in the comparison.
>
> **The main analysis has been presented in lines 248-251**: *"in the three higher levels, Qwen-QwQ-32B consistently shows the highest std. and range. Models such as OpenAI-o3-mini-medium, Deepseek-R1-671B, and Gemini250 2.5-pro display significant language gaps at some levels, though inconsistently. In contrast, models such as OpenAI-o1-mini and Gemini-2.0-flash-thinking generally maintain smaller gaps."*
>
> Another special case — Qwen3-235B-A22B-Thinking — has also already been analyzed in **lines 252–254**. Its reduced performance gap is likely due to its multilingual reasoning training during the post-training phase. We believe these interpretations are well-justified.
>
>
> > *(2) While Line 240 highlights that ACC differences (measured by std. and range) are used to reveal language gaps, the paper lacks deeper analysis of the underlying causes. For example, all models are said to exhibit substantial language gaps.*
>
> Regarding the cause of the *significant language gap*, we believe that the distribution of training data across languages remains a dominant factor under current research progress. Available information shows that some open-source models, such as Deepseek-R1, explicitly state in their technical reports that they only optimize reasoning capabilities in English and Chinese. QwQ-32B follows a similar pattern. While we do not rule out the possibility of some degree of internal language generalization, high-quality multilingual alignment remains underexplored.
>
> In contrast, Qwen3-Thinking explicitly incorporates multilingual long-CoT data during its post-training phase. As a result, compared to other reasoning models with similar overall performance, it exhibits a relatively smaller language gap. (You may refer to the evolution from QwQ-32B/Deepseek-R1-671B to Qwen3-235B-A22B-Thinking)
>
> That said, we believe the deeper underlying causes merit further investigation from both academia and industry. PolyMath can serve as a valuable foundation for such exploration.
>
> ---
>
> > W3: Inconsistencies and Missing Explanations in Thinking Length Analysis
>
> Thank you for pointing it out. We will elaborate on our perspective in detail.
>
> > *(1) models such as Deepseek-v3 and Llama-3.3-70B-Instruct do not follow this trend*
>
> Yes, we will revise this statement more rigorously in the updated version, for example, changing “both reasoning and non-reasoning models” to “reasoning models and many non-reasoning models.”
>
>
> > *(2) the average thinking length of English inputs shows inconsistent behavior across difficulty levels... These inconsistencies require further explanation*
>
> This phenomenon is more pronounced in non-reasoning models, especially in the ChatGPT-4o-latest model you mentioned.
>
> * Regarding "English responses are the longest at higher difficulty levels": **We have already provided an analysis in lines 614–616.** --- *"When extremely long values occur in English — such as in Deepseek-v3 and ChatGPT-4o-latest — it is likely that these models have developed preliminary slow-thinking capabilities in English, while such capabilities remain underdeveloped in other languages."*
>
> * Regarding "English responses are the shortest at low difficulty levels": Since these simpler problems place less demand on "slow thinking" capabilities, performance is likely to depend more on the model's ability to fit the language itself. Given that English occupies a larger proportion of the training data, models are more likely to produce concise, human-like responses in English. In contrast, low-resource languages, due to limited training exposure, often lead to less controlled and more verbose outputs (e.g. Telugu in Qwen-2.5-Max, ChatGPT-4o-latest, GPT-4.5-Preview).
>
>
> ---
>
> > F1: Clarification Needed on the Role of Human Intervention in Difficulty Labeling
>
> What we aim to emphasize is that **human intervention is required when labelling difficulty levels to the data**.
>
> From the embedding distribution, we observe that the last three difficulty levels exhibit some aggregation due to relatively small domain differences, indicating that embeddings alone do not capture difficulty distinctions effectively. As a result, automatically categorizing a batch of data by difficulty --- such as via embedding-based clustering --- becomes infeasible, so that human labelling is more reliable. This is the "labeling intervention" we want to highlight.
>
> > F2: Further Details Needed on Accuracy Assurance in Mathematical Translation: it is unclear how language experts ensure accuracy and consistency of mathematical term across languages.
>
> Regarding ***accuracy***, native speakers have a natural advantage as they can directly consult local educational materials. All second-language experts hold degrees in linguistics and have taken relevant courses in *terminology translation*, equipping them with the fundamental skills needed for terminology lookup and validation.
>
> Regarding ***consistency***, we conducted a sampling-based review of the annotated content. Specifically, we randomly sampled 60 completed annotations for each language (15 from each difficulty level), which were then reviewed by an additional native speaker assigned by the vendor we collaborated with. **The extra annotators confirmed that there were no terminology or logical errors that would significantly impact the meaning of the questions.**
>
>
> > F3: Ambiguity in the Definition of “Standard Accuracy” for Complex Answers
>
> Yes, we designed a complex and comprehensive formal parser and matching script that handles various forms of LaTeX equivalence. **As long as two expressions are judged equivalent, they are considered correct.** All evaluation code is publicly available at [https://github.com/QwenLM/PolyMath/tree/main/eval](https://github.com/QwenLM/PolyMath/tree/main/eval). Based on sampled verification, our matching rules already cover **98% of cases** (line 219).
>
> ---
>
> We hope that these explanations and additional results address your concerns. We look forward to your positive feedback.
>
> [1] DeepSeek-R1: Incentivizing Reasoning Capability in LLMs via Reinforcement Learning.
>
> [2] Qwen3 Technical Report.

---

### Official Review · Reviewer_eUks · 2025-07-03

**Rating:** 5
**Confidence:** 3

**Summary:**

This paper presents PolyMath, a multilingual mathematical reasoning benchmark across 18 languages and four difficulty levels. PolyMath ensures high-quality translations through human expert calibration to avoid LLM translation errors and employs a difficulty-weighted accuracy as the metric. Experiments on LLMs and LRMs reveal significant cross-lingual performance gaps, low input-output language consistency, and varying thinking lengths across languages, highlighting directions for improving multilingual reasoning capabilities in LRMs.

**Additional Feedback:**

Overall, I think the data collection and annotation processes of this benchmark are reasonable and valuable to the community.

**Dataset Code Accessibility:**

Partly

**Dataset Code Comments:**

Yes. I can access the datasets.

**Ethical Considerations:**

No, there are no or only very minor ethics concerns

**Final Justification:**

The author addressed most of my concerns, and I would raise my rating to 5.

**Limitations Weaknesses:**

- The difficulty level division in Section 2.1 does not seem to match the final performance (or selected data) perfectly. The Top level appears to still lack sufficient differentiation. The HLE dataset only achieves around 20% performance at maximum, yet the Top-level data here reaches over 40% performance. Could this be due to other Top-level data being too simple? Meanwhile, the difficulty between Medium and High levels is quite close, which might explain why Figure 1 shows aggregation among the three higher levels (other than the Low level), indicating their proximity in difficulty.
- How were the difficulty weights determined? It might be preferable to weight each level based on their accuracy rates, ensuring the weighted average scores across levels are more balanced.
- Has a small-scale sampling inspection been conducted on the final dataset? For example, having different annotators for the same language perform small-scale checks to derive the annotation accuracy rate of the dataset?

**Strengths Contributions:**

- The final translation is checked and revised by humans, which I consider highly feasible.
- There is indeed a shortage of multilingual reasoning datasets in the community, and I believe this dataset is valuable.

---

> ### Author Rebuttal · Authors · 2025-07-30
>
> Thanks for your recognition of our work, especially the comprehensive annotation process and its significance to the research community. Regarding the weaknesses (W1 - W3) you pointed out, we respond to them one by one as follows.
>
> ---
>
> > W1: The difficulty level division in Section 2.1 does not seem to match the final performance (or selected data) perfectly. (1) The Top level appears to still lack sufficient differentiation. The HLE dataset only achieves around 20% performance at maximum, yet the Top-level data here reaches over 40% performance. Could this be due to other Top-level data being too simple? (2) Meanwhile, the difficulty between Medium and High levels is quite close, (3) which might explain why Figure 1 shows aggregation among the three higher levels (other than the Low level), indicating their proximity in difficulty.
>
>
> Thank you very much for your careful observation and thoughtful suggestions. We will elaborate on our perspective in detail.
>
>
> > *(1) Could this be due to other Top-level data being too simple?*
>
> We deeply appreciate the outstanding contribution of HLE, which presents an exceptionally challenging dataset curated by top-tier mathematicians—a feat that is difficult to replicate.
>
> In comparison, it is true that PolyMath’s top-level difficulty does not surpass that of HLE. However, we wish to emphasize an important distinction: **PolyMath is not designed to push the upper bound of difficulty, but rather to offer a significantly more challenging benchmark than existing multilingual datasets.**. Within this multilingual context, PolyMath provides meaningful advancement, and we believe its contribution remains substantial. We also look forward to future iterations to further improve its quality and difficulty.
>
>
> > *(2) The difficulty between Medium and High levels is quite close*
>
> We do not believe that the gap between medium and high difficulty levels is close, for the following reasons:
>
> * First, although the drop in performance from medium to high difficulty is smaller than the drop between other levels, **many strong reasoning models still show a considerable decline — nearly 20 percentage points** (e.g., o1-mini: -17.7%; Deepseek-R1-671B: -19.7%; Qwen-QwQ-32B: -16.0%; Gemini-2.0-flash-thinking: -16.5%). This already represents a significant drop.
>
> * Furthermore, **our difficulty annotations are fundamentally human-centered. It is logically unsound to infer the precision of human annotations based solely on model behavior.**
>     * (a) Human expert annotations are guided by interpretable and consistent standards, whereas model behavior is inherently less controllable and often biased. Models should be treated as subjects, aligning with human-defined criteria, not as judges of them. **We believe that the validity of human standards can only be questioned through inconsistencies among human annotators.** However, as emphasized in lines 103–105, domain-expert PhD students assigned our labels and were unanimously confirmed by two additional experts. We therefore do not believe there is any bias in this labeling process.
>     * (b) As a human-defined benchmark, using human annotations as the gold standard to observe variations in model behavior provides a more meaningful and interpretable basis for discussion. We can discuss from the evaluation results that models may be less sensitive to the medium-to-high difficulty transition than to other levels, but this should not be used to question the validity of human annotations.
>
> > *(3) Figure 1 shows aggregation among the three higher levels (other than the Low level), indicating their proximity in difficulty.*
>
>
> We believe that Figure 1 does not directly explain difficulty, as **embeddings primarily capture semantic information, so it may be more closely related to domain association**. For example, the domain of "number theory" may appear in college-level exercises (medium-level), standard competition problems (high-level), and Olympiad questions (top-level), but rarely in K-12 mathematics (low-level). This suggests a higher degree of domain overlap among the last three levels. **Contrary to the interpretation that "similar difficulty leads to aggregation," the figure may better support the speculation that "embeddings do not reflect difficulty."**
>
>
> ---
>
>
> > W2: How were the difficulty weights determined? It might be preferable to weight each level based on their accuracy rates, ensuring the weighted average scores across levels are more balanced.
>
>
> Thank you very much for your insightful suggestion. We will elaborate on our perspective in detail.
>
> > *(1) How were the difficulty weights determined?*
>
>
> Our difficulty weights follow a multiplicative scheme, ensuring that level $x+1$ carries more weight than level $x$, with a consistent and smooth gradient. To maintain meaningful differentiation in practice, answering one level $x+1$ question should be worth more than answering two level $x$ questions --- otherwise, the weighting loses pracitical significance. If the multiplier were 3 or higher, one top-level question would equal 27 low-level questions, greatly undervaluing easier problems.
>
> Considering these factors, we believe a multiplier of 2 offers the best balance:
> (a) it maintains a smooth difficulty progression,
> (b) ensures practical interpretability, and
> (c) preserves the evaluative value of low-level questions.
>
>
>
> > *(2) It might be preferable to weight each level based on their accuracy rates...*
>
>
> Thank you for your insightful suggestion. However, we respectfully disagree that estimating weights based on accuracy rates is a more appropriate approach.
>
> From a human-centered perspective, a mathematician might perform well across all levels, while a middle school student may only succeed on low-level questions --- making it difficult to define a universal threshold for “human accuracy”. Similarly, using model accuracy introduces the same bias, as models vary widely in performance across levels.
>
> On the other hand, if what you mean is to adjust the weights dynamically based on model evaluation results, it would lead to a “circular argument” trap --- using the evaluation results to modify the weights used to compute those results. This undermines the fairness and objectivity of the evaluation.
>
> In contrast, **predefined weights based on difficulty are consistent with standard human testing practices**. In formal exams, such as entrance tests, question scores are determined in advance --- not retroactively based on how well a population performs. Expert test designers can better use a holistic, forward-looking approach to assign weights, aiming to reduce bias and ensure meaningful evaluation.
>
>
> ---
>
>
> > W3: Has a small-scale sampling inspection been conducted on the final dataset?
>
> Thanks for your rigorous consideration. Yes, we conducted a sampling-based review of the annotated content. Specifically, we randomly sampled 60 completed annotations for each language (15 from each difficulty level), which were then reviewed by an additional native speaker assigned by the vendor we collaborated with.
>
> **The extra annotators confirmed that there were no terminology or logical errors that would significantly impact the meaning of the questions.** However, about 10-15% of the texts contained issues related to less natural or non-colloquial expressions (e.g., "square pyramid" was translated as "正方锥" instead of the more natural "正四棱锥"). Meanwhile, they also agreed that these minor issues did not affect the overall quality. This further ensures the annotation quality of our PolyMath.
>
> ---
>
> We hope these explanations help to alleviate your concerns. We sincerely look forward to your positive feedback.

---

> > ### Comment · Reviewer_eUks · 2025-08-04
> >
> > Thanks for your thorough response, which has addressed most of my concerns. I think I'll raise my rating to 5.

---

> > > ### Author Response · Authors · 2025-08-04
> > >
> > > We are pleased that our responses have addressed your concerns, and we sincerely appreciate your support in raising your rating to 5. Thanks again for your time and thoughtful review!

---

### Official Review · Reviewer_rkf7 · 2025-07-05

**Rating:** 4
**Confidence:** 3

**Summary:**

This paper presents a significant contribution to multilingual AI evaluation through the introduction of PolyMath, a rigorously designed benchmark spanning 18 languages and four granular difficulty levels. Its major strengths lie in its comprehensive benchmark construction, which integrates human-validated translations and a novel Difficulty-Weighted Accuracy (DW-ACC) metric to fairly weight high-level problems, and its pioneering analysis of linguistic phenomena, revealing crucial insights such as the negative correlation between input-output language inconsistency and reasoning performance in models like Qwen-QwQ-32B.

However, three notable limitations merit attention. First, the analysis of language consistency (Section 4.1) focuses narrowly on only three reasoning models (Qwen-QwQ-32B, Deepseek-R1-671B, Claude-3.7-sonnet-thinking), omitting broader trends across the eight reasoning LLMs evaluated and limiting generalizability. Second, while the benchmark covers 18 languages, its per-language data scale (125 problems per difficulty level) may be insufficient for robust low-resource language evaluation, particularly given the high variance in model performance for languages like Bengali and Swahili (Table 5). Third, the proposed language control strategy (Section 4.2), though promising, lacks practical implementation guidance for real-world deployment and fails to address the computational overhead of forcing English outputs across all languages.

Despite these limitations, PolyMath establishes a valuable foundation for future multilingual reasoning research, offering both a high-quality discriminative benchmark and empirically grounded insights into cross-lingual reasoning disparities.

**Dataset Code Accessibility:**

Partly

**Ethical Considerations:**

No, there are no or only very minor ethics concerns

**Limitations Weaknesses:**

1. The analysis of language consistency (Section 4.1) focuses narrowly on only three reasoning models (Qwen-QwQ-32B, Deepseek-R1-671B, Claude-3.7-sonnet-thinking), omitting broader trends across the eight reasoning LLMs evaluated and limiting generalizability.

2. While the benchmark covers 18 languages, its per-language data scale (125 problems per difficulty level) may be insufficient for robust low-resource language evaluation, particularly given the high variance in model performance for languages like Bengali and Swahili (Table 5).

3. The proposed language control strategy (Section 4.2), though promising, lacks practical implementation guidance for real-world deployment and fails to address the computational overhead of forcing English outputs across all languages.

**Strengths Contributions:**

Its major strengths lie in its comprehensive benchmark construction, which integrates human-validated translations and a novel Difficulty-Weighted Accuracy (DW-ACC) metric to fairly weight high-level problems, and its pioneering analysis of linguistic phenomena, revealing crucial insights such as the negative correlation between input-output language inconsistency and reasoning performance in models like Qwen-QwQ-32B.

---

> ### Author Rebuttal · Authors · 2025-07-30
>
> Thank you for your recognition of our work, especially its contributions to advancing multilingual reasoning research and providing empirical insights. Regarding the three weaknesses (W1 - W3) you raised, we address them one by one as follows:
>
> ---
>
> > W1: The analysis of language consistency (Section 4.1) focuses narrowly on only three reasoning models
>
> Thank you for pointing this out. Due to closed-source restrictions, OpenAI's o-series and Google's Gemini-series has disabled access to their full thinking part. Here, we supplement the language consistency of their answer part, as well as the language consistency in both the thinking and answer parts of Qwen3:
>
>
> ### Answer Part:
>
> ||*average*|en|zh|ar|bn|de|es|fr|id|it|ja|ko|ms|pt|ru|sw|te|th|vi|
> |-|-|-|-|-|-|-|-|-|-|-|-|-|-|-|-|-|-|-|-|
> |OpenAI-o1-mini|**91.1**|100.0|99.0|85.5|91.9|80.7|96.5|88.1|92.3|83.2|98.6|87.5|90.1|95.0|84.6|89.2|98.8|82.8|95.4|
> |OpenAI-o3-mini-medium|**90.8**|100.0|97.3|83.1|91.8|86.4|98.8|94.6|80.1|87.3|99.6|85.0|89.7|90.5|82.4|84.9|96.6|88.2|97.2|
> |Gemini-2.5-pro|**92.6**|100.0|88.9|98.6|96.1|91.0|90.2|84.2|92.7|94.4|90.2|91.3|96.6|92.1|87.8|98.3|96.7|89.5|88.9|
> |Qwen3-235B-A22B-thinking|**62.3**|96.7|95.2|76.3|49.9|54.9|45.6|64.6|73.2|53.4|39.5|63.1|50.1|95.7|69.5|64.6|43.7|70.8|14.3|
>
>
>
> ### Thinking Part:
>
> ||*average*|en|zh|ar|bn|de|es|fr|id|it|ja|ko|ms|pt|ru|sw|te|th|vi|
> |-|-|-|-|-|-|-|-|-|-|-|-|-|-|-|-|-|-|-|-|
> |Qwen3-235B-A22B-thinking|**16.7**|100.0|100.0|0.0|0.0|0.0|0.0|0.0|0.0|0.1|0.0|0.2|0.0|0.0|99.2|0.2|0.1|0.0|0.0|
>
>
> From the results, we observe that
>
> * o1, o3, and Gemini-2.5-Pro exhibit relatively high language consistency in the *answer* part, which is similar to the phenomenon on Claude-3.7-Sonnet-Thinking.
>
> * In contrast, Qwen3 shows lower language consistency, especially in the *thinking* part. This can be attributed to the fact that Qwen3 was trained with English thinking data during its multilingual reasoning post-training phase.
>
>
> We will include these results and add the corresponding discussion in the next version.
>
>
> ---
>
>
> > W2: While the benchmark covers 18 languages, its per-language data scale (125 problems per difficulty level) may be insufficient for robust low-resource language evaluation, particularly given the high variance in model performance for languages like Bengali and Swahili (Table 5).
>
> Thank you for pointing out this potential robustness concern. In addition to PolyMath, we present results on two other multilingual benchmarks — **MMMLU** (knowledge domain) and **MGSM** (math domain) — covering both high-resource languages (**English** and **Chinese**) and the low-resource languages you mentioned (**Bengali** and **Swahili**), which tend to show greater performance variances in our PolyMath.
>
> *Notably, MMMLU has a larger data scale than ours, while MGSM is smaller in scale.*
>
>
> * PolyMath
>
> ||en|zh|bn|sw|
> |-|-|-|-|-|
> |QwQ-32B|52.1|47.5|44.9|41.0|
> |Deepseek-R1-671B|47.4|45.2|46.5|45.5|
> |Gemini-2.5-pro|54.5|52.9|50.8|49.4|
>
>
> * MMMLU
>
> ||en|zh|bn|sw|
> |-|-|-|-|-|
> |QwQ-32B|87.97|83.35|77.72|54.23|
> |Deepseek-R1-671B|90.25|88.19|84.06|78.51|
> |Gemini-2.5-pro|90.04|86.41|85.98|84.56|
>
>
> * MGSM
>
> ||en|zh|bn|sw|
> |-|-|-|-|-|
> |QwQ-32B|97.6|91.6|88.4|74.0|
> |Deepseek-R1-671B|98.8|92.4|91.6|92.0|
> |Gemini-2.5-pro|97.60|92.4|92.4|90.8|
>
>
> * bn and sw consistently underperform compared to en and zh on PolyMath, and this performance gap is also observed on the other two benchmarks.
> * For QwQ-32B, the gap between bn/sw and en/zh on PolyMath is substantial, and this large disparity also appears clearly on MMMLU and MGSM. In contrast, for the other two models, the performance gap between bn/sw and en/zh on PolyMath is much smaller, and this trend of reduced disparity is also reflected across the other two benchmarks.
>
>
> These results suggest that the high performance variance in bn and sw is mainly due to the models' inherent limitations in these low-resource languages, because **similar variance is also observed across other benchmarks, regardless of dataset size. Therefore, PolyMath's dataset size does not introduce unfair evaluation bias for these languages.**
>
> ---
>
> > W3: The proposed language control strategy (Section 4.2), though promising, lacks practical implementation guidance for real-world deployment and fails to address the computational overhead of forcing English outputs across all languages.
>
> Thank you for pointing out the follow-up issues related to the *language control strategy*.
>
>
> > *(1) practical implementation guidance for real-world deployment*
>
> If the goal is purely to improve inference-time accuracy, our paper already provides a lightweight strategy: simply specifying the thinking language in the prompt. This approach is plug-and-play and requires no model modification.
>
> If we consider how this phenomenon could inspire broader real-world applications, in fact, **Qwen3 has already made a successful attempt in this direction during its post-training phase.** Inspired by the observation that enforcing English thinking improves model performance, Qwen3’s multilingual reasoning data was constructed with *English thinking processes* during training. This approach helped better activate the model's long reasoning capabilities and indeed led to noticeable improvements in Qwen3's multilingual reasoning performance.
>
>
> > *(2) the computational overhead of forcing English outputs across all languages*
>
>
> We measured the thinking length of reasoning models under forcing English output across different language contexts. Taking English as the baseline (1×), we report the relative length ratios for other languages compared to English:
>
>
> ||en|zh|ar|bn|de|es|fr|id|it|ja|ko|ms|pt|ru|sw|te|th|vi|
> |-|-|-|-|-|-|-|-|-|-|-|-|-|-|-|-|-|-|-|
> |QwQ-32B|1×|0.87x|1.67x|1.19x|0.71x|1.46x|1.10x|0.99x|1.38x|1.03x|1.74x|1.29x|0.79x|0.93x|1.23x|1.59x|1.13x|0.82x|
> |Deepseek-R1-671B|1×|1.03x|1.52x|0.89x|1.11x|0.72x|1.68x|1.26x|0.93x|1.39x|1.32x|1.20x|0.85x|1.60x|1.77x|0.97x|0.75x|1.46x|
> |Claude-3.7-sonnet-thinking|1×|1.12x|1.78x|0.83x|1.26x|1.01x|0.90x|1.31x|0.72x|1.44x|1.16x|0.97x|1.36x|1.63x|1.08x|1.69x|0.76x|1.55x|
>
>
> Overall, **forcing the model to respond in English results in a controllable computational overhead, without leading to excessively long outputs.**
>
> ---
>
> We hope these additional experimental results address your concerns. We sincerely look forward to your positive feedback.

---

> > ### Author Response · Authors · 2025-08-07
> >
> > Dear Reviewer rkf7,
> >
> > Thank you for your insightful comments. In response to the three limitations you raised, we have conducted and included detailed experiments to address them.
> >
> > With that in mind, we would greatly appreciate knowing whether our responses have fully resolved your concerns and whether they might influence your final decision, as this would help facilitate convergence in the reviewer-author discussion phase.
> >
> > Thank you again for your time!
> >
> > Best,
> >
> > Authors

---

### Note · Authors · 2025-08-13

We sincerely thank all reviewers for constructive reviews. Here, we provide the reviewers, AC, and SAC with a concise summary of the discussion.

# Strengths
* **High-value and timely contribution to the reasoning research community** (Reviewer rkf7, eUks, v5eL, xMfm).
* **Novel, comprehensive dataset construction** (Reviewer rkf7, v5eL, xMfm) with
  * **High-quality human annotations** (Reviewer rkf7, eUks, v5eL)
  * **Comprehensive difficulty settings & fair weighted metric** (Reviewer rkf7, v5eL)
* **Pioneering and insightful empirical analysis** (Reviewer rkf7, v5eL, xMfm).
* **Well-written paper** (Reviewer xMfm).

# Concerns
> (1) Experimental (more objective)
- *More language consistency results (Reviewer rkf7)*: We added for all accessible outputs.
- *Computational overhead under language forcing (Reviewer rkf7)*: We reported output lengths when forcing, showing no cost issue.
- *Decoding ablation (Reviewer v5eL)*: We tested non-reasoning models with sampling strategies, showing no effect on conclusions.
- *Error analysis (Reviewer xMfm)*: We provided detailed error types and cross-model comparison.

> (2) Interpretive (more subjective)
* *Data scale robustness (Reviewer rkf7)*: We validated it via correlation with other multilingual benchmarks.
* *Difficulty differentiation (Reviewer eUks)*: We clarified human-centered annotation practices, recognized by Reviewer eUks.
* *Evaluation setup fairness (Reviewer v5eL)*: We resolved this misunderstanding, recognized by Reviewer v5eL.
* *Annotation detail confirmation (Reviewer eUks & v5eL)*: We added details, recognized by both reviewers.
* *Further analysis of experimental phenomena (Reviewer v5eL & xMfm)*: We added, recognized by Reviewer v5eL.

# Conclusion
During the discussion phase, we received the following feedback:
- **Reviewer v5eL** recognized rebuttal and intends to raise rating to **4**.
- **Reviewer eUks** recognized rebuttal and intends to raise rating to **5** (but without official edit as of Aug 12, 20:00, AOE, ***we kindly ask AC to consider this in the decision***).
- **Reviewers rkf7 and xMfm** have not responded, but given their positive scores (**4 & 5**) and their main concerns are experimental, ***we believe our results address their issues***.

Given that all reviewers provided positive ratings and no further issues were raised, we believe our rebuttal was effective, and PolyMath can provide substantial value to the research community.

Best regards,
Submission568 Authors

---

### Decision · Program_Chairs · 2025-09-18

**Decision:**

Accept (poster)

**Comment:**

This paper presents a benchmark for a multilingual mathematical reasoning. It covers 18 languages and 4 difficulty levels.

Strengths
- The dataset contains significant language diversity.
- This quality of dataset is high because the translation is annotated by humans.
- Comprehensive experiments has been provided to reveal the performance disparity across different languages in the state-of-the-art reasoning models.
- The paper also presents some interesting and insightful findings, including low input-output language consistency, and significant thinking length difference by language.
- The paper also proposes a language control strategy with promising results to improve the multilingual reasoning capabilities in LLMs.

Weaknesses
- A reviewer has suggested more models for experiments on language consistency results. The author has provided more results based on OpenAI o1, o3, Gemini-2.5-pro, Qwen3-235B.
- Another concern is on the low data size per language, only 125 problems per difficulty level, which is inadequate for low-resource languages with high performance variance. The author has provided more results two other multilingual benchmarks MMMLU and MGSM.
- Practicality and computation overhead of the language control strategy are not thoroughly discussed. The author also provided justification and empirical analysis of their approach to address this issue.
- Some design choices and experimental details are not well explained. In particular, these include the calibration difficulty level division, difficulty weights, sampling inspection, and fairness of evaluation setup between reasoning and non-reasoning models. The author provided further explanation to address this.
- More explanations about experiment results can be provided. These include explanations in thinking length analysis, more error analyses, the different patterns between non-reasoning and reasoning LLMs.

Overall, this dataset is a timely and valuable contribution to reasoning research by calling for progress across different languages, and the experiments are comprehensive with insightful analysis. Author rebuttal is overall effective to address reviewers' concerns, and authors are highly encouraged to incorporate the discussion and feedback.